# GENerator of reduced Organic Aerosol mechanism (GENOA v1.0): An automatic generation tool of semi-explicit mechanisms

Zhizhao Wang[1,2], Florian Couvidat[2], and Karine Sartelet[1]

[1]CEREA, École des Ponts ParisTech, EDF R&D, 77 455 Marne-la-Vallée, France.
[2]INERIS, Institut National de l'Environnement Industriel et des Risques, Verneuil-en-Halatte, France

**Correspondence:** zhizhao.wang@enpc.fr

**Abstract.** This paper describes the GENerator of Reduced Organic Aerosol Mechanisms (GENOA) that produces semi-explicit mechanisms for simulating the formation and evolution of secondary organic aerosol (SOA) in air-quality models. Using a series of predefined reduction strategies and evaluation criteria, GENOA trains and reduces SOA mechanisms from explicit chemical mechanisms (e.g., the master chemical mechanism (MCM)) under representative atmospheric conditions. As a consequence, these trained SOA mechanisms can preserve the accuracy of explicit VOC mechanisms on SOA formation (e.g., molecular structures of crucial compounds, effect of non-ideality and hydrophilic/hydrophobic partitioning of aerosols), with a size (in terms of reaction and species numbers) that is manageable for three-dimensional (3-D) aerosol modeling (e.g., regional chemical transport models). Applied to the degradation of sesquiterpenes (as $\beta$-caryophyllene) from MCM, GENOA builds a concise SOA mechanism (2% of the MCM size), consisting of 23 reactions and 15 species, six of them being condensable. The generated SOA mechanism has been evaluated for its ability to reproduce SOA concentrations under varying atmospheric conditions encountered over Europe, with an average error lower than 3%.

## 1 Introduction

Atmospheric aerosols attract attention due to their effects on climate and human health: they change the earth's radiation balance and cloud formation (Ramanathan et al., 2001; McNeill, 2017); they trigger a wide variety of acute and chronic diseases (Breysse et al., 2013). Because the effects of aerosols on health depend on their size and composition (Schwarze et al., 2006), adequate representations of aerosol composition, mass, and number concentrations are required in air quality models (AQMs).

Besides being directly emitted, aerosols can be secondary, i.e., formed in the atmosphere through chemical reactions and gas-particle mass transfer. Based on the chemical composition, they can be further divided into secondary inorganic aerosol (SIA) and secondary organic aerosol (SOA). SOA, which represents a significant fraction of aerosols (e.g., Gelencsér et al., 2007), is largely formed by the condensation of the oxidation products from the degradation of volatile organic compounds (VOC). As SOA formation involves multiple processes such as the emission of SOA precursor gases, VOC gas-phase chemistry, gas-to-particle partitioning (Kanakidou et al., 2005; Hallquist et al., 2009), there are great complexity and uncertainty to accurately predict SOA formation with the simplified representations currently used in air quality models (Porter et al., 2021).

The state of knowledge on VOC chemistry can be reflected by explicit gas-phase chemical mechanisms, which contain all known essential reaction pathways in VOC degradation. For instance, Jenkin et al. (1997); Saunders et al. (2003) developed the near-explicit Master Chemical Mechanism (MCM), which describes detailed gas-phase chemical processes related to VOC oxidation. Another example is the Generator for Explicit Chemistry and Kinetics of Organics in the Atmosphere (GECKO-A) (Aumont et al., 2005), which uses a prescribed protocol to assign complete reactions pathways and kinetic data to the degradation of VOCs. Explicit mechanisms represent the current understanding of atmospheric chemistry, including information about reaction pathways, kinetics data, and chemical structures (which may be used to deduce thermodynamic properties based on structure-activity relationships). The MCM mechanism has been used by two-dimensional (2-D) Lagrangian models to simulate the chemical evolution of major air pollutants and some SOAs in plumes (e.g., Evtyugina et al., 2007; Sommariva et al., 2008; Zhang et al., 2021). Moreover, it has been used for simulating the formation of more complex SOAs at a regional level in three-dimensional (3-D) models over a few weeks (e.g., modified MCM with 4642 species and 13,566 reactions in the simulations of Ying and Li (2011), and with 5727 species and 16,930 reactions in the simulations of Li et al. (2015)). Even so, explicit mechanisms of that size are too computationally intensive to be widely employed in 3-D AQMs for SOA formation. For computational efficiency, AQMs generally use implicit gas-phase chemical mechanisms. Two major approaches are frequently adopted to build implicit chemical mechanisms:

– The lumped-species approach, which gathers into one surrogate compounds with analogous formulas and properties (e.g., SAPRC-07 Carter (2010), RACM2 Goliff et al. (2013))

– The carbon-bond or lumped-structure approach, which assumes that organic molecules have chemical behaviors equivalent to those of their decomposed functional groups (e.g., CB05 Sarwar et al. (2008))

Implicit gas-phase mechanisms have been developed and validated to simulate the concentrations of oxidants and other conventional air pollutants such as ozone and $NO_2$. In these mechanisms, VOCs have been grouped into a limited number of model species because of computational considerations, and the SOA formation is usually not considered.

To complete implicit gas-phase mechanisms, implicit SOA mechanisms have been developed (Kim et al., 2011), which model the SOA formation specifically without modifying ozone and radical concentrations. In 3-D modeling, implicit SOA mechanisms or parameterizations are usually added to implicit gas-phase mechanisms, conserving the oxidant chemistry of the implicit gas-phase mechanism.

Implicit SOA mechanisms are often established based on experimental data from smog chamber experiments to represent the formation and evolution of SOA, such as the two-product empirical SOA model (Odum et al., 1996) and the volatility basis set (VBS) that splits VOC oxidation products into a uniform set of volatility "bins" (Donahue et al., 2006). In the VBS approach, the successive evolution of oxidation products by aging is determined regardless of the chemical composition and structure of the species. Another approach is based on the molecular surrogate approach (e.g., Griffin et al., 2003; Pun et al., 2006; Couvidat et al., 2012). Similarly to the gas-phase chemistry lumped-species approach, the VOC oxidation products are represented via the formation of a few SOA surrogates that are attached to a molecular structure (assumed to be representative of a myriad of semi-volatile compounds). By attaching a molecular structure to the surrogate, several processes otherwise not

accounted for (like non-ideality, hygroscopicity, condensation on the aqueous phase of particles) can be represented in this approach. However, the choice of adequate molecular structures, which could be highly uncertain, is crucial and requires a precise estimation.

The computation of thermodynamic properties of aerosol (e.g., hydrophilicity, hydrophobicity, viscosity), additionally, requires knowing the molecular composition to take into account the whole complexity of the gas-particle partitioning (Kim et al., 2019). Therefore, tracking the whole complexity of the formation and aging of SOA with implicit SOA mechanisms can be problematic as it may not account for (or may oversimplify) some processes, such as non-ideality. These processes may be particularly important for explaining the non-linear relationship between the emissions of pollutants and the formation of aerosols (Huang et al., 2020).

Since the current SOA representations in AQMs are implicit and may not accurately reflect the true SOA formation process, there is a need for improvement. This has led to the development of semi-explicit mechanisms of condensed sizes. The development of semi-explicit mechanisms is a compromise between the high computational time of explicit mechanisms and the lack of accuracy in the representation of chemical phenomena in the implicit SOA mechanisms. They are generated by reducing explicit mechanisms to a level of complexity suitable to the computational constraints of AQMs. Recent developments of reduced mechanisms include the Common Representative Intermediates (CRI) mechanism (Jenkin et al., 2008; Watson et al., 2008; Khan et al., 2017) from the MCM reduction (Szopa et al., 2005), and the volatility basis set – Generator for Explicit Chemistry and Kinetics of Organics in the Atmosphere (VBS-GECKO) (Lannuque et al., 2018) from a GECKO-A reduction. However, the reduced mechanisms mentioned above do not track the detailed molecular structure of surrogates, but only consider some of their specific properties:

- CRI characterizes surrogates by their number of carbon-carbon and carbon-hydrogen bonds, which are reactive in the NO-to-$NO_2$ conversions concerning ozone formation.

- VBS-GECKO groups organic surrogates by their volatility, as in the VBS approach (Donahue et al., 2006).

This study presents the development of the first version of the GENerator of reduced Organic Aerosol mechanism (GENOA) that generates customized semi-explicit chemical mechanisms appropriate for AQM from explicit mechanisms, using surrogates assigned to molecular structures. As described in Sect. 2, the new reduced mechanisms can effectively and efficiently reproduce the complexity of gas-phase oxidation by training under various atmospheric conditions and the non-ideality of gas-particle partitioning using a molecular structure-preserving approach. GENOA also provides practical user-defined options, enabling users to specify the required reduction scale or accuracy. For gas/particle partitioning, a 0-D box model SSH-aerosol (Sartelet et al., 2020) is modified and coupled with GENOA to simulate aerosol concentrations. With SSH-aerosol, the effects of mass transfer between gas-phase and organic/aqueous phases, hygroscopicity, and non-ideality are taken into account in the reduction.

The application of GENOA to the MCM degradation scheme of $\beta$-caryophyllene (BCARY) (Jenkin et al., 2012) is described in Sect. 3. $\beta$-caryophyllene is selected to demonstrate the GENOA algorithm, because it is one of the most abundant and representative sesquiterpenes (SQT). Sesquiterpenes are a well-known source of SOAs (Hellén et al., 2020; Tasoglou and

Pandis, 2015), and their degradation mechanism (as BCARY) is well documented in the near-explicit MCM mechanism (Jenkin et al., 2012). Studies have also compared SOA yields simulated using the MCM mechanism to chamber data for sesquiterpenes

(e.g., Xavier et al., 2019). BCARY is, therefore, an ideal candidate for model development and demonstration of the reduction methodology. In this paper, the near-explicit MCM BCARY degradation scheme serves as a reliable benchmark for GENOA. The experiment data from Tasoglou and Pandis (2015); Chen et al. (2012) are also compared to the newly developed reduced mechanism in Appendix A. Finally, the conclusion is drawn in Sect. 4.

## 2 Model development

The GENerator of Reduced Organic Aerosol Mechanisms (GENOA) is an algorithm that generates semi-explicit chemical mechanisms focusing on SOA formation. The generated semi-explicit mechanisms are designed to preserve the accuracy of explicit mechanisms for SOA formation, while keeping the number of reactions/species low enough to be suitable for large-scale modeling, particularly in 3-D AQMs. The focus of the semi-explicit mechanism is solely on the accurate modeling of SOA. Because ozone, major radicals, and other inorganics are also affected by inorganic and other VOC chemistry, their

concentrations are not tracked with the semi-explicit mechanism. Instead, they are simulated using existing implicit gas-phase chemical mechanisms.

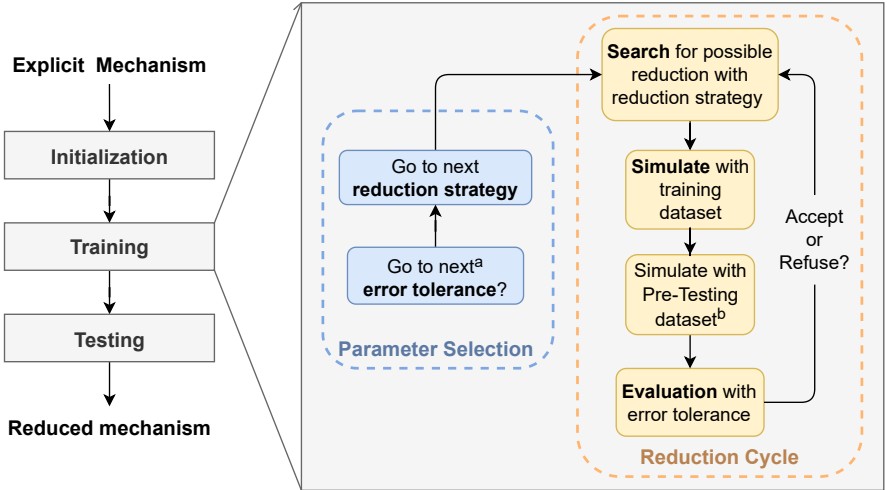

**Figure 1.** Flow chart indicating the three major procedures in GENOA and illustrating the main execution of the training section.

[a] GENOA uses the first value of the targeted variables for initialization, and passes to the next values for subsequent parameter updates.

[b] Activated under certain circumstances.

As illustrated in Fig. 1, the processes in GENOA can be divided into two main sections: training and testing. The training section, as detailed in Fig. 1, can be divided into two parts:

– Parameter selection, where the parameters to be used in the reduction cycle are selected automatically by GENOA from user-specified or preset values.

– Reduction cycle, where the actual reduction of the mechanism occurs.

In the parameter selection, GENOA first assigns the error tolerance defined as the largest acceptable error induced by each change in the mechanism (see Sect. 2.5), and then employs one of the reduction strategies along with its required parameters (see Sect. 2.2).

 Afterward, in the reduction cycle, GENOA searches for potential reductions according to the selected reduction strategy. The new mechanism with the first found reduction is then simulated over the conditions from the training dataset (a limited set of conditions used through all the reduction processes, Sect. 2.3.1) or from the pre-testing dataset (a more extensive set of conditions used only at the end of the reduction process see Sect. 2.3.2). The simulated total SOA concentrations are then compared to those simulated with the reference mechanism, where the differences are used to evaluate the potential reduction  (see Sect. 2.5). In case the SOA differences are under the pre-defined error tolerances, the mechanism with the current reduction is accepted and serves as the basis for the next search for reduction. If the reduction is refused, the following reduction attempt starts with the previously validated mechanism. Once no more reduction is found, the current reduction cycle ends. The next step is either selecting the subsequent error tolerance and/or reduction strategy in the next parameter selection, or terminating the GENOA training section. Finally, the performance evaluation of the final reduced mechanism is evaluated under a variety  of environmental conditions denoted as the testing dataset (see Sect. 2.3.3). The 0-D aerosol model SSH-aerosol is used to simulate SOA concentration and composition, which is required in all the GENOA sections (e.g., the initialization of reduction parameters, and the evaluation of the reduced mechanism).

## 2.1 Pre-reduction

A pre-reduction process is conducted on the original MCM mechanism before it is used as the reference mechanism for the  reduction. This process skips extremely fast unimolecular reactions (i.e., the reaction rate constant of $10^6$ s$^{-1}$ corresponding to a lifetime of 1 $\mu$s) to avoid numerical problems. For computational efficiency, the process also combines elementary reactions with the same reactants into combined reactions with non-integer stoichiometric coefficients.

An example is shown in Table 1, where the original MCM reactions No. 1 to 7 have first been merged into the combined reactions No. 8 to 10. The pre-reduction compacts the reaction list (from 1 626 to 1 242 reactions), improving the reduction  efficiency. The pre-reduction also skips two biradicals (i.e., BCALOOA and CH2OOF) that are extremely reactive and disintegrate instantaneously with a kinetic rate coefficient of $10^6$ s$^{-1}$. As a result, reactions No. 8 to 10 can then be repented by one reaction No. 11, whose kinetic rate coefficient corresponds to that of the reaction producing the skipped species (in this case, the ozonolysis of BCAL, reaction No. 9).

## 2.2 Reduction strategies

 GENOA supports four types of reduction strategies:

**Table 1.** Reactions before and after pre-reduction , where MCM species BCALOOA and CH2OOF are skipped over by their degradation products. [a]

| No. | Reaction | Kinetic rate coefficient [b] |
|-----|----------|------------------------------|
| 1 | BCAL + $O_3$ → BCALOOA + HCHO | $1.1 \times 10^{-16} \times 0.670$ |
| 2 | BCAL + $O_3$ → BCLKET + CH2OOF | $1.1 \times 10^{-16} \times 0.330$ |
| 3 | BCALOOA → BCALOO | $1.0 \times 10^{6} \times 0.500$ |
| 4 | BCALOOA → C146O2 + OH | $1.0 \times 10^{6} \times 0.500$ |
| 5 | CH2OOF → CH2OO | $1.0 \times 10^{6} \times 0.370$ |
| 6 | CH2OOF → CO | $1.0 \times 10^{6} \times 0.500$ |
| 7 | CH2OOF → $HO_2$ + CO + OH | $1.0 \times 10^{6} \times 0.130$ |
| 8 | BCAL + $O_3$ → 0.67 BCALOOA + 0.33 BCLKET <br> + 0.33 CH2OOF + 0.67 HCHO | $1.1 \times 10^{-16}$ |
| 9 | BCALOOA → 0.5 BCALOO + 0.5 C146O2 + 0.5 OH | $1.0 \times 10^{6}$ |
| 10 | CH2OOF → 0.37 CH2OO + 0.63 CO + 0.13 $HO_2$ + 0.13 OH | $1.0 \times 10^{6}$ |
| 11 | BCAL + $O_3$ → 0.5 BCALOO + 0.5 C146O2 + 0.37 CH2OO <br> + BCLKET + HCHO + 0.13 $HO_2$ <br> + 0.63 CO + 1.13 OH | $1.1 \times 10^{-16}$ |

[a] MCM v3.3.1. The molecular structures of all mentioned MCM species can be found in Fig. C1.

[b] unit in $s^{-1}$ for unimolecular reactions and $molecule^{-1} cm^3 s^{-1}$ for bimolecular reactions.

- Removing: reactions, species, or gas-particle partitioning with negligible effects on SOA formation are removed from the mechanism.

- Jumping: one compound is substituted by its oxidation product, as if the compound had been "jumped over" in the reaction pathway.

- Lumping: compounds with similar properties are combined to form a new compound.

- Replacing: one compound is replaced by another existing compound with similar properties.

The reduction strategies are illustrated with examples from the BCARY reduction in sections 2.2.1 to 2.2.4. A detailed list of all the options and parameters controlling the BCARY reduction is summarized in the supplementary material.

For the BCARY reduction, the reduction strategies are employed in the following order: removing reactions, jumping, lumping, replacing, removing species, and finally removing gas-particle partitioning. The reduction strategies are ordered based on their potential influences on the mechanism. The first applied strategies, removing reactions and jumping, trim trivial reactions and species without altering the properties of the species. They are followed by lumping and replacing (as an extension to lumping), which refine the mechanisms considerably by merging the species and reactions involved. Afterward, the "removing

species" strategy attempts to delete all merged and unmerged species. Finally, the strategy of removing gas-particle partition-
ing is applied in order to remove the partitioning of condensable species, which cannot be removed by removing species. This
current order has been tested and found to be efficient for the BCARY mechanism, but it can be changed by the user along with
other user-chosen parameters.

### 2.2.1 Removing strategy

The removing strategy assumes that chemical reactions and/or species with a low probability of contributing to the formation
and evolution of SOA can be eliminated from the mechanism. In general, three types of removing are applied depending on
the removed subject:

- Removing reactions.

- Removing compounds in both the gaseous and particle phases (completely removing a species from the scheme).

- Removing the gas-particle partitioning of semi-volatile compounds (consider the semi-volatile compounds as VOCs that
  do not condense to the particle phase, but retain their gas-phase chemistry).

There is no particular restriction to exclude species from the reduction attempt via the strategy of removing compounds
or removing gas-particle partitioning. However, for removing reactions, a threshold on the branching ratio of the reaction is
applied to the reduction. The branching ratio is defined as the ratio of the destruction rate of one reaction to the sum of the
destruction rates of all reactions of the targeted species. In the BCARY reduction, a maximum branching ratio ($B_{rm}$) is defined
as a restriction criterion. All reactions with an hourly branching ratio (averaged over the training conditions) under this value
(reactions that are likely to have a minimal effect on SOA formation) are considered candidates for removal.

To avoid over-reduction, a small $B_{rm}$ is applied at the beginning of reduction. After going through the reductions for all
reduction strategies, the value of $B_{rm}$ is then incremented. In the reduction of BCARY, an ascending list of $B_{rm}$ values equal
to 5 %, 10 %, 50 % is employed, which is changed to 10 %, 50 %, and 100 % at the late stage (explained in Sect. 2.5). When
$B_{rm}$ equals 100 %, GENOA evaluates the removal of each reaction.

### 2.2.2 Jumping strategy

The jumping strategy relies on the assumption that compounds can be skipped in successive reactions, as long as it does
not adversely impact the SOA concentration. In other words, the predecessor of an organic compound may directly form its
destruction products. The jumping strategy is perfectly suited to intermediate compounds whose fast degradation may cause
numerical stiffness, commonly including radicals such as oxy radicals (RO) or alkoxy radicals (ROO), as well as Criegee
intermediates.

As shown in Table 2, the Criegee intermediate BCALOO formed during the ozonolysis of BCAL (reaction No. 11 in Table 1)
is jumped over to its only destruction product BCLKET. Consequently, reactions No. 12 to 16 are removed, and reaction No.
11 is updated to reaction R1 ("R" for reaction after reduction strategy). Currently, the jumping strategy is considered when

**Table 2.** Reactions before and after the jumping strategy, where MCM species BCALOO is jumped over by its degradation product BCLKET.

| No. | Reaction | Kinetic rate coefficient [a] |
|---|---|---|
| 12 | $BCALOO + CO \rightarrow BCLKET$ | $1.2 \times 10^{-15}$ |
| 13 | $BCALOO + NO \rightarrow BCLKET + NO_2$ | $1.0 \times 10^{-14}$ |
| 14 | $BCALOO + NO_2 \rightarrow BCLKET + NO_3$ | $1.0 \times 10^{-15}$ |
| 15 | $BCALOO + SO_2 \rightarrow BCLKET + SO_3$ | $7.0 \times 10^{-14}$ |
| 16 | $BCALOO \rightarrow BCLKET + H2O2$ | $1.4 \times 10^{-17} \times [H_2O]$ |
| R1[b] | $BCAL + O_3 \rightarrow 1.5\ BCLKET + 0.5\ C146O2$ $+ 0.37\ CH2OO + HCHO$ $+ 0.13\ HO_2 + 0.63\ CO + 1.13\ OH$ | $1.1 \times 10^{-16}$ |

[a] $[H_2O]$ is the concentration of $H_2O$. [b] Reaction R1 is updated from reaction No.11 of Table 1.

the destruction of a single compound (to be jumped) results in the production of a single compound (jumping). The difference in carbon numbers between reduced species can not exceed three in order to prevent significant differences in organic mass before and after jumping.

There are similarities between reduction by jumping and pre-reduction in the sense that both can jump reactions without affecting organic compounds. However, the two processes serve different purposes, as pre-reduction is intended to provide a reliable reference mechanism for training, whereas jumping is used in training to search for possible reductions. On the one hand, the current pre-reduction only reduces very fast degraded radicals that undergo a single unimolecular reaction with a constant kinetic rate coefficient (e.g., no temperature effect). In this case, one species may lead to several degradation products. As these reactions are extremely fast and independent of atmospheric conditions, they only cause numerical issues in simulation and should be removed from the reference mechanism. On the other hand, jumping may be relatively slow or affected by environmental conditions, and therefore, an evaluation is necessary. Jumping is currently limited from one species to another at a time. As shown in Table 2, the degradation of BCALOO into BCLKET involves five bimolecular reactions, which may affect SOA formation under different atmospheric conditions (e.g., with different inorganic concentrations and relative humidity (RH)).

### 2.2.3 Lumping strategy

The lumping strategy (i.e., lumping different compounds into a single surrogate compound) assumes that organic compounds with similar chemical structures may exhibit similar properties and undergo similar physico-chemical processes and may therefore be lumped together. With lumping, both the number of species and reactions decrease.

The lumping strategy is illustrated by the comparison of Table 3 (reactions before lumping) and Table 4 (reactions after lumping). In this example, a total of 13 chemical reactions (No. 17 to 29) involving three organic compounds are reduced to five reactions (production reaction R2 and four destruction reactions R3 to R6 of the new surrogate).

**Table 3.** Explicit reactions of MCM species BCAO2, BCBO2, BCCO2 in the degradation scheme of $\beta$-caryophyllene (BCARY).

| No. | Reaction [a] | Kinetic rate coefficient [b] |
|---|---|---|
| 17 [c] | BCARY + OH → 0.408 BCAO2 + 0.222 BCBO2 + 0.37 BCCO2 | $1.97 \times 10^{-10}$ |
| 18 | BCAO2 + HO$_2$ → BCAOOH | KAHO2 = KRO2HO2×0.975 |
| 19 | BCAO2 + NO → 0.753 BCAO + 0.753 NO$_2$ + 0.247 BCANO3 | KANO = KRO2NO |
| 20 | BCAO2 + NO$_3$ → BCAO + NO$_2$ | KANO3 = KRO2NO3 |
| 21 | BCAO2 + RO$_2$ → 0.7 BCAO + 0.3 BCAOH | KARO2 = 9.2× $10^{-14}$ |
| 22 | BCBO2 + HO$_2$ → BCBOOH | KBHO2 = KRO2HO2×0.975 |
| 23 | BCBO2 + NO → 0.753 BCBO + 0.753 NO$_2$ + 0.247 BCBNO3 | KBNO = KRO2NO |
| 24 | BCBO2 + NO$_3$ → BCBO + NO$_2$ | KBNO3 = KRO2NO3 |
| 25 | BCBO2 + RO$_2$ → 0.6 BCBO + 0.2 BCAOH + 0.2 BCBCO | KBRO2 = 8.8× $10^{-13}$ |
| 26 | BCCO2 + HO$_2$ → BCCOOH | KCHO2 = KRO2HO2×0.975 |
| 27 | BCCO2 + NO → 0.753 BCCO + 0.753 NO$_2$ + 0.247 BCCNO3 | KCNO = KRO2NO |
| 28 | BCCO2 + NO$_3$ → BCCO + NO$_2$ | KCNO3 = KRO2NO3 |
| 29 | BCCO2 + RO$_2$ → 0.7 BCCO + 0.3 BCCOH | KCRO2 = 9.2× $10^{-14}$ |

[a] Species RO$_2$ represents the sum of all peroxy radicals.

[b] The same symbols is used to demonstrate the reduction strategies shown in Table 4,5. The precise values of kinetic rate coefficients (i.e., KRO2HO2, KRO2NO, and KRO2NO3) can be found on the MCM website, in the unit molecule$^{-1}$cm$^3$s$^{-1}$.

[c] Reaction No. 17 shows the production of BCAO2, BCBO2, and BCCO2, while the other reactions (No. 18 to 29) depict their destruction processes.

As demonstrated in the tables, the organic compounds BCAO2, BCBO2, and BCCO2 from the original MCM scheme are the peroxy radicals formed from the OH-initiated oxidation of $\beta$-caryophyllene (Table 3). It is evident from their structures (shown in Fig. C1) that they are isomers and may share similar chemical properties. When applying the lumping strategy, BCAO2, BCBO2, and BCCO2 are merged into a new surrogate named "mBCAO2" (Table 4). Additional lumping examples are provided in Appendix C1, describing the lumping of compounds with differing structural groups derived from different oxidation reactions.

The key parameter that drives the reduction accuracy is the weighting ratio $f_w$ of lumping, corresponding to the weight of the original species in the new surrogate compound. As detailed in Table 4, $f_w$ is computed as a function of the chemical lifetime $\tau$ following the computation of Seinfeld and Pandis (2016), and the reference concentrations $C_r$ that are the arithmetic mean concentrations calculated from 0-D simulations using the explicit VOC mechanism. Both $\tau$ and $C_r$ are based on averages of simulations across all training conditions. The properties of the new surrogate compound (e.g., molecular structure, saturation vapor pressure, molar mass, degradation kinetics) are estimated by weighing the properties of the initial compounds, while the stoichiometric coefficients and the kinetic rate coefficient of the new reaction are obtained by weighing those of the initial reactions.

**Table 4.** Reduced reactions of Table 3 and the computation of the weighting ratios, in case of lumping BCAO2, BCBO2, and BCCO2 into a new surrogate mBCAO2. [a]

| No. [b] | Lumped [c] | Reaction [d] | Kinetic rate coefficient |
|---|---|---|---|
| R2 | 17 | BCARY + OH → mBCAO2 | $1.97 \times 10^{-10}$ |
| R3 | 18,22,26 | mBCAO2 + HO2 → $f_{w,a}$ BCAOOH + $f_{w,b}$ BCBOOH + $f_{w,c}$ BCCOOH | $f_{w,a}$ KAHO2 + $f_{w,b}$ KBHO2 + $f_{w,c} \times$ KCHO2 |
| R4 | 19,23,27 | mBCAO2 + NO → 0.753×($f_{w,a}$ BCAO + $f_{w,b}$ BCBO + $f_{w,c}$ BCCO) + 0.247×($f_{w,a}$ BCANO3 + $f_{w,b}$ BCBNO3 + $f_{w,c}$ BCCNO3) + 0.753×($f_{w,a}$+ $f_{w,b}$+ $f_{w,c}$) NO2 | $f_{w,a}$ KANO + $f_{w,b}$ KBNO + $f_{w,c}$ KCNO |
| R5 | 20,24,28 | mBCAO2 + NO3 → $f_{w,a}$ BCAO + $f_{w,b}$ BCBO + $f_{w,c}$ BCCO + ($f_{w,a}$+ $f_{w,b}$+ $f_{w,c}$) NO2 | $f_{w,a}$ KANO3 + $f_{w,b}$ KBNO3 + $f_{w,c} \times$KCNO3 |
| R6 | 21,25,29 | mBCAO2 + RO2 → 0.7×$f_{w,a}$ BCAO + 0.8×$f_{w,b}$ BCBO + 0.7×$f_{w,c}$ BCCO + 0.2×$f_{w,b}$ BCBCO + (0.3×$f_{w,a}$ + 0.2×$f_{w,b}$) BCAOH + 0.3×$f_{w,c}$ BCCOH | $f_{w,a}$ KARO2 + $f_{w,b}$ KBRO2 + $f_{w,c} \times$KCRO2 |

| Symbol [d] | Meaning | Computation |
|---|---|---|
| $C_{r,a}$ | reference concentration of BCAO2 [e] | average BCAO2 concentrations from five-day 0-D simulations under training dataset |
| $C_{r,b}$ | reference concentration of BCBO2 | average BCBO2 concentrations from five-day 0-D simulations under training dataset |
| $C_{r,c}$ | reference concentration of BCCO2 | average BCCO2 concentrations from five-day 0-D simulations under training dataset |
| $\tau_a$ | chemical lifetime of BCAO2 [f] | 1/(KAHO2 [HO2] + KANO [NO] + KANO3 [NO3] + KARO2 [RO2]) |
| $\tau_b$ | chemical lifetime of BCBO2 | 1/(KBHO2 [HO2] + KBNO [NO] + KBNO3 [NO3] + KBRO2 [RO2]) |
| $\tau_c$ | chemical lifetime of BCCO2 | 1/(KCHO2 [HO2] + KCNO [NO] + KCNO3 [NO3] + KCRO2 [RO2]) |
| $f_{w,a}$ | weighting ratio of BCAO2 | $\tau_a C_{r,a}/(\tau_a C_{r,a} + \tau_b C_{r,b} + \tau_c C_{r,c})$ |
| $f_{w,b}$ | weighting ratio of BCBO2 | $\tau_b C_{r,b}/(\tau_a C_{r,a} + \tau_b C_{r,b} + \tau_c C_{r,c})$ |
| $f_{w,c}$ | weighting ratio of BCCO2 | $\tau_c C_{r,c}/(\tau_a C_{r,a} + \tau_b C_{r,b} + \tau_c C_{r,c})$ |

[a] Name of new surrogate contains the letter "m" revealing lumping and the name of the relatively dominant lumped species. This notation of lumping is used hereafter.

[b] reaction number after lumping, where reactions R3 to R6 preserve the destruction of BCAO2, BCBO2, and BCBO2, and reaction R2 presents the production.

[c] reaction numbers before lumping as presented in Table 3.

[d] subscript a, b, and c stands for BCAO2, BCBO2, and BCCO2, respectively.

[e] The calculation method also applies to other BCARY derived organics.

[f] [X] in the calculations is the reference concentration of radical and other inorganic species, where X is HO2, NO, NO3, or RO2 in this case. For radicals derived from the SOA precursor, the reference concentration is the produced concentration without considering their rapid destruction.

Chemical lifetimes and reference concentrations may be close for species that share similar structures and undergo analogous reactions. In cases where these species originate from the same reaction, they can be lumped directly, with the branching ratios of the formation reaction serving as weighting ratios. As an example, BCAO2, BCBO2, and BCCO2 undergo equivalent reactions, with the exception of the $RO_2$ reaction of BCBO2. Since the BCARY degradation is not much sensitive to $RO_2$, BCAO2, BCBO2, and BCCO2 can be lumped together with $f_{w,a}$, $f_{w,b}$, and $f_{w,c}$ equal to the branching ratios of reaction

No.17, i.e., 0.408, 0.222, and 0.37, respectively.

  Most lumping involves species that are not isomers and undergo different reactions, which makes lumping multiple species at the same time highly uncertain. Therefore, in practice, GENOA attempts to lump only two species in a single reduction in order to ensure the effectiveness of computation. A lumping of multiple species can be achieved by combining several reductions (e.g., first lumping BCAO2 with BCCO2 to form mBCAO2, and then lumping BCBO2 into mBCAO2).

In BCARY reduction, lumping is subject to certain restrictions:

- No lumping between a compound and its oxidation products.

- Compounds with specific structural groups sharing common chemical behavior may be more appropriately merged together. Thus, compounds containing the following functional groups can only be lumped with compounds containing the same groups: peroxyacetyl nitrates (PAN), organic nitrates (RONO2), organic radicals (R), oxy radicals (RO), peroxy

radicals ($RO_2$), carboxylic acids (RC(O)OH), percarboxylic acids (RC(O)OOH).

- The difference in the molecular weight should be negligible (i.e., smaller than 100 g mol$^{-1}$).

- The difference in the carbon number should be no more than two.

- The difference in the chemical lifetime should be less than 10-fold.

- Lumping is not considered for biradicals (ROO) that degrade rapidly into closed shell molecules, as jumping is consid-

ered to be more appropriate for these compounds.

The difference in saturation vapor pressure between lumpable condensables is not explicitly restricted in BCARY reduction. However, it is implicitly considered, as GENOA searches and attempts to lump species with similar saturation vapor pressures first. Nonetheless, the user can activate the option to limit the range of saturated vapor pressure differentials between lumpable condensables, along with other user-chosen reduction options listed in the supplementary material.

**2.2.4 Replacing strategy**

The replacing strategy assumes that a compound with a negligible contribution to SOA formation can be substituted by a compound having a similar structure or undergoing the same reactions. In comparison to lumping, the replacing strategy reduces the number of reactions/species without creating new surrogate species.

  Table 5 illustrates a reduction occurring via the replacing strategy (to be compared to the original mechanism in Table 3),

assuming that BCAO2 is predominant in SOA formation. By substituting both BCBO2 and BCCO2 with BCAO2, the OH

**Table 5.** Reduced reactions of Table 3, in case of replacing BCBO2 and BCCO2 with one existing species BCAO2.

| No.[a] | Replaced | Reaction | Kinetic rate coefficient |
|---|---|---|---|
| R2' | 17 | BCARY + OH → BCAO2 | $1.97 \times 10^{-10}$ |
| R3' | 18,22,26 | BCAO2 + HO$_2$ → BCAOOH | KAHO2 |
| R4' | 19,23,27 | BCAO2 + NO → 0.753 BCAO + 0.753 NO$_2$ + 0.247 BCANO3 | KANO |
| R5' | 20,24,28 | BCAO2 + NO$_3$ → BCAO + NO$_2$ | KANO3 |
| R6' | 21,25,29 | BCAO2 + RO$_2$ → 0.7 BCAO + 0.3 BCAOH | KARO2 |

[a] Symbol ' is used to distinguish from the corresponding number of lumping reactions in Table 4.

reaction of BCARY only leads to the production of BCAO2. The MCM reactions No. 17 to 29 can then be reduced to reactions R2' to R6' via replacing.

The replacing strategy (Table 5) is expected to reduce more computational time than the lumping one (Table 4), since all reactions originating from the replaced species are removed from the mechanism. Hence, it does not require the computation
of weighting ratios and new surrogates. However, as a compromise, replacing could be less accurate than lumping, because replacing may discard some compounds and part of the mechanism, and therefore, lead to more error.

Thus, in efforts to prioritize the accuracy of reduction, GENOA currently employs replacing only after lumping and exclusively on species from the same reaction. In this way, species that were not lumped (because the lumping was rejected or because they do not respect the lumping restriction) can be reduced by replacing. During the training of BCARY reduction, a
260 restriction is applied on small organic compounds with a molar mass less than 100 g mol$^{-1}$, which are excluded from replacing.

Overall, the searches for viable reductions are conducted in reverse order of the reaction/species list. For removing, GENOA attempts to remove reactions from the bottom of the list and moves to the previous reactions. The same reverse sequence is followed for other strategies. When applied to the jumping strategy, for instance, GENOA tries to jump the species that has the highest generation and then move down to the species that has the lowest generation. Among all reduction strategies, only
265 lumping alters the saturation vapor pressure of condensable species. Therefore, a rank of saturation vapor pressure is used exclusively in lumping to determine the most appropriate lumpable species. At each reduction, GENOA attempts to reduce only one species/reaction via removing, or one pair of compounds via lumping/ replacing/ jumping. This restriction allows exhaustive tracking of every detailed modification and its effect on SOA concentrations.

### 2.3 Datasets of atmospheric conditions applied to reduction

All the atmospheric conditions applied to the reduction are extracted from a 3-D simulation spanning the latitudes from 32 °N to 79 °N and the longitudes from 17 °W to 39.8 °E over continental Europe in a one-year period (2015) using the chemistry-transport model CHIMERE. The CHIMERE model and the configuration used for the simulation are described in Lanzafame et al. (2022). The 3-D CHIMERE simulation was conducted with the implicit gas-phase MELCHIOR2 mechanism (Derognat

et al., 2003), which contains 120 reactions and less than 80 lumped species. The MELCHIOR2 mechanism describes the
degradation of sesquiterpenes by three oxidant-initiated reactions (HUMULE reacts with OH, $O_3$, and $NO_3$, respectively),
where the species HUMULE represents the lumped class of all sesquiterpenes.

The monthly diurnal profiles of hourly meteorological data (e.g., temperature, RH), and hourly concentrations of oxidant,
radical, and other inorganic species were extracted from each location. That information is required in the 0-D simulations
with SSH-aerosol (see section 2.4) to reproduce SOA concentrations and compositions under near-realistic conditions. Since
the reduced SOA mechanism focuses only on SOA formation, the meteorological data and the concentrations of oxidants,
radicals, and inorganics are assumed to remain intact during the 0-D SOA simulation. The coordinates and time of each
condition are also provided to calculate the solar zenith angle. Because the reduction focuses on the impact on SOA variation,
and because no inorganic reactions are considered in the reduced chemical mechanism, the oxidant, radical and inorganic
concentrations, as well as the environmental parameters, are fixed to the diurnal profiles obtained from the CHIMERE data in
0-D SOA simulations. The concentration of HUMULE (denoted $C_{SQT}$ as the CHIMERE surrogate for sesquiterpene) is used to
estimate the SQT concentration. For the purpose of calculating reduction parameters (e.g., weighting ratio $f_w$, branching ratio
$B$) and evaluating the reduced mechanisms, a dataset of representative physio-chemical conditions extracted from CHIMERE
simulation results is employed in GENOA. Depending on their usage, three groups of conditions are defined: the training
dataset, the pre-testing dataset, and the testing dataset.

### 2.3.1 Training dataset

The training dataset is the set of conditions used to initialize the reduction parameters, estimate the reference concentrations,
as well as to evaluate the mechanism at each potential reduction. For a mechanism containing over 1000 reactions and 500
species, a complete reduction may require more than 10 000 SOA simulations to evaluate all the reduction attempts. To reduce
the number of simulations and the computational cost, a limited number of conditions can be evaluated at each reduction
attempt.

For the reduction of BCARY degradation, a training dataset of eight conditions is selected, which contains six chemistry-
relevant conditions and two additional meteorological conditions. The geographic and meteorological information of each
condition is described in Table 6, where the conditions cover a broad range in time (summer and winter conditions), tempera-
tures ranging from 260 K to 302 K, and RH from 39 % to 89 %.

The six chemistry-relevant conditions, which are named after the dominant oxidants (OH, $O_3$ and $NO_3$), focus on the
influences of chemical regimes on SOA formation under either a high $NO_X$ regime (represented by high NO concentrations)
or a low $NO_X$ regime (represented by high $HO_2$ concentrations). The two additional conditions included in the training dataset
to improve the reduction, are referred to as ADD1 and ADD2.

The chemical regimes of the different conditions can be illustrated by seven competitive reacting ratios (equations are listed
in Appendix Table C1):

**Table 6.** Geographic and meteorological conditions of the training dataset

| Condition Name [a] | Lat °N | Lon °E | Time month | TEMP K | RH % | $R_{RO_2-NO}$ [b] % | SOA [c] $\mu g\ m^{-3}$ |
|---|---|---|---|---|---|---|---|
| OH NO | 36.0 | 15.4 | Jul. | 299 | 79 | 60 | 4.1 |
| OH HO$_2$ | 32.0 | -9.4 | Jul. | 296 | 77 | 20 | 6.1 |
| NO$_3$ NO | 40.25 | -3.4 | Jul. | 302 | 28 | 69 | 4.4 |
| NO$_3$ HO$_2$ | 32.0 | 36.6 | Aug. | 302 | 38 | 29 | 5.7 |
| O$_3$ NO | 69.0 | 33.8 | Jan. | 261 | 84 | 99 | 5.2 |
| O$_3$ HO$_2$ | 68.0 | 18.2 | Dec. | 266 | 89 | 25 | 4.6 |
| ADD1 | 41.5 | -14.2 | Dec. | 289 | 76 | 20 | 5.5 |
| ADD2 | 45.75 | 9.0 | Dec. | 279 | 85 | 100. | 4.4 |

[a] from left to right: name, latitude, longitude, time period, average temperature, average RH, average daily NO reacting ratio, simulated total SOA concentration of the training conditions.

[b] the average daily NO reacting ratio is calculated out of the RO$_2$ reactivity of NO, HO$_2$, NO$_3$, and RO$_2$. Conditions with a high $R_{NO}$ ratio are considered in the high NO$_x$ regime.

[c] simulated with an initial BCARY concentration of 5 $\mu g\ m^{-3}$.

– The reacting ratios of the precursor with the oxidants O$_3$ ($R_{O_3}$), OH ($R_{OH}$), and NO$_3$ ($R_{NO_3}$), whose sum equals 1, indicate the relative reactivity of the first-generation oxidation pathways that lead to the formation of distinct kinds of RO$_2$ species.

– The reacting ratios of RO$_2$ species with NO ($R_{RO_2-NO}$), HO$_2$ ($R_{RO_2-HO_2}$), NO$_3$ ($R_{RO_2-NO_3}$), and other RO$_2$ species ($R_{RO_2-RO_2}$), whose sum equals 1, indicate the relative reactivity of successive reactions with RO$_2$ species.

These ratios indicate the competition between autoxidation and bimolecular reactions that result in different SOA types. A combination of these seven reacting ratios determines the chemical regime and favorable reaction pathways under a given atmospheric condition.

Fig. 2 describes the reacting ratios at midnight (0 h) and noon (12 h) for the training conditions. Under the majority of atmospheric conditions, O$_3$ is the dominant oxidant of BCARY due to the carbon-carbon double bonds that are subject to ozonolysis. The high O$_3$ training conditions have $R_{O_3}$ exceeding 98 % at both noon and midnight. The bimolecular reactions with NO and HO$_2$ dominate RO$_2$ reactions in the MCM mechanism. Due to the low kinetic rate constants and low concentrations, the ratios of OH and NO$_3$ reacting with BCARY are relatively low (under 40 %). The high OH conditions are determined by the OH ratio at noon, while the high NO$_3$ conditions are determined by $R_{RO_2-NO_3}$ at midnight. One specific exception is the additional condition ADD2, which is located in the northern part of Italy, within the Alpine arch, close to the metropolitan city of Milan. This condition is in the extremely high-NO$_x$ regime, as high concentrations of NO are transported from polluted areas. These high NO concentrations consume O$_3$ and NO$_3$, causing low concentrations of O$_3$ and NO$_3$. At night, ADD2 has

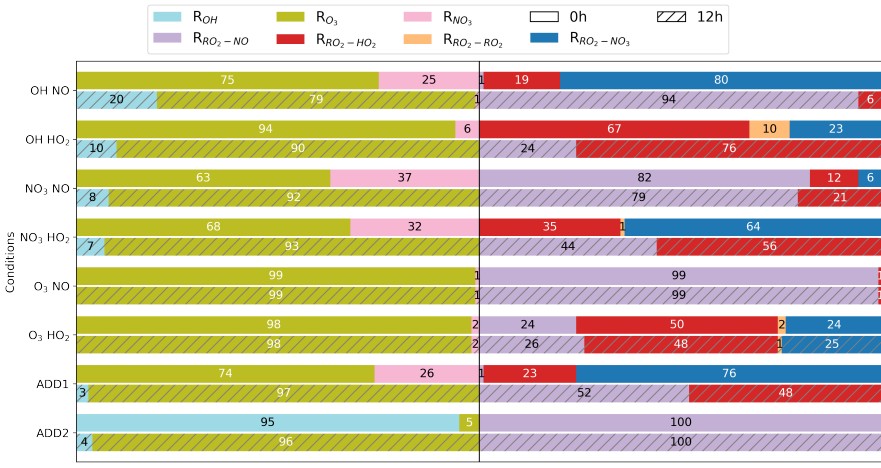

**Figure 2.** A bar plot showing the occupancy of seven reacting ratios in BCARY initiation reactions and $RO_2$ reactions, under the training conditions at midnight (0 h, top bar) and noon (12 h, bottom bar with slash). From left to right, six ratios are presented on each bar in the following order: $R_{O_3}$, $R_{OH}$, $R_{NO_3}$, $R_{RO_2-NO}$, $R_{RO_2-HO_2}$, $R_{RO_2-NO_3}$, and $R_{RO_2-RO_2}$ (No display if ratio is zero).

a high $R_{OH}$ of 95 % at midnight is not due to an abundance of OH, but rather to extremely low concentrations of $O_3$ ($2.9 \times 10^{-4}$ ppb) and $NO_3$ ($1.1 \times 10^{-9}$ ppb) that leads to an absence of nighttime reactivity.

### 2.3.2 Pre-testing dataset

The pre-testing dataset contains a greater number of conditions than the training dataset, providing a more accurate estimation of the reduction mechanism on SOA formation. After the mechanism has been significantly reduced, the pre-testing dataset is included along with the training dataset in order to evaluate the reduction attempts at the late-stage reduction. At this point of reduction, a slight change in the mechanism significantly impacts the SOA concentrations; therefore, merely evaluating reduction based on the training dataset may not be adequate. Meanwhile, the size of the mechanism has already been significantly reduced, which makes the evaluation of each reduction attempt on the pre-testing dataset less computationally expensive.

In principle, the pre-testing dataset should be able to provide a fairly accurate representation of the testing dataset. However, this may not always be the case, since the pre-testing dataset is selected almost randomly from the testing dataset. Therefore, an adjustment may be required to increase the representativeness of the pre-testing dataset by adding or removing a few conditions. For the application to BCARY, a pre-testing dataset with 150 atmospheric conditions is selected from the testing dataset, among which 50 conditions for each level (low, medium, and high) of SQT emissions. The locations of the training and pre-testing conditions are presented in Fig. 3.

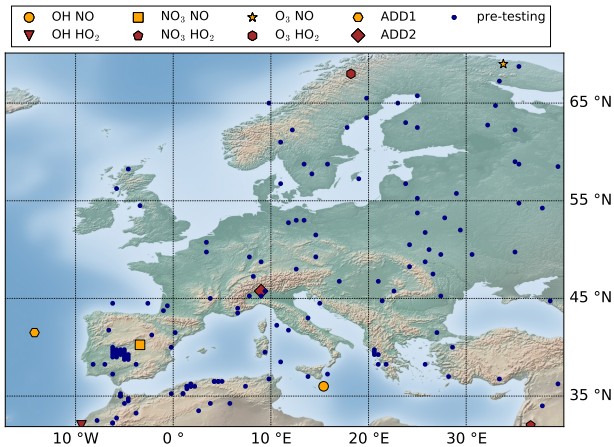

**Figure 3.** Simulation domain and locations of training (see the figure legend) and pre-testing (black scattered dots) datasets applied to the reduction.

### 2.3.3 Testing dataset

The final reduced mechanism, obtained from training, is eventually evaluated with a large number of atmospheric conditions in the testing section. This set of conditions for the final evaluation is referred to as the testing dataset. Among all datasets, the results on the testing dataset are most likely to reflect the performance of the reduced mechanism for 3-D modeling.

In the BCARY reduction, the testing dataset is selected based on the concentrations of the CHIMERE sesquiterpene surrogate. Its maximum hourly concentration $C_{SQT}$ in ppb is used to exclude conditions with negligible SQT concentration. A testing dataset within a total of 12 159 conditions is applied (see Sect. 3.2), including all conditions (2 159 conditions) with high SQT concentration ($C_{SQT} \geq 0.1$ ppb), 5 000 random-select ones with medium SQT concentration ($C_{SQT}$ $\epsilon$ between 0.01 and 0.1 ppb) and 5 000 random-select ones with low SQT concentration ($C_{SQT}$ $\epsilon$ (0.001, 0.01]). The conditions with extremely low SQT concentration ($C_{SQT} < 0.001$ ppb) are not included in the testing dataset. Fig. B1 indicates the locations of the testing dataset as well as the testing results for BCARY reduction.

### 2.4 Settings for SOA simulations

The chemical composition and time variation of SOA due to gas-phase chemistry and condensation/evaporation are simulated using the 0-D aerosol module SSH-aerosol (Sartelet et al., 2020). As detailed in Couvidat and Sartelet (2015), the gas/particle partitioning is estimated with Raoult's law (for the partitioning between the gas phase and the organic phase) and Henry's law (for the partitioning between the gas phase and the aqueous phase). Therefore, some properties of condensable compounds, such as the saturation vapor pressure $P_{sat}$ and the decomposition in functional groups, are crucial for modeling. For BCARY derived organics, $P_{sat}$ is calculated using UManSysProp (Topping et al., 2016). The vapor pressure is computed using the

method of Nannoolal et al. (2008) and the boiling point estimation from Joback and Reid (1987). This functional group method was selected since it provides the best performance when compared with the chamber experiment data of Chen et al. (2012); Tasoglou and Pandis (2015), as discussed in Appendix A. Furthermore, the activity coefficient $\gamma$ is calculated with the UNIQUAC Functional-group Activity Coefficients (UNIFAC) thermodynamic model (Fredenslund et al., 1975) for short-range interactions and the Aerosol Inorganic–Organic Mixtures Functional groups Activity Coefficients (AIOMFAC) model for medium-range and long-range interactions (Zuend et al., 2008).

Unless stated otherwise, two simulations are performed for each condition starting at midnight (0 h) and noon (12 h), taking into account both the daytime and nighttime chemistry. All 0-D simulations are run for five days in order to consider SOA formation and aging processes adequately. The initial BCARY concentration is set to 1 $\mu$g m$^{-3}$ in order to ensure high SOA production the (SOA concentration is always greater than 1 $\mu$g m$^{-3}$ at all evaluated conditions at all conditions). For optimal computational efficiency, the gas-particle partitioning is assumed to be at thermodynamic equilibrium.

## 2.5 Settings for evaluation

For the different datasets, the performance of the reduced mechanism on SOA concentrations is evaluated using the fractional mean error (FME) computed with Eq. 1, where $C_{val,i}$ and $C_{ref,i}$ denote the SOA mass concentration at time step i simulated with the reduced and the reference mechanisms, respectively.

The error of one simulation is defined as the larger of the FME on day one and the FME on days two to five, in order to address the difference in the performance of the reduced mechanisms at the early stage of the simulations (SOA formation dominates) and at the later stage (SOA aging dominates). This error is used to evaluate reduction by comparing it to the error tolerance specified in training. For the evaluation on the training dataset, two errors are estimated compared to the previously verified reduced mechanism with a tolerance denoted $\epsilon_{pre}$, and the MCM mechanism with a tolerance denoted $\epsilon_{ref}$. The error tolerances are used to restrict both the maximum and the average (half of the tolerance) errors of the training conditions. As for the evaluation on the pre-testing dataset, only the error compared to the MCM mechanism is calculated. The error tolerances $\epsilon_{pre-testing}^{ave}$ and $\epsilon_{pre-testing}^{max}$ are set to the average and maximum errors, respectively.

$$FME = \frac{2\sum_n^{i=1} abs(C_{val,i} - C_{ref,i})}{n\sum_n^{i=1}(C_{val,i} + C_{ref,i})}. \tag{1}$$

In order to begin with a conservative BCARY reduction, the initial values of $\epsilon_{pre}$ and $\epsilon_{ref}$ are both set to 1 %. The values of these error tolerances are then increased to larger values, reflecting the looser criteria used throughout the reduction. $\epsilon_{ref}$ is used to track the performance of the reduction, while $\epsilon_{pre}$ is used to avoid large errors introduced by one reduction attempt. Therefore, $\epsilon_{pre}$ is lower or equal than $\epsilon_{ref}$. For every 1 % increase in $\epsilon_{ref}$, $\epsilon_{pre}$ is stepped up by 1 % from 1 % to the value of $\epsilon_{ref}$. By doing this, GENOA first accepts reductions that introduce small errors compared to the previously validated mechanism, and then accepts reductions that introduce larger errors up to $\epsilon_{ref}$.

The maximum values for both $\epsilon_{ref}$ and $\epsilon_{pre}$ are set to 10 %. When $\epsilon_{ref}$ reaches 3 %, the mechanism is expected to be largely reduced. From then, the evaluation under the pre-testing dataset is considered to be added to the reduction. This means

that all subsequent reductions are evaluated using both the training and pre-testing datasets. The average and maximum errors ($\epsilon^{ave}_{pre-testing}$ and $\epsilon^{max}_{pre-testing}$) are restricted to be lower than 3 % and 20 %, respectively. As a result of the above error tolerances, a reduced SQT-SOA mechanism with an average inaccuracy on SOA formation lower than 3 % (maximum 20 %) is expected.

Additionally, another error factor noted as the fractional bias (FB, computed as detailed in Eq. 2) is used to visualize the temporal performance of the reduced mechanism at each simulation time step. As examples, Fig. 8 and Fig. 10 show the average FB at each time step for the pre-testing conditions.

$$FB_i = 2 \frac{C_{val,i} - C_{ref,i}}{C_{val,i} + C_{ref,i}} \tag{2}$$

When trying to remove reactions, GENOA first removes reactions with low hourly branching ratios ($B_{rm} \leq 5$ %), since the removing reactions with $B_{rm}$ is likely to have a minimal effect on SOA formation. After no reduction is accepted by all applied reduction strategies under the defined error tolerance, the value of $B_{rm}$ is increased to 10 % and then 50 %.

## 2.6 Settings for aerosol-oriented treatments

In late-stage training, an intense competition between different potential reductions is observed, and a minor modification may induce significant uncertainty in the mechanism and prevent further reduction. Besides, because the formation of aerosols costs more CPU time than gas-phase chemistry, specific treatments are employed in the late stage of training to reduce the number of condensable species preferentially. These treatments, which reduce species rather than reactions, are done when the size of the mechanism is below a certain threshold (20 for BCARY reduction). Consequently, late-stage treatments encourage the reduction via removing condensable species and are referred to as aerosol-oriented treatments. The treatments consist in:

- Restricting the reduction of the number of reactions. Thus, strategies that reduce the number of aerosols are favored to result in fewer condensable species.

- Bypassing the evaluation of aerosol-oriented reductions on the training dataset when applied to lumping, replacing, and jumping. As a result, the aerosol-oriented reduction is evaluated only on the pre-testing dataset to avoid being rejected under some of the extreme conditions in the training dataset (which are less representative of average atmospheric conditions than the conditions of the pre-testing dataset).

- Applying an additional type of removing: removing elementary-like reactions.

The additional reduction strategy of removing elementary-like reactions is targeted to reaction with multiple products. After rewriting the reaction into a set of elementary-like reactions, each with one oxidation product and integer stoichiometric coefficient, GENOA investigates the possibility of removing the elementary-like reactions one by one. In practice, removing elementary-like reactions is inserted after the strategy of removing reactions and before jumping, when no further reduction that reduces condensable species can be found with the current parameters.

# 3 Application to $\beta$-caryophyllene mechanism

GENOA is applied to the SQT degradation mechanism of the Master Chemical Mechanism v3.3.1 (Jenkin et al., 2012). Here $\beta$-caryophyllene (BCARY) is considered a surrogate for SQT primary VOCs. The degradation of $\beta$-caryophyllene in the original MCM mechanism consists of 1 626 reactions and 579 species (223 radicals and 356 stable species). After pre-reduction, the mechanism contains 1 241 reactions and 493 species (137 radicals and 356 stable species), which is employed as the starting point and the reference for the reduction (hereafter referred to as MCM).

Moreover, at the beginning of the GENOA training, all the stable species are assumed to be condensable (referred to as condensables), and their saturation vapor pressures and activity coefficients are calculated based on their molecular structures (as detailed in Sect. 2). Applying the effective partitioning coefficients ($K_p$ at 298 K) described by Seinfeld and Pandis (2016), condensables can be classified into: semi-volatile organic compound (SVOC, $K_p$ between $10^{-2}$ and $10^1$ m$^3$ $\mu$g$^{-1}$), low volatile organic compound (LVOC, $K_p$ between $10^1$ and $10^4$ m$^3$ $\mu$g$^{-1}$), and extremely low volatile organic compound (ELVOC, $K_p$ larger than $10^4$ m$^3$ $\mu$g$^{-1}$).

The semi-explicit SQT-SOA mechanism "Rdc." presented in this section is trained from MCM with GENOA. Detailed descriptions of the building process and its chemical scheme are provided in Sect. 3.1. By the end of the training, "Rdc." is reduced from MCM to only 23 reactions and 15 species (reaction/species lists see Appendix B). The size of the "Rdc." mechanism is of the same order of magnitude as the BCARY degradation scheme of Khan et al. (2017) (28 reactions and 15 species) used for global modeling. As presented in Sect. 3.2, the "Rdc." mechanism accurately reproduces the SOA concentration and composition simulated by MCM with only six condensables. Table B3 summarizes the new surrogates and the lumped MCM species that are included in the final "Rdc" mechanism.

## 3.1 Building of the reduced SOA mechanism

As shown in Fig. 4, the "Rdc." mechanism is built from 113 validated reduction steps. In GENOA, a reduction step refers to all reduction attempts based on the performed reduction strategy and reduction parameters, while a validated reduction step indicates at least one reduction attempt has been accepted at this step. The entire building process can be divided into three stages:

- Early stage, from the first to the 74 [th] reduction step. By the end of the 74 [th] reduction step, the mechanism is reduced to 68 reactions and 41 species (including 20 condensables). The early-stage reduction is trained only on the training dataset with the seven pre-described reduction strategies. After $\epsilon_{ref}$ reaching 3 %, the list of $B_{rm}$ is changed from [0.05, 0.10, 0.50] to [0.10, 0.50, 1.0].

- Late stage I, from the 75 [th] to the 107 [th] reduction step. By the end of the 107 [th] reduction step, the reduced mechanism consists of 38 reactions and 19 species (including seven condensables), and no further reduction can be found within $\epsilon_{ref} \leq 10$ % and $\epsilon_{pre} \leq 10$ %. In this stage, the reduction is trained on the pre-testing dataset if condensables are removed with lumping, replacing, or jumping. For reduction with other types of reduction strategies, it is first trained on the

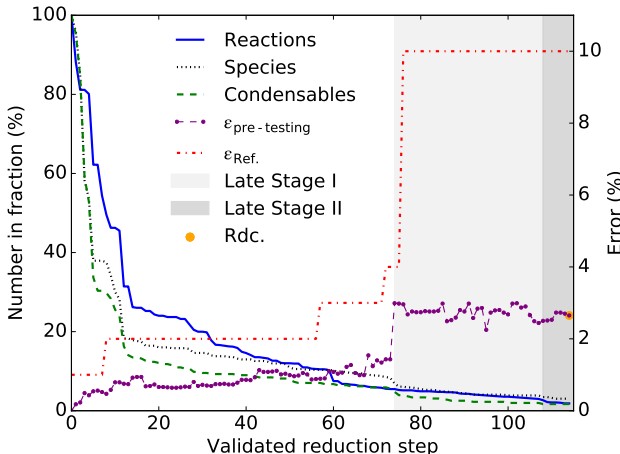

**Figure 4.** Reduction process of the "Rdc." mechanism showing the decrease of the number of reactions/species/condensables, and the evolution of the average error on the pre-testing dataset ($\epsilon_{pre-testing}$, with an error tolerance $\epsilon_{pre-testing}^{ave}$ of 3 %), along with the error tolerance compared to MCM ($\epsilon_{ref}$).

training dataset and then on the pre-testing datasets. From all reduced mechanisms with seven condensables, GENOA selected the one with the minimum average errors on the pre-testing dataset (2.44 %) to start the next stage.

    – Late stage II, from the $108^{th}$ to the $113^{rd}$ reduction step. At this stage, the reduction strategy of removing elementary-like reactions is applied to the training. All reductions that reduce the condensables are evaluated exclusively on the pre-testing dataset. The size of the reduced mechanism was reduced to 23 reactions and 15 species, among which the
number of condensables is reduced to 6. The average (maximum) error of the final reduced mechanism "Rdc." is 2.65 % (17.00 %) under the pre-testing dataset compared to MCM.

    The extent of the reduction due to each strategy is summarized in Table 7. Compared to MCM, up to 99 % of reactions and 97 % of species are reduced in "Rdc.". As expected, the reduction strategy of removing reactions contributes the most to the decrease in the number of reactions (48 %), followed by the strategy of removing species with a contribution of 37 %.
Meanwhile, both lumping and removing species are significant in the reduction of species, by 35 % and 31 %, respectively. The number of condensables decreases in proportion to the number of species, except for the strategy of removing partitioning. In that case, the gas-particle partitioning is removed and the species remains in the gas phase with no changes in the chemical mechanism.

    As shown in Fig.5, which describes the chemical scheme of the "Rdc." mechanism, the three oxidants (i.e., $O_3$, OH, and
465 $NO_3$) initiate reactions, leading to common oxidation products (e.g., mBCSOZ, mBCALO2) that dominate the successive oxidations. The different reaction pathways under high or low $NO_X$ regimes are presented in "Rdc." with reactions with NO or $HO_2$, respectively, which results in the formation of different types of SOA: mBCKSOZ, mC133O, and C131PAN (in the presence of $NO_2$) under high $NO_X$ conditions, and mC132OOH under low $NO_X$ conditions. Other pathways, for example,

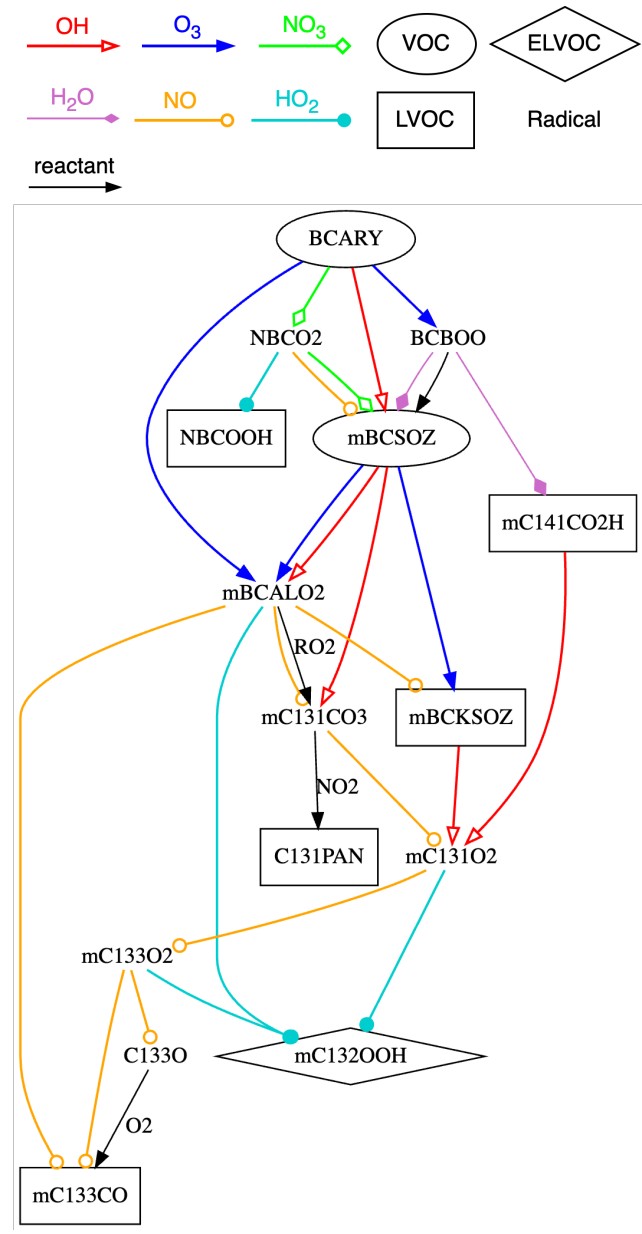

**Figure 5.** Representation of the chemical scheme of the "Rdc." mechanism. VOC, LVOC, and ELVOC are presented in ellipse, square, and diamond boxes, respectively. Radicals are written in plain text, without boxes. Reactions with OH, $O_3$, $NO_3$, NO, $HO_2$, and $H_2O$ are shown by edges with different colors and arrowheads (see the figure legend). Other reactants (if any) are labeled near the edges. The complete species and reaction lists of the "Rdc." mechanism are in Appendix Table B1 and B2, respectively.

**Table 7.** Reduction accomplished per each reduction strategy during the building process of the "Rdc." mechanism.

| Strategy | No. (fraction[a] %) | | |
|---|---|---|---|
| | reaction[b] | species | condensables |
| Removing reactions | 594(48) | 38(8) | 26(7) |
| Removing elementary-like reactions[c] | 8(1) | 0(0) | 0(0) |
| Lumping | 0(0) | 171(35) | 110(31) |
| Replacing | 25(2) | 39(8) | 31(9) |
| Jumping | 138(11) | 79(16) | 43(12) |
| Removing species | 453(37) | 151(31) | 108(30) |
| Removing partitioning | 0(0) | 0(0) | 32(9) |
| Removed in total | 1218(98) | 478(97) | 350(98) |

[a] fraction of the original number (of reactions or species) that is reduced by the strategy.

[b] columns from left to right are the number (and fraction) of reduced chemical reactions, reduced total gas-phase species, and reduced gas-phase species that can condense on the particle phase, compared to MCM with 1 241 reactions and 493 species (356 condensables).

[c] only applied in the reduction at late stage II.

the bimolecular reactions of the Criegee intermediate BCBOO with water vapor, and the $RO_2$ reaction of mBCALO2, are also preserved in the "Rdc." mechanism. The six condensables in "Rdc." can be categorized into one SVOC, four LVOCs, and one ELVOC, according to the effective partitioning coefficient calculated on the pre-testing dataset. The SOA concentration per volatility class is discussed in Sect. 3.2.

Compared to MCM, "Rdc." simplifies a considerable number of reactions that have small impacts on SOA formation (e.g., photolysis reactions) under the majority of atmospheric conditions, and merges a large number of compounds with similar chemical properties. The main oxidation products from the first two generation oxidation of MCM are preserved mainly through the "Rdc." species mBCSOZ, which is a lumped surrogate of several MCM representative BCARY derived oxidation products: BSCOZ (the major secondary ozonize with a molar yield $\geq$ 65 % reported by Jenkin et al. (2012)), BCAL (the primary product formed from both OH and $O_3$-initiated chemistry), and BCKET (from OH-initiated reactions).

## 3.2 Evaluation of the reduced SOA mechanism

### 3.2.1 Reproduction of the SOA concentrations

During the testing procedure, the "Rdc." mechanism is evaluated at 12 159 locations, with two different starting times (0 h and 12 h). The testing for "Rdc." took approximately 2% of the CPU time consumed for MCM.

Compared to MCM, "Rdc." presents a high level of accuracy with an average error of 2.66 % and a maximum error of 17.29 %. The monthly distribution of the number of the testing conditions as well as the testing errors are described in Fig. 6.

The error is lower than 10 % for more than 99 % of the simulations. The summer conditions, between June and September, covering more than half of the testing conditions (63 %, 15 294 conditions), result in an average error of 2.37 % and a $3^{rd}$ quartile error of 2.85 %. As compared to summer conditions, testing results in winter conditions from October to January (19 % of the testing dataset, 4 570 conditions) display slightly higher uncertainty, with an average error of 3.79 % and a $3^{rd}$ quartile error of 5.36 %.

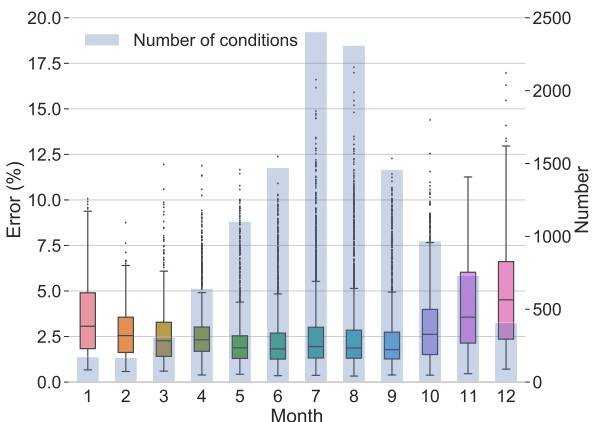

**Figure 6.** Monthly distribution of the testing results (errors compared to MCM) of the "Rdc." mechanism in the box plot, and the number of testing conditions in the histogram.

An error map of testing conditions in July and August is displayed in Fig. 7. It indicates the locations of testing conditions and the errors of each condition, especially highlighting outliers during this period. Detailed error maps of all testing conditions can be found in Appendix B. It shows that the "Rdc." mechanism induces low errors, lower than 6 %, for most of the testing conditions. The conditions with errors over 6 % are mainly concentrated in northern Africa near the Atlas Mountains and in the Eastern Mediterranean, where the conditions most likely correspond to a dry Mediterranean climate with low RH and high

temperature. Other conditions with errors above 6 % are dispersed in the Pô valley of Northern Italy and along the coasts of southern Spain. More accurate results could be obtained with stricter parameters for reduction (e.g., lower error tolerance), or by updating the conditions (e.g., training and pre-testing datasets) covering more extreme conditions in the training process.

### 3.2.2  Reproduction of the SOA composition

The SOA concentrations and chemical composition simulated with the "Rdc." mechanism and with MCM are compared in this

section. The temporal profiles of the total SOA concentrations on an average of the pre-testing dataset and non-ideal conditions are displayed in Fig. 8. Throughout the entire five-day simulation period, there is excellent agreement between hourly SOA concentrations simulated with MCM and those obtained from the "Rdc." mechanism. The SOA concentration builds up rapidly

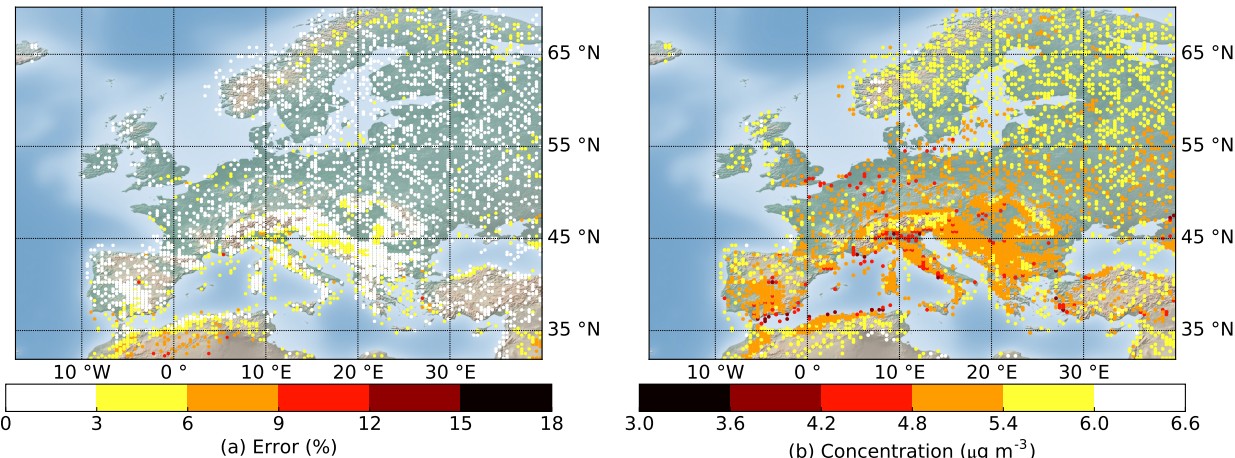

(a) Error (%)

(b) Concentration (µg m⁻³)

**Figure 7.** Geographic distributions of (a) error and (b) average SOA concentration of the testing results in July and August simulated using the "Rdc." mechanism. The total number of conditions displayed is 4 717 out of 12 159 that were tested. The results of all testing conditions are shown in Appendix B for reference.

in the first few hours, where the results of the "Rdc." mechanism present relatively larger fluctuations (The maximum FB of 3.74 % is observed at 1 h on the average pre-testing results).

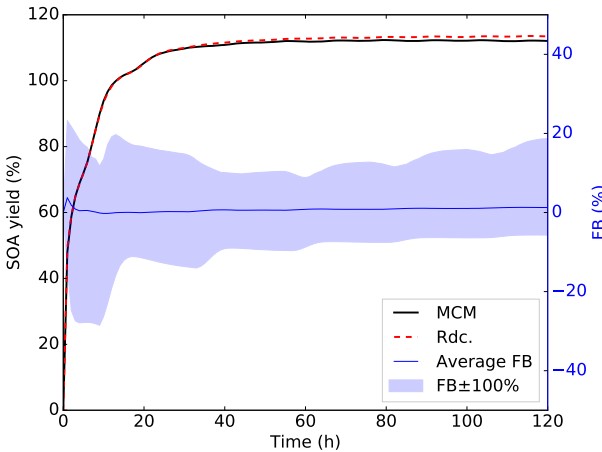

**Figure 8.** Temporal variation of total SOA concentration simulated with the pre-testing dataset and MCM (red, dotted), and the "Rdc." mechanism (black, plain) mechanisms under the non-ideal conditions. The average (blue, plain) and maximum FB (blue, shadow) between MCM and the "Rdc." mechanism is also presented.

The average SOA concentrations per volatility class on the pre-testing dataset at two simulation times (8 h and 72 h) are listed in Table 8. At both 8 h and 72 h, the "Rdc." mechanism accurately reproduces the total SOA mass with a relative difference

lower than 0.1 % compared to MCM. An accumulation of the SOA mass into the ELVOC class is observed (51 % of the total SOA mass at 8 h and 66 % at 72 h), with both the MCM and the "Rdc" mechanisms. The aging of SOA produces compounds of low and extremely low volatility. Regarding the volatility classes, the "Rdc." mechanism tends to slightly overestimate the SOA resulting from ELVOCs and underestimate the SOA resulting from LVOCs, especially at 72 h. This suggests that aging leads to "Rdc." condensables of slightly lower volatility than the MCM ones. However, the differences are low, up to 0.4 $\mu$g m$^{-3}$ difference (10 %) at 72 h.

**Table 8.** Average SOA concentrations per volatility class simulated with MCM and the "Rdc." mechanisms on the pre-testing dataset at 8 h and 72 h.[a]

| Conditions | SVOC | LVOC | ELVOC | Total |
|---|---|---|---|---|
| MCM at 8 h | 0.18 | 1.91 | 2.17 | 4.26 |
| "Rdc." at 8 h | 0.13 | 1.80 | 2.31 | 4.24 |
| MCM at 72 h | 0.02 | 1.90 | 3.69 | 5.61 |
| "Rdc." at 72 h | 0.02 | 1.51 | 4.12 | 5.64 |

[a] unit in $\mu$g m$^{-3}$.

The average SOA composition per functional group simulated on the pre-testing dataset at 72 h is displayed in Fig.9. No significant change in the functional group distributions is found between 8 h and 72 h of oxidation. The alkyl (RC) and carbonyl groups (RCO) contribute the most to the SOA mass, by more than 1 $\mu$g m$^{-3}$, whereas the other functional groups contribute by less than 1 $\mu$g m$^{-3}$. Overall, the "Rdc." mechanism satisfactorily reproduces the composition of the MCM-simulated SOA composition for most functional groups, except for nitrogen-containing groups. In comparison to MCM, only two condensables containing nitrogen are retained in the "Rdc." mechanism: NBCOOH and C131PAN, leading to an underestimation of the organic nitrate group (0.31 in MCM and 0.04 in "Rdc.") and an overestimation of the nitrate mass of the peroxyacetyl nitrate group (0.10 $\mu$g m$^{-3}$ in MCM and 0.30 in "Rdc."). To obtain better results on the reproduction of nitrogen groups, GENOA may be further restricted to distinguish nitrogen compounds in training. Additionally, the peroxyacetyl acid group results in an extremely low SOA mass in MCM (less than 0.01 %), and therefore, is not kept in the "Rdc." mechanism.

Moreover, the temporal profiles of the OM/OC ratio, as well as the H/C, O/C, and N/C atomic ratios are presented in Fig. 10. Comparable patterns are observed in the OM/OC (1.65 in MCM and 1.63 in "Rdc." on average), the O/C (0.37 in MCM and 0.36 in "Rdc."), as well as the H/C ratios (1.62 in MCM and 1.60 in "Rdc."). During the first 8-hour simulation, "Rdc." tends to slightly overestimate the OM/OC and O/C ratios, while the H/C ratio remains fairly stable throughout the entire simulation with a negligible difference (0.02) between MCM and "Rdc.". The N/C ratio, however, is underestimated by the "Rdc." mechanism by 37 % on average (ratio equal to 0.019 in MCM and to 0.012 in "Rdc."), indicating the over-reducing organic nitrites in "Rdc.". A total of three nitrogen-containing organics (NBCO2, NBCOOH, and C131PAN) are preserved in "Rdc.", of which two (NBCO2, NBCOOH) are first-generation products. Therefore, during the first 10 hours, the N/C ratio curve simulated by "Rdc." drops, whereas in MCM it increases as higher-generation nitrates are produced.

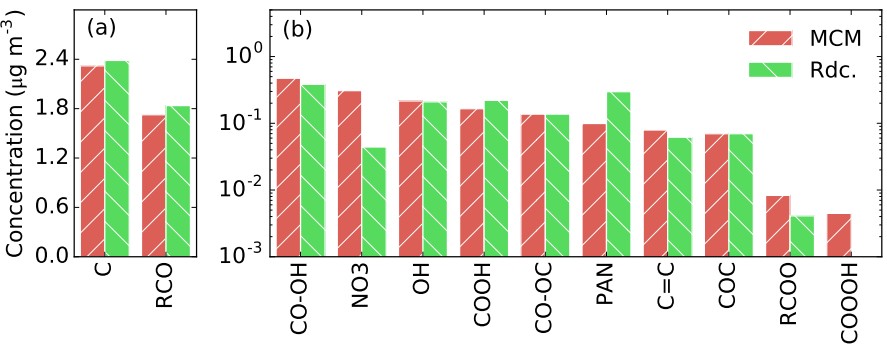

**Figure 9.** Average SOA mass per functional group simulated with the pre-testing dataset and MCM (red bar with slash), the "Rdc." mechanism (green bar with backslash) at 72 h. The figure is divided into two panels (a) and (b) due to the large gap in mass between the groups. The labels of the functional groups from left to right are RC carbon bond, RCO carbonyls (ketone and aldehyde), CO-OH hydroxy peroxide, NO3 organic nitrates, OH alcohol, COOH carbonyl acid, CO-OC peroxide, PAN peroxyacyl nitrates, C=C carbon double bond, COC ether, RCOO ester, COOOH peroxyacetyl acid.

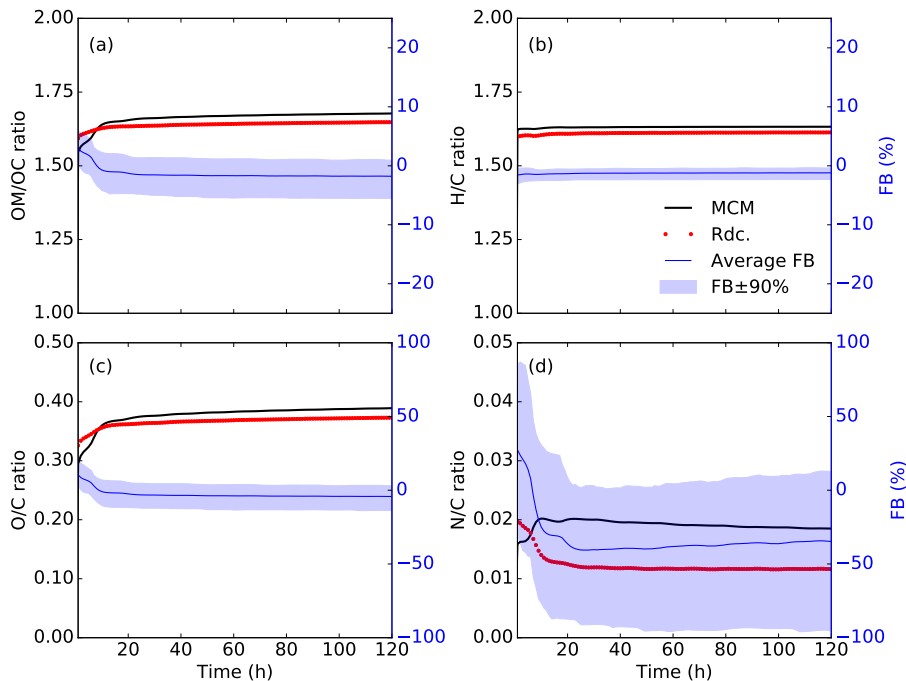

**Figure 10.** Time variations of (a) average organic mass to organic carbon mass (OM/OC) ratio, (b) hydrogen to carbon (H/C) atomic ratio, (C) oxygen to carbon (O/C) atomic ratio, and (d) nitrogen to carbon (N/C) atomic ratio, simulated with MCM (solid black curves) and the "Rdc." mechanism (dotted red curves) on the pre-testing dataset. The average FB (blue plain line) with the 90 % range of FB (blue shadow) is also presented.

### 3.2.3 Sensitivity on environmental parameters

The sensitivities of the "Rdc." mechanism to temperature, RH, and SOA mass conditions are investigated with the pre-testing dataset. The default value of BCARY concentration is 5 $\mu$g m$^{-3}$, and the default RH and temperature are set to constant 50 % and 298 K, respectively. As presented in Fig. 11, the SOA yields simulated by the "Rdc." mechanism with different environmental parameters show a remarkable resemblance with the SOA yields simulated by MCM.

Under 10 $\mu$g m$^{-3}$, the simulated SOA yields are not affected by the SOA mass loading. This result is consistent with the large contribution of ELVOC reported in Table 8. A discrepancy of 25 % in the average SOA yield at 1 h with an SOA mass loading of 10$^3$ $\mu$g m$^{-3}$ at 1 h and a discrepancy of 8 % at 72 h with an SOA mass loading of 10$^{-3}$ $\mu$g m$^{-3}$ are observed. The result indicates that the "Rdc." mechanism may introduce relatively large uncertainty with extreme SOA loading (larger than 500 $\mu$g m$^{-3}$), which was outside the range of conditions used for the construction of the "Rdc." mechanism. SOA formation is affected by RH, because of both the gas-phase chemistry (reaction with H$_2$O vapors) and the gas-particle transfer (condensation of hydrophilic SOA precursors on aqueous aerosols). The sensitivity tests show that the "Rdc." mechanism reproduces well (differences lower than 2 %) the SOA yields of MCM with RH ranging from 5 % to 95 %. For temperature, the "Rdc." mechanism reproduces very well the SOA aging at 72 h, but larger discrepancies are observed in the earlier period, when the oxidation products are more volatile. However, the discrepancies in SOA yield stay low: differences up to 7 % (at 1 h and 72 h) and 10 % (at 8 h) are observed for temperatures equal to 263 K and 323 K, respectively. This finding is consistent with the testing results. To sum up, the discrepancies suggest that the reduced mechanism performs quite well, although larger discrepancies with MCM are observed under conditions that are outside the range of conditions used during training.

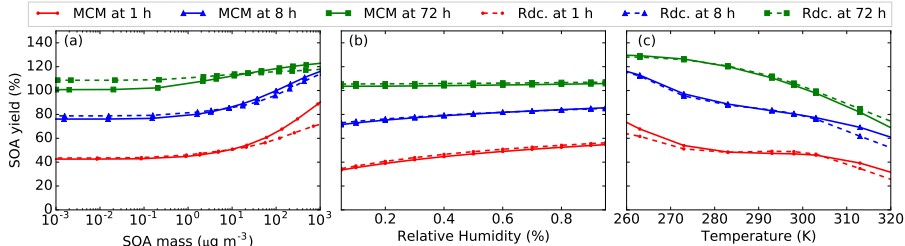

**Figure 11.** Dependence of the average SOA yield simulated by the pre-testing dataset with MCM (solid line) and the "Rdc." mechanism (dotted line) on (a) BCARY-SOA mass, (b) relative humidity (RH), and (c) temperature at 1 h (red point), 8 h (blue triangle) and 72 h (green square).

### 4 Conclusions

The development and application of the GENerator of Reduced Organic Aerosol Mechanisms (GENOA) have been presented in this study. GENOA generates semi-explicit SOA mechanisms designed for large-scale air quality modeling by reducing explicit VOC mechanisms with a series of automatic training and testing processes. During the training procedure of GENOA, four

types of reduction strategies (lumping, replacing, jumping, and removing) are adopted to locate the potential reduction in the
mechanism. Each reduction attempt is evaluated against the explicit mechanism under a sequence of near-realistic atmospheric
conditions (the training dataset, and/or the pre-testing dataset at the late stage of reduction). Finally, the reduced mechanism is
evaluated on various conditions of a testing dataset. Under each condition, two five-day 0-D simulations starting at midnight
and noon are conducted with the aerosol model SSH-aerosol to evaluate the performance of the reduced SOA mechanism.

GENOA successfully generated semi-explicit SOA chemical mechanisms for the degradation of sesquiterpene, for which the
explicit $\beta$-caryophellene mechanism of the Master Chemical Mechanism serves as the reference mechanism and the starting
point. The final reduced SQT-SOA mechanism contains 23 reactions (down from 1626 reactions in MCM), 15 gas-phase
species (down from 579 gases), and six aerosol species (down from 356 aerosols). It reproduces the SOA formation and aging
by introducing an average error of 2.7 % under conditions over Europe with only 2 % of the size of MCM. The SOA volatility
is well reproduced with the reduced mechanism, as well as the decomposition into functional groups, and the OM/OC (1.55 in
the "Rdc." mechanism and 1.60 in MCM), H/C, and O/C ratios. Nitrogen-containing SOA, which contributes to only 7 % of
the total mass, is not as well represented as other groups, and the ratio N/C is slightly underestimated in the "Rdc." mechanism
(0.016 against 0.021 in MCM). The similarity of the representation of the functional group decomposition allows reproducing
the non-ideality of SOA similarly in the "Rdc." mechanism and in MCM. Additionally, the sensitivity tests on RH, temperature,
and organic mass loading show that the SOA simulated with the "Rdc." mechanism is in good agreement with MCM results
under most conditions (except for conditions with extremely high temperature or with massive organic aerosol loading where
discrepancies in the SOA yields may reach 8 % (temperature) and 25 % (massive mass loading)). It indicates that the reduced
mechanism performs well for conditions in the training range, but the performance may decrease for conditions outside of
this range. To improve the performance of the semi-explicit SOA mechanism under conditions outside the training range, two
methods can be employed: the first is to include the outlier conditions in the training procedure if they are considered influential
to SOA formation, and the second is to adopt strict error tolerance to restrict the reduction.

*Code and data availability.* The source code for GENOA v1.0 is hosted on GitHub at https://github.com/tool-genoa/GENOA/tree/v1.0 (last
access: 25 April 2022). The associated Zenodo DOI is https://doi.org/10.5281/zenodo.6482978. The dataset we used to run the BCARY
MCM reduction is publicly available online on Zenodo: https://doi.org/10.5281/zenodo.6483088.

## Appendix A: The computation of saturation vapor pressure of BCARY-SVOCs

The ozonolysis experimental data reported in Tasoglou and Pandis (2015) and Chen et al. (2012) are used to evaluate the perfor-
mances of different computation methods for the saturation vapor pressure of BCARY oxidation products. In our simulations,
the saturation vapor pressure is computed by UManSysProp with the SMILES structures of organic compounds. Eight methods
are provided in UManSysProp, including SIMPOL.1 of Pankow and Asher (2008) ("sim"), EVAPORATION of Compernolle
et al. (2011) ("evp"), and six methods out of the combination of two methods to compute the vapor pressure ("v0": Myrdal and

585 Yalkowsky (1997) and "v1": Nannoolal et al. (2008)) and three methods to compute the boiling point ("b0": Nannoolal et al. (2004), "b1": Stein and Brown (1994), and "b2": Joback and Reid (1987)). As shown in Fig. A1, the SOA distribution simulated with "v1b2" agrees best with the experimental data. Therefore, this method with the vapor pressure computed by Nannoolal et al. (2008) and the boiling point computed by Joback and Reid (1987) is used in the BCARY reduction. The results simulated with the final reduced mechanism "Rdc." is also presented in Fig. A1, which has a great resemblance to the experimental data.

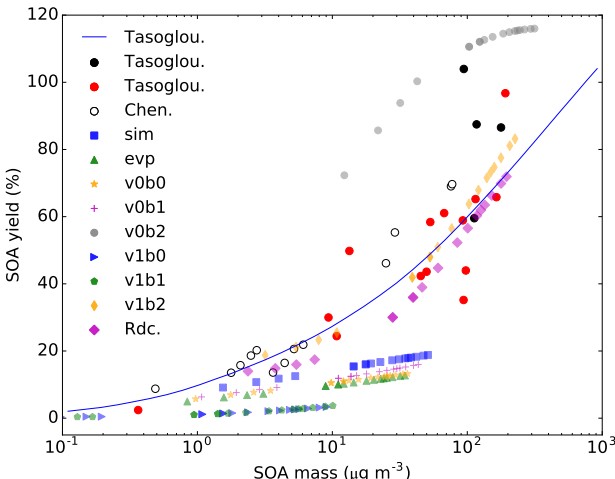

**Figure A1.** The SOA yields versus the total SOA mass from the experimental data reported by Chen et al. (2012); Tasoglou and Pandis (2015), simulated in SSH-aerosol with the MCM mechanism and different saturation vapor pressures methods, and simulated with the "Rdc." mechanism. The "Rdc." mechanism is trained from the MCM mechanism with the "v1b2" method.

 **Appendix B: An overview of the "Rdc." mechanism**

**Table B1.** Species list of the "Rdc." mechanism.

Notice that the species in the reduced case may be different from the MCM species with identical names.

| Surrogate[a] | Type[b] | Molecular formula | MW[c] | $P_{sat}^{d}$ | $\Delta H_{vap}^{e}$ | H[f] | $\gamma^{g}$ |
|---|---|---|---|---|---|---|---|
| BCARY | VOC | $C_{15}H_{24}$ | 204.4 | | | | |
| NBCO2 | Radical | $C_{15}H_{24}NO_5$ | 298.4 | | | | |
| BCBOO | Radical | $C_{15}H_{24}O_3$ | 252.3 | | | | |
| mBCALO2 | Radical | $C_{14.68}H_{24.08}O_{4.87}$ | 278.5 | | | | |
| mBCSOZ | VOC | $C_{15}H_{24}O_{2.74}$ | 248.2 | | | | |
| NBCOOH | LVOC | $C_{15}H_{25}NO_5$ | 299.4 | $4.04 \times 10^{-12}$ | $1.19 \times 10^{2}$ | $7.63 \times 10^{6}$ | $1.8 \times 10^{6}$ |
| mC141CO2H | LVOC | $C_{14.94}H_{23.85}O_{3.06}$ | 252.4 | $4.43 \times 10^{-11}$ | $1.14 \times 10^{2}$ | $2.53 \times 10^{4}$ | $4.95 \times 10^{7}$ |
| mBCKSOZ | SVOC | $C_{14}H_{22}O_{3.9}$ | 252.7 | $2.02 \times 10^{-8}$ | $9.12 \times 10^{1}$ | $1.89 \times 10^{5}$ | $1.45 \times 10^{4}$ |
| mC131CO3 | Radical | $C_{14.09}H_{21.28}O_{4.91}$ | 269.1 | | | | |
| mC131O2 | Radical | $C_{13.14}H_{21.22}O_{4.14}$ | 245.5 | | | | |
| mC133CO | ELVOC | $C_{13.42}H_{20.83}O_{4.59}$ | 255.6 | $1.62 \times 10^{-12}$ | $1.25 \times 10^{2}$ | $2.45 \times 10^{2}$ | $1.4 \times 10^{11}$ |
| mC132OOH | ELVOC | $C_{13.97}H_{23.92}O_{4.59}$ | 279.5 | $4.13 \times 10^{-14}$ | $1.36 \times 10^{2}$ | $5.70 \times 10^{3}$ | $2.36 \times 10^{11}$ |
| C131PAN | LVOC | $C_{14}H_{21}NO_7$ | 315.3 | $4.39 \times 10^{-11}$ | $1.13 \times 10^{2}$ | $2.07 \times 10^{4}$ | $6.09 \times 10^{7}$ |
| mC133O2 | Radical | $C_{13}H_{21}O_{5.97}$ | 272.8 | | | | |
| C133O | Radical | $C_{13}H_{21}O_5$ | 257.3 | | | | |

[a] species with "m" are the new surrogates that merged multiple MCM-BCARY species. [b] VOC: stable species without gas-particle partitioning; The volatility classes is defined in Sect. 3. [c] Molar weight ($gmol^{-1}$).

Only compute for SOA precursors: [d] saturation vapor pressure at 298 K (atm) ; [e] Enthalpy of vaporation ($KJmol^{-1}$); [f] Henry's law constant ($matm^{-1}$); [g] activity coefficient at infinite dilution in water.

**Table B2.** Reaction list of the "Rdc." mechanism.

| No. | Reactions | Kinetic Rate constant[a] |
|-----|-----------|--------------------------|
| 1 | BCARY + NO$_3$ -> NBCO2 + NO$_3$ | $1.9 \times 10^{-11}$ |
| 2 | BCARY + O$_3$ -> 0.874 BCBOO + 0.111 mBCALO2 + O$_3$ | $1.19 \times 10^{-14}$ |
| 3 | BCARY + OH -> mBCSOZ + OH | $1.97 \times 10^{-10}$ |
| 4 | NBCO2 + HO$_2$ -> NBCOOH + HO$_2$ | $2.837 \times 10^{-13} \times \exp(\frac{1300}{T})$ |
| 5 | NBCO2 + NO -> mBCSOZ + NO | $2.7 \times 10^{-12} \times \exp(\frac{360}{T})$ |
| 6 | NBCO2 + NO$_3$ -> mBCSOZ + NO$_3$ | $2.3 \times 10^{-12}$ |
| 7 | BCBOO -> 0.5 mC141CO2H + 0.5 mBCSOZ | $[H_2O] \times 4 \times 10^{-16}$ |
| 8 | BCBOO -> mBCSOZ | $2 \times 10^{2}$ |
| 9 | mBCSOZ + O$_3$ -> 0.915 mBCKSOZ + 0.085 mBCALO2 + O$_3$ | $1.1 \times 10^{-16}$ |
| 10 | mBCSOZ + OH -> 0.92 mBCALO2 + 0.08 mC131CO3 + OH | $7.6 \times 10^{-11}$ |
| 11 | mC141CO2H + OH -> mC131O2 + OH | $6.494 \times 10^{-11}$ |
| 12 | mBCALO2 + NO -> 0.505 mBCKSOZ + 0.353 mC131CO3 + 0.099 mC133CO + NO | $6.56 \times 10^{-12} \times \exp(\frac{360}{T})$ |
| 13 | mBCALO2 -> 0.6 mC131CO3 | $[RO_2] \times 1.711 \times 10^{-12}$ |
| 14 | mBCALO2 + HO$_2$ -> mC132OOH + HO$_2$ | $1.939 \times 10^{-13} \times \exp(\frac{1300}{T})$ |
| 15 | mBCKSOZ + OH -> mC131O2 + OH | $3.28 \times 10^{-11}$ |
| 16 | mC131CO3 + NO -> mC131O2 + NO | $6.377 \times 10^{-12} \times \exp(\frac{290}{T})$ |
| 17 | mC131CO3 + NO$_2$ -> C131PAN + NO$_2$ | $0.8502 \times KFPAN$ |
| 18 | mC131O2 + HO$_2$ -> mC132OOH + HO$_2$ | $2.288 \times 10^{-13} \times \exp(\frac{1300}{T})$ |
| 19 | mC131O2 + NO -> mC133O2 + NO | $2.213 \times 10^{-12} \times \exp(\frac{360}{T})$ |
| 20 | mC133O2 + HO$_2$ -> mC132OOH + HO$_2$ | $2.695 \times 10^{-13} \times \exp(\frac{1300}{T})$ |
| 21 | mC133O2 + NO -> 0.757 C133O + 0.243 mC133CO + NO | $2.61 \times 10^{-12} \times \exp(\frac{360}{T})$ |
| 22 | C133O -> mC133CO | $[O_2] \times 2.5 \times 10^{-14} \times \exp(\frac{-300}{T})$ |
| 23 | C133O -> | $2.7 \times 10^{14} \times \exp(\frac{-6643}{T})$ |

[a] [H$_2$O] concentration of H$_2$O, [RO$_2$] the total concentration of RO$_2$ species pool,[O$_2$] concentration of O$_2$. KFPAN is one of the complex rate coefficients from the MCM mechanism.

**Table B3.** The new surrogates in the "Rdc." mechanism and the corresponding lumped species in the original MCM mechanism. Noted that the "Rdc." surrogates may also go through other reductions (e.g., jumping) that does not affect their structure.

| Rdc. surrogate | lumped MCM species |
|---|---|
| mBCSOZ | BCSOZ, BCAL, BCKET |
| mC141CO2H | C141CO2H, C143CO, C1310CO, BCALCCO, C143OH, BCCOH, BCAOH |
| mBCALO2 | BCALO2, C146O2, C142O2, BCKAO2, C147O2 |
| mBCKSOZ | BCKSOZ, BCLKET, BCALOH, BCKBCO, BCKAOH, BCSOZOH |
| mC131CO3 | C131CO3, C141CO3, C1211CO3, C137CO3 |
| mC131O2 | C131O2, C144O2, C143O2, BCLKAO2, C152O2, BCLKCO2 |
| mC132OOH | C132OOH, BCSOZOOH, C133OOH, C146OOH, C147OOH |
| | C1313OOH, BCLKBOOH, BCLKAOOH, C152OOH, C145OOH |
| | C148OOH, C144OOH, BCALOOH, BCKBOOH, C151OOH |
| mC133O2 | C133O2, C1313O2 |
| mC133CO | C133CO, C131CO2H, C148CO, C145OH, C1313OH, BCLKBOH, BCLKAOH |
| | C152OH, C151OH, C147OH, BCLKACO, C148OH, C1211CO2H |

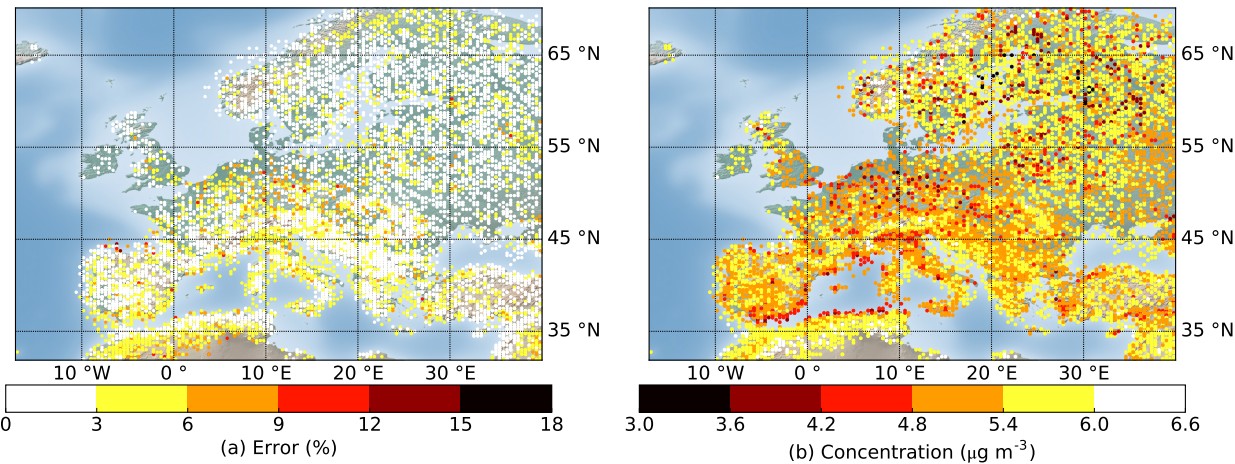

**Figure B1.** Maps of (a) error and (b) average SOA concentration of the testing results simulated using the "Rdc." mechanism on all (i.e., 12 159) testing conditions.

## Appendix C: Information related to the reduction

**Table C1.** The computation of estimating chemical activity ratios used to display training dataset in Fig. 2

| Name[a] | Reactant[b] | Computation[c] | Kinetic rate coefficient[d] |
|---|---|---|---|
| $r_{OH}$ | OH | $k_{OH}[OH] / (k_{OH}[OH] + k_{O_3}[O_3] + k_{NO_3}[NO_3])$ | $k_{OH} = 1.97 \times 10^{-10}$ |
| $r_{O_3}$ | $O_3$ | $k_{O_3}[O_3] / (k_{OH}[OH] + k_{O_3}[O_3] + k_{NO_3}[NO_3])$ | $k_{O_3} = 1.20 \times 10^{-14}$ |
| $R_{NO_3}$ | $NO_3$ | $k_{NO_3}[NO_3] / (k_{OH}[OH] + k_{O_3}[O_3] + k_{NO_3}[NO_3])$ | $k_{NO_3} = 1.90 \times 10^{-11}$ |
| $R_{RO_2-NO}$ | NO | $k_{NO}[NO]/(k_{NO}[NO] + k_{HO_2}[HO_2] + k_{RNO_3}[NO_3] + k_{RO_2}[RO_2])$ | $k_{NO} = 2.70 \times 10^{-12} \times \exp(\frac{360}{T})$ |
| $R_{RO_2-HO_2}$ | $HO_2$ | $k_{HO_2}[HO_2]/(k_{NO}[NO] + k_{HO_2}[HO_2] + k_{RNO_3}[NO_3] + k_{RO_2}[RO_2])$ | $k_{HO_2} = 2.91 \times 10^{-13} \times \exp(\frac{1300}{T})$ |
| $R_{RO_2-RO_2}$ | $RO_2$ | $k_{RO_2}[HO_2]/(k_{NO}[NO] + k_{HO_2}[HO_2] + k_{RNO_3}[NO_3] + k_{RO_2}[RO_2])$ | $k_{RO_2} = 9.20 \times 10^{-14}$ |
| $R_{RO_2-NO_3}$ | $NO_3 + RO_2$ | $k_{NO_3}[NO_3]/(k_{NO}[NO] + k_{HO_2}[HO_2] + k_{RNO_3}[NO_3] + k_{RO_2}[RO_2])$ | $k_{RNO_3} = 2.30 \times 10^{-12}$ |

[a] names of the reacting ratio of OH radical, $O_3$, and $NO_3$ radical reacted with BCARY. ($r_{OH} + r_{O_3} + R_{NO_3} = 1$); and of the reacting ratio of NO, $HO_2$ radical, $RO_2$ radical, and $NO_3$ radical (at the presence of $RO_2$) reacted with $RO_2$ species ($R_{RO_2-NO} + R_{RO_2-HO_2} + R_{RO_2-RO_2} + R_{RO_2-NO_3} = 1$).

[b] Reactions with those compounds are preferred when the corresponding reacting ratios are high.

[c] [species_name] (e.g.,[OH]) is the monthly average concentration of oxidants concentration extracted from CHIMERE.

[d] kinetic rate coefficient are provided by MCM, where $k_{OH}$, $k_{O_3}$, and $k_{NO_3}$ are the kinetic rate coefficient of first-generation BCARY reaction with OH, $O_3$, and $NO_3$, respectively; $k_{NO}$, $k_{HO_2}$, and $k_{RNO_3}$ are the simple rate coefficients KRO2NO, KRO2HO2, and KRO2NO3, respectively; $k_{RO_2}$ is self-reaction rate coefficients for the tertiary peroxy radicals (e.g., BCAO2, BCCO2). T: temperature (K).

## C1    Additional examples of lumping

Besides the example shown in Sect. 2.2.3, two additional examples have been added from the BCARY reduction: one illustrates the lumping of two similar compounds formed by different reactions, and the other illustrates the lumping of two more distinct compounds. The first example is the MCM species C1313NO3 and C152NO3 (see Fig. C1). These two species come from different reactions. The molecular structures of both compounds are similar (they contain organic nitrates, aldehydes, and alcohols), but C152NO3 contains an additional carboxylic acid where C1313NO3 contains an aldehyde. The corresponding reactions before and after lumping are summarized in Table C2, where the new surrogate "mC1313NO3" is built from C1313NO3 with a weighting ratio of 83% and C152NO3 with a weighting ratio of 17%. As a result of this lumping, the average error increase under training conditions is 0.001 % (the tolerance is 0.01 %).

Another example of lumping is the MCM species BCALBOC and C1310OH. Unlike the previous example, these two species are more distinct. According to MCM, BCALBOC are generated through $O_3$-initiated reactions, while C1310OH are generated through high-generation oxidations. There is less similarity in the structures or chemical reactions of the two molecules. MCM contains the OH reaction of BCALBOC, and the $O_3$ and OH reactions of C1310OH. However, this reduction was accepted since lumping them only increased the average error by 0.01 % under training conditions (the tolerance was 1 %). The new

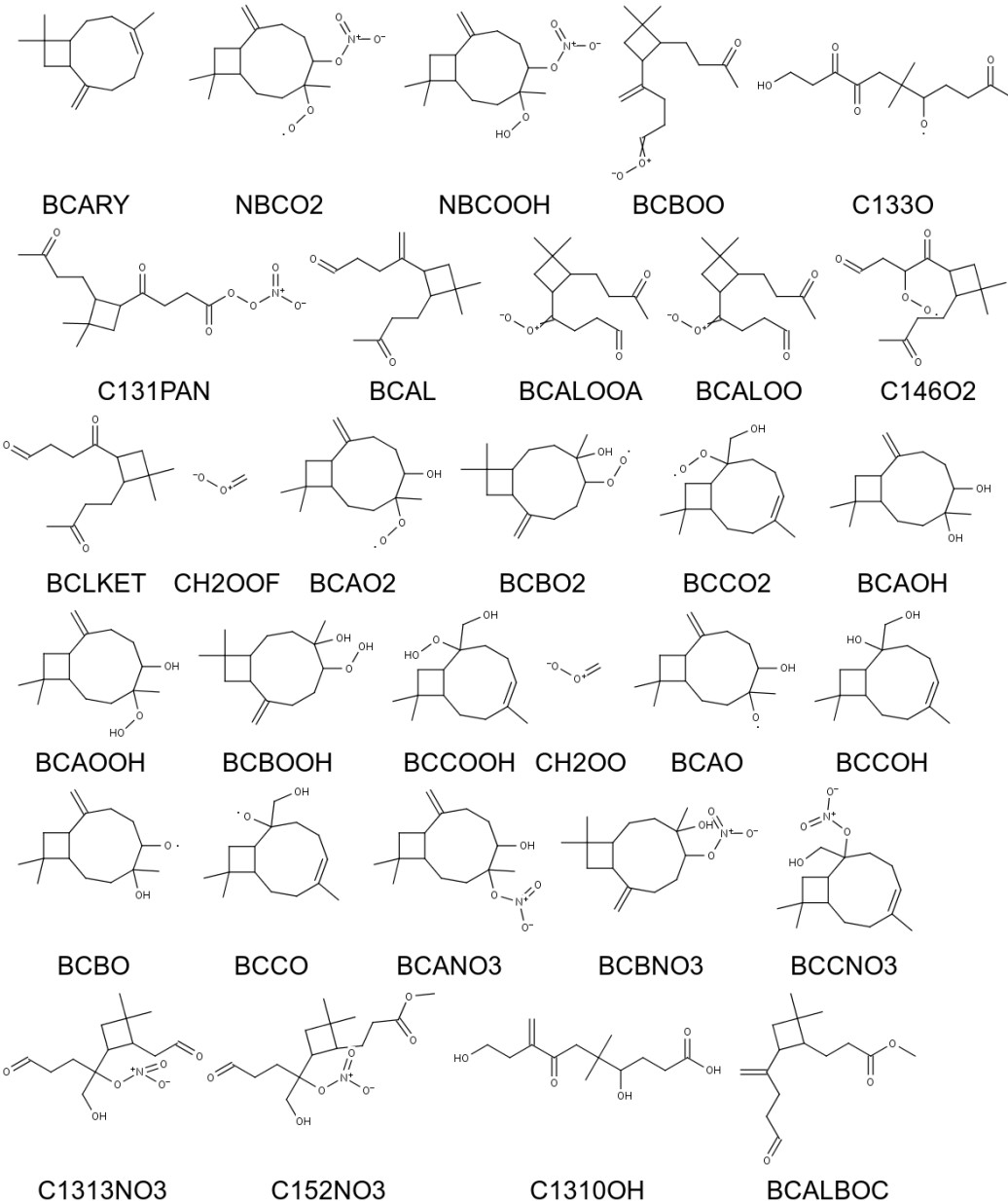

**Figure C1.** Molecular structures of the MCM species that are mentioned in the paper. For more information, please visit the MCM website.

**Table C2.** Reactions related to the reduction of MCM species C1313NO3 and C152NO3 via lumping.

| Reactions before lumping | Kinetic coefficient [a] |
|---|---|
| Production | |
| C1313O2 + NO → C1313NO3 | KRO2NO×0.134 |
| C152O2 + NO → C152NO3 | KRO2NO×0.136 |
| Destruction | |
| C1313NO3 + OH → C116CHO + HCHO + NO$_2$ | $5.59{\times}10^{-11}$ |
| C152NO3 + OH → BCLKBOC + HCHO + NO$_2$ | $1.58{\times}10^{-11}$ |
| Reactions after lumping | |
| Production | |
| C1313O2 + NO → 0.134 mC1313NO3 + 0.866 C1313O + 0.866 NO$_2$ | KRO2NO |
| C152O2 + NO → 0.136 mC1313NO3 + 0.864 C152O + 0.864 NO$_2$ | KRO2NO |
| Destruction | |
| mC1313NO3 + OH → C116CHO + HCHO + NO$_2$ | $5.59{\times}10^{-11}$ × 0.82945 |
| mC1313NO3 + OH → BCLKBOC + HCHO + NO$_2$ | $5.59{\times}10^{-11}$ × 0.17055 |

[a] 0.82945 is the exact weighting ratio of C1313NO3 and 0.17055 is the exact weighting ratio of C152NO3.

surrogate "mBCALBOC" is constructed from BCALBOC with a weighting ratio of 98 % and C1310OH with a weighting ratio of 2 %.

As C1310OH has a low weighting ratio, the lumping would be substituted by replacing (a special case of lumping), where the weighting ratio of BCALBOC is set to 100 % and of C1310OH is set to 0 %. In that case, instead of forming a new surrogate, 610 C1310OH is replaced by BCALBOC. In BCARY reduction, this type of replacing was not used, but it can be activated by the user by setting the weighting ratio threshold.

*Author contributions.* ZW developed the model code and performed the simulations. ZW, FC, KS designed the research, developed the methodology. ZW wrote the manuscript with contributions from FC and KS. FC and KS were responsible for funding acquisition.

*Competing interests.* The authors declare that no competing interests are present.

*Acknowledgements.* This work was financially supported by INERIS and DIM QI$^2$ (Air Quality Research Network on air quality in the Île-de-France region). The authors would like to thank Youngseob Kim from CEREA for his help in using the SSH-aerosol model.

**Table C3.** Reactions related to the reduction of MCM species BCALBOC and C1310OH via lumping.

| Reactions before lumping | Kinetic coefficient |
|---|---|
| Production | |
| BCOOA → BCALBOC | $1.0 \times 10^6 \times 0.15$ |
| C1310O2 → C1310OH | $2.5 \times 10^{-13} \times [RO_2] \times 0.2$ |
| Destruction | |
| BCALBOC + O$_3$ → BCBOOA + HCHO | $1.1 \times 10^{-16} \times 0.670$ |
| BCALBOC + O$_3$ → BCLKBOC + CH2OOF | $1.1 \times 10^{-16} \times 0.330$ |
| BCALBOC + OH → C152O2 | $6.98 \times 10^{-11}$ |
| C1310OH + OH → C1310CO + HO$_2$ | $6.2 \times 10^{-11}$ |
| Reactions after lumping | |
| Production | |
| BCOOA → mBCALBOC | $1.0 \times 10^6 \times 0.15$ |
| C1310O2 → mBCALBOC | $2.5 \times 10^{-13} \times [RO_2] \times 0.2$ |
| Destruction | |
| mBCALBOC + O$_3$ → BCBOOA + HCHO | $1.10 \times 10^{-16} \times 0.670 \times 0.97675$ |
| mBCALBOC + O$_3$ → BCLKBOC + CH2OOF | $1.10 \times 10^{-16} \times 0.330 \times 0.97675$ |
| mBCALBOC + OH → C1310CO + HO$_2$ | $6.20 \times 10^{-11} \times 0.97675$ |
| mBCALBOC + OH → C152O2 | $6.98 \times 10^{-11} \times 0.023251$ |

[a] 0.97675 is the exact weighting ratio of BCALBOC and 0.023251 is the exact weighting ratio of C1310OH.

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
