# Peer review of "GENOA User's Manual"

_EGUsphere, 2022_

## Referee Comment (RC2)

The paper discusses the development of a semi-explicit reduced organic aerosol mechanism for sesquiterpenes (Bcary), with the aim for employing it in air quality and large scale models. GENOA mechanism is based on the widely used near-explicit Master chemical mechanism (MCM). The mechanism used different strategies namely, lumping, replacing, jumping an removing to reduce the MCM scheme. The reduction procedure is tested under various environmental conditions (RH, temp etc.), resulting in a final reduced mechanism (Rdc.) suitable to simulate SOA. The simulated SOA using Rdc has low average error when compared to the near-explicit MCM scheme. This is a well thought out work, with suitable implications to better reproduce SOA in large scale and air-quality models. I would therefore, recommend the publication of this work after the authors have answered the following questions:

**General**

The main question is why did the authors chose sesquiterpenes? Why not isoprene or monoterpenes? The motivation to use sesquiterpenes should be highlighted.

Since GENOA is a semi-explicit mechanism, can it be used with any box or air quality model?

Although comparison has been made against MCM, the performance of a model can be made by comparing it against exisiting experimental SOA yields. There has been quite a lot of published experimental Bcary SOA yield experiments. I would suggest the authors to discuss GENOA derived SOA yields in comparison to these expeiments.

Is GENOA a carbon number conserving mechanism. Its is not clear from the manuscript if the mechanism is carbon conserving or not? If it is not then how do the authors justify it?'

**Specific questions**

L13-15: Motivation → Although the health and climate effects of aerosols are introduced in every paper, the authors should maybe consider to explain these in a few words or a sentence. Also, it would be nice to explain why there is a need to improve the SOA representation in AQMs.

L18-22: This sentence seems to contain quite a lot of information. I would suggest the authors to rephrase it into smaller sentences.

L24: "box models". Although, it is true that explicit schemes are used in box models due to relaxed computational burdens, they have been also been used in 1-D column models or 2-D Lagrangian models (these are still not as computationally expensive compared to AQMs or Global climate models).

L32: "carbon-bond" instead of "carbon-bound".

L34-35: Are all the above mechanisms (lumped, CB05, MCM, GECKO-A) developed primarily for ozone simulation? Also it would be good to give examples of a few model species.

L55: "suitable to"

L 112-113: Why is this order used in the reduction strategy for BCARY? How would any other order influence the reduction strategy?

Table1: Typo in reaction 8. I think it is supposed to say 0.2BCBOH instead of 0.2BCAOH.

Table2: Typo in lumped reaction R2: $0.753 * (f_{w,a} BCANO3 + f_{w,b} BCBNO3 + f_{w,c} BCCNO3)$. I would guess the factor is 0.247 instead of 0.753.

Table2: $C_{r,b}$ and $C_{r,c}$ are not defined. It should be defined similar to $C_{r,a}$ for better clarity.

L119: It would be much clear to write it as "In this example, a total of 12 chemical reactions involving three organic compounds are reduced to **five** reactions (4 lumped ($R_{1-4}$) + 1 surrogate (R0))".

L122-123: How are the BCARY isomers undergoing similar reactions with $HO_2$,NO and $NO_3$? Are the authors referring to the $R_{1-4}$ in the lumped scheme. Please make this clear to the readers.

Why is $C_{r,a,b,c}$ an arithmetic mean of 5 day simulations? I.e was this 5 day period selected?

L129: kinetic → kinetics

L130: weighting → weighing

L135: What is this specific behavior?

L162: Aren't alkoxy radicals are $RO^{.}$

L179: Can the authors explain what the maximum hourly branching ratio is?

L185: Do the authors mean that that after one loop of reduction (as shown in Figure 1), the subsequent reduction is carried out in the reverse order? And why is saturation vapor pressure used only for the lumping strategy. This part needs more clarity.

Table 5: Are the two conditions ADD1 and ADD2 high or low $NO_x$ regimes?

Figure 2: How do I interpret this? Is the top bar representing 0H and the bottom one representing 12 h? Please add this to the figure caption, since 0 h and 12 h are not represented by empty and hashed lines.

L233: I wonder what are the conditions which cause such low $O_3$ and $NO_3$ concentrations?

L258: Is Kelvin effect not taken into account for gas-to-particle partitioning?

L262: "vapor pressure is computed using Mydral and Talkowsky". This phrase is repeated already in the earlier part of the sentence.

Appendix A: It should be "v1:Nannolal (2008)" not Nannolal 2004.

Appendix A, L485: There seems to be a contradiction here. The authors say that v1b2 (Nannolal and Jacob and reid) show the best estimate in comparison with the experiments *"As shown in Fig. A1, the SOA distribution simulated with "v1b2" agrees best with the experimental data. Therefore, this method with the vapor pressure computed by Myrdal and Yalkowsky (1997) and the boiling point computed by Joback and Reid (1987) is used in the BCARY reduction"*. But in the manuscript why has Mydral and Talkowsky been selected in place of Nannolal 2008. Why?

L275: I think it would be better to rephrase the sentence defining the FME. Is it so that the simulation error is the larger FME of the two errors I.e the FME of day 1 and the FME of rest of the simulation days? This has be more clear in the text.

Eq 1: What does i1 and i2 represent?

L283: Why does the $\varepsilon_{pre}$ vary in such a way with respect to $\varepsilon_{ref}$. More explanation is need here.

L288: Is this true? As mentioned the <3% avg error is for pre-testing + training dataset. Will the average error still be < 3% for test dataset?

L303: How does GENOA decide what condensable species to remove? Is it based on species super saturation values?

L326: Effective partitioning coeff. is temperature dependent. What is the assumed temperature for the classification of SVOCs, LVOCs and ELVOCs? And since the authors use a range of temperatures in their training conditions (268-302 K) does it really make any sense in classifying the SVOCs, LVOCs and ELVOCs at an arbitrary temperature?

L332: Are the species in the reduced Rdc and Khan 2017 mechanism overlapping or identical?

L341: Condensable species drop to less than 20 ? It seems that the condensable species is ~10 % of its original values (10% of 493 ~48/49 species or thereabout) as shown in Figure 3. Also $\varepsilon_{pre}$ is not shown in Figure 3. Also on L 346 it is mentioned as 41 species on the 75$^{th}$ time. So what is the correct number?

L346-348: Aren't all the reduction strategies trained first with training data sent and then with pre-testing data set? So what is the difference here between, lumping, jumping and replacing compared to other strategies?

L351: "evacuated" → "evaluated"

L358: It should be specified that lumping reduces the condensable species by 35 %.

L366: There is not mC133O in Figure 4.

L392: Why is more uncertainty found in regions with low RH and high temperatures? Is it because the training data set does not have enough data to work with in these conditions? Looking at figure 6, it shows that even Russian data points have high uncertainty between 3-6%. This cannot be only due to low RH and high temperature. What could be the other possible reasons for that?

Figure 8: Didn't the authors claim that PAN is under-represented in GENOA on L 418? But Figure 8 shows higher PAN concentrations for GENOA and lower for MCM.

L438: It should be explained why due to different volatility species Rdc delays SOA production. Is it due to low LVOC concentrations or high ELVOC concentrationss?

---

## Author Comment (AC1)

**Authors' Response to Reviews of**

**GENerator of reduced Organic Aerosol mechanism (GENOA v1.0): An automatic generation tool of semi-explicit mechanisms**

Zhizhao Wang, Florian Couvidat, Karine Sartelet
*Geoscientific Model Development,* `https://doi.org/10.5194/egusphere-2022-245`
* * *
**RC:** *Reviewers' Comment*,    AR: Authors' Response,    ☐ Manuscript Text

**Reviewer # 3**

**RC:** *This paper presented a reduction strategy and developed a software package to reduce the mechanism of SOA formation. The paper is reasonably well written. The method described is innovative and effective. However, there are some information misleading and inappropriate. The current manuscripts needs to be modified before it is accepted for publication.*

**AR:** We would like to thank reviewer # 3 for the positive comments and constructive suggestions, which helped us to substantially improve our manuscript. We have carefully considered all of these comments and revised the manuscript accordingly. Please note that the line and section numbers mentioned in the response correspond to the version of the manuscript before revision.

**1. General comments**

**RC:** *The authors chose sesquiterpenes as an example to show the methods to reduce SOA formation mechanism.*

**AR:** Sesquiterpene was selected because it is a well-known source of SOA, and its formation mechanism is well documented. Sesquiterpene is also an ideal candidate for model development and demonstration of the reduction methodology, as the oxidation products of sesquiterpene are less volatile and tend to condense more readily than those of lighter molecules such as monoterpene and isoprene.

We have added the reason why we selected sesquiterpene to the main paper:

> *Section 1, line 72:*
>
> The application of GENOA to the MCM degradation scheme of $\beta$-caryophyllene (BCARY) (Jenkin et al., 2012) is described in Sect. 3.  $\beta$-caryophyllene  is selected for investigation and demonstration of the GENOA algorithm, because it is one of the most abundant and representative sesquiterpene (SQT)  Sesquiterpenes are well-known source of SOA (Hellén et al., 2020; Tasoglou and Pandis, 2015 and the formation mechanism of BCARY is well documented in the

near-explicit MCM mechanism (Jenkin et al., 2012). Studies have also compared BCARY SOA yields simulated using the MCM mechanism to chamber data (e.g., Xavier et al., 2019). BCARY is therefore an ideal candidate for model development and demonstration of the reduction methodology. In this paper, the near-explicit MCM BCARY degradation scheme serve as a reliable benchmark for GENOA. The experiment data from Tasoglou and Pandis (2015); Chen et al. (2012) are also compared to the newly developed reduced mechanism in Appendix A.

**RC:** *One obvious question is that most of the reactions listed are linear and therefore analytical solutions can be achieved. In such case, the reduction may be of less significance.*

The formation of SOA from VOC degradation involves many processes, including gas-phase multi-generation oxidation and gas-to-particle mass transfer. These processes are highly non-linear. For example, the gas-to-particle mass transfer depends on the concentration and composition of both the gas and the particle phases. As the particle composition depends on the environmental conditions and on the formation of other organic compounds (i.e., non-ideality), it is not possible to define an analytical solution. The process of SOA formation from sesquiterpene is, therefore, complex and non-linear.

**RC:** *Moreover, the authors did not show the impact of reduction on the concentration of short-lifetime oxidants, such as OH, HO2 and NO3, which have an important influence on SOA yields. How are the coupling between VOC species and oxidants resolved? Reasonable explanations should be highlighted.*

**AR:** The reduced SOA mechanism focuses solely on reproducing the SOA concentration of those simulated with explicit mechanisms, with fewer species and reactions. Thus, the oxidant concentrations are simulated using implicit gas-phase chemical mechanisms. Current reduced mechanisms do not affect the concentrations of $O_3$ and major radicals (OH, $NO_3$, NO, $HO_2$) produced by the implicit gas-phase mechanism.

In air quality modeling, implicit SOA mechanisms are often based on the surrogate approach (e.g., Odum's two-product) or the Volatility Basis Set. They are added to implicit gas-phase mechanisms without altering the pathways of oxidants. The same approach is adopted for our semi-explicit SOA mechanisms. Our reduced SOA mechanisms, which include gas-phase chemical mechanisms and aerosol properties for condensable gas-phase species derived from certain SOA precursors, can be added to the implicit gas-phase mechanism in the 3-D model. For example, the 3-D CHIMERE model can simulate SOA using the implicit gas-phase mechanism MELCHIOR2 and SOA mechanism $H^2O$ [Couvidat et al., 2018]. With our reduced SOA mechanisms, SOAs in CHIMERE are simulated by MELCHIOR2, the reduced SOA mechanisms generated by GENOA, and $H^2O$ for other SOA precursors not covered by our mechanisms.

For the paper, We have added the explanation of how the coupling is generally resolved between VOC species and oxidants in 3-D models:

*Section 1, line 36:*

 To complete implicit gas-phase mechanisms, implicit SOA mechanisms have been developed to model specifically the SOA formation without modifying the concentrations of ozone and major radicals (Kim et al., 2011) . In 3-D modeling, implicit SOA mechanisms or parameterizations are usually added to implicit gas-phase mechanisms, conserving the oxidant chemistry of the implicit gas-phase mechanism.

Implicit SOA mechanisms are often established based on experimental data from smog chamber experiments to represent the formation and evolution of SOA, such as the two-product empirical SOA

model (Odum et al., 1996) and the volatility basis set (VBS) that splits VOC oxidation products into a uniform set of volatility "bins" Donahue et al., 2006).

We have also added the following statements about our semi-explicit mechanisms:

> *Section 2, line 80:*
>
> The generated semi-explicit mechanisms are designed to preserve the accuracy of explicit mechanisms for SOA formation, while keeping the number of reactions/species low enough to be suitable for large-scale modeling, particularly  in 3-D AQMs. The focus of the semi-explicit mechanism is solely on the accurate modelling of SOA. Because ozone, major radicals, and other inorganics are also affected by inorganic and other VOC chemistry, their concentrations are not tracked with the semi-explicit mechanism. Instead, they are simulated using existing implicit gas-phase chemical mechanisms.

Additionally, details about implicit gas-phase mechanisms of the 3-D CHIMERE model and how 3-D CHIMERE results are used for 0-D simulations have been added to the paper:

> *Section 2.2, line 193:*
>
> The version of CHIMERE and its configuration is described in Lanzafame et al. (2022). The  3-D CHIMERE simulations were conducted with the implicit gas-phase MELCHIOR2 mechanism (Derognat et al., 2003), which contains 120 reactions and less than 80 lumped species. The MELCHIOR2 mechanism describes the degradation of sesquiterpenes by three oxidant initiated reactions (HUMULE reacts with OH, O3, and NO3, respectively), where the the species HUMULE represents the lumped class of all sesquiterpenes.
>
> The monthly diurnal profiles of hourly meteorological data (e.g., temperature, relative humidity), and hourly concentrations of  oxidant, radical, and other inorganic species were extracted from each location. That information is required in the 0-D simulations with SSH-aerosol (see section 2.4) to reproduce SOA concentrations and compositions under near-realistic conditions. Since the reduced SOA mechanism focuses only on SOA formation, the meteorological data and the concentrations of oxidants, radicals and inorganics are assumed to remain intact during the 0-D SOA simulation. The coordinates and time of each condition are also provided to calculate the solar zenith angle. Because the reduction focuses on the impact on SOA variation, and because no inorganic reactions are considered in the reduced chemical mechanism, the oxidant, radical and inorganic concentrations , as well as the environmental parameters, are fixed to the diurnal profiles obtained from the CHIMERE data in 0-D SOA simulations. The concentration of HUMULE (denoted $C_{SQT}$ as the CHIMERE surrogate for sesquiterpene  is used to estimate the SQT concentration.

Moreover, the GENOA methodology would preserve both SOA and oxidant concentrations in the reduced SOA mechanism if relevant criteria were included in the training process. This may be a potential direction for future development of the GENOA algorithm.

**RC:** *Since GENOA is a semi-explicit mechanism and designed to be used in 3-D models, how can the species in GENOA be matched with the species in 3-D models. One problem that arises is whether GENOA is scale adaptive so that the transport of organic species is well resolved depending on the problem of concern. If not, the package may be useful in optimizing mechanisms for specific urban scenarios and then used in regulatory modeling rather than stated "multi-scale AQM" by the authors.*

AR:  As we mentioned in the previous comment, semi-explicit SOA mechanisms are added to the implicit mechanisms of the 3-D model. The additional species from semi-explicit SOA mechanisms are then transported by the 3-D model. In the case of sesquiterpene-SOA, only 14 gas-phase species within six condensable species need to be added to the 3-D model, which is a computational scale that is fully acceptable for transport in 3-D modelling. When applied to nested simulation domains (for example over an European domain, a domain over a country and over a city), the user need to use the same semi-explicit SOA mechanisms across all domains, so that the species are the same for transport. In that case, the SOA mechanism has to be trained under all conditions that might be encountered in each domain. As such, GENOA is not "scale adaptive", since the size of the mechanisms does not change with the domain size. Therefore, transport of those additional VOC species is not a problem. To avoid confusion, we have removed the word "multi-scale" from the paper.

RC:  *It would be more useful if the manuscript can include some information on memory optimization and computational efficiency.*

AR:  Thanks for the suggestion. We have included the reduced CPU time for the testing process in the paper:

> *Section 3.2.1, line 380:*
>
> During the testing procedure, the "Rdc." mechanism is evaluated at 12 159 locations, with two different starting times (0 h and 12 h). The 0-D testing for "Rdc." took approximately 2% of the CPU time consumed by MCM.

In addition, we have tested the memory usage of the MCM and RDC mechanisms with the box model SSH-aerosol. As shown in Fig.1, the peak memory consumption is reduced by 96% (from 11.8 MiB to 577.1 KiB) after the reduction. It should be noted that the result of 0-D testing may not be linear to the computational efficiency of 3-D modeling since 3-D simulations include other processes (e.g., transport, deposition). However, the results confirm that the semi-explicit SOA mechanism is conducive to large-scale modeling.

Generally speaking, in 3-D modelling, the most time-consuming process is solving the gas-particle partitioning in SOA modelling. The number of condensable species has a significant impact on CPU performance. As for memory consumption, it varies with the number of species transported especially in the condensed phase, as those are often considered for different particle sizes. Therefore, decreasing the number of compounds that may partition to the condensed phase is a priority for semi-explicit mechanisms. For sesquiterpene, the number of condensables increases only by six, which is acceptable for most large-scale simulations.

**2. Specific comments**

RC:  *L26: three-dimensional models -> three-dimensional (3-D) models*

AR:  Revised.

RC:  *L27: 3D -> 3-D*

AR:  Changed.

RC:  *L32: carbon-bound -> carbon-bond*

AR:  Changed.

[Figure]

Figure 1: Memory maps for MCM (top panel) and "Rdc." (bottom panel) mechanisms simulated with SSH-aerosol under the same conditions. The graphs are generated with massif-visualizer.

**RC:** *L34: The expression "These mechanisms were primarily developed for ozone simulation" may be NOT appropriate.*

**AR:** Sorry for the ambiguity. This phrase has been revised as follows:

> *Section 1, line 34:*
>
>  Implicit gas-phase mechanisms were developed and validated to simulate the concentrations of oxidants and other conventional air pollutants such as ozone and $NO_2$. In these mechanisms, VOCs have been grouped into a limited number of model species because of computational considerations, and the SOA formation is usually not considered.

**RC:** *Table 1: As the authors stated "using surrogates assigned to molecular structures", listing the information of molecular structure of major species in another table would be better for other researchers to understand the reduction strategy.*

**AR:** Thanks for the helpful comment. An Excel file containing molecular structures of condensable species in the "Rdc" mechanism (Fig. 2) will be added as a new supplementary material.

**RC:** *Table2: please check the expressions of reactions and coefficients.*

**AR:** Sorry for the misleading. The wrong coefficient has been corrected in Table 2 of the paper (0.8 BCBO to 0.6 BCBO). We have also checked that the RO2 reaction of BCBO2 indeed produces species BCAOH, not BCBOH (which does not exist in the MCM mechanism).

**RC:** *L425-428: Some quantitative explanations associated with the results in L415-418 is necessary.*

**AR:** Thanks for the suggestion. We have added the explanation as follows:

> *Section 3.2.2, line 425:*
>
> The N/C ratio, however, is underestimated by the "Rdc." mechanism by 37 % on average (ratio equal to 0.019 in MCM and to 0.012 in "Rdc."), indicating the over reducing organic nitrites in  "Rdc.". A total of three nitrogen-containing organics (NBCO2, NBCOOH, and C131PAN) are preserved in "Rdc.", of which two (NBCO2, NBCOOH) are first-generation products. Therefore, during the first 10 hours, the N/C ratio curve simulated by "Rdc." drops, whereas in MCM it increases as higher-generation nitrates are produced.

**References**

[Couvidat et al., 2018] Couvidat, F., Bessagnet, B., Garcia-Vivanco, M., Real, E., Menut, L., and Colette, A. (2018). Development of an inorganic and organic aerosol model (chimere 2017$\beta$ v1. 0): seasonal and spatial evaluation over europe. *Geosci. Model Dev*, 11(1):165–194.

| Name of the UNIFAC groups | Index | NBCOOH | C141CO2H | BCKSOZ | C133CO | C132OOH | C131PAN |
|---|---|---|---|---|---|---|---|
| alkane CH3 | 0 | 3.00E+00 | 1.89E+00 | 2.90E+00 | 8.20E-01 | 2.97E-02 | 2.00E+00 |
| alkane CH2 | 1 | 5.00E+00 | 4.69E+00 | 3.10E+00 | 1.64E+00 | 5.73E-02 | 4.00E+00 |
| alkane CH | 2 | 2.00E+00 | 1.89E+00 | 2.00E+00 | 8.19E-01 | 1.70E-02 | 2.00E+00 |
| alkane C | 3 | 1.00E+00 | 9.45E-01 | 9.99E-01 | 4.10E-01 | 1.49E-02 | 1.00E+00 |
| methanol CH2OH | 5 | | 5.53E-02 | 5.24E-04 | 5.85E-01 | 9.85E-01 | |
| methanol CH2OH | 6 | | 6.90E-04 | 9.27E-07 | 3.09E-04 | | |
| methanol COH | 7 | | 1.42E-04 | 5.30E-04 | 6.96E-05 | 3.02E-04 | |
| calcohol between two alcohols OHCH2OH | 9 | | 4.09E-04 | 2.04E-05 | 1.22E-04 | 1.21E-03 | |
| calcohol between two alcohols OHCHOH | 10 | | 2.72E-04 | 1.30E-05 | 1.22E-04 | 6.05E-04 | |
| calcohol between two alcohols OHCOH | 11 | | 6.90E-04 | 6.48E-06 | 6.10E-05 | 3.02E-04 | |
| calcohol in tails of alcohol OHCH3OH | 12 | | 1.11E-01 | 1.34E-03 | 1.18E+00 | 2.27E+00 | |
| calcohol in tails of alcohol OHCH2OH | 13 | | 1.11E-01 | 2.34E-03 | 6.06E-01 | 3.47E+00 | |
| calcohol in tails of alcohol OHCHOH | 14 | | 1.05E-05 | 1.05E-03 | 1.00E-02 | 1.32E+00 | |
| calcohol in tails of alcohol OHCOH | 15 | | 5.47E-02 | 5.24E-04 | 5.90E-01 | 9.85E-01 | |
| alkene CH2=C | 18 | 1.00E+00 | 1.00E+00 | | | | |
| alkene CH=C | 19 | | 5.27E-06 | | | | |
| alcohol OH | 26 | | 5.61E-02 | 1.05E-03 | 5.90E-01 | 9.85E-01 | |
| ketone CH3CHO | 29 | | 9.92E-01 | 1.01E-01 | 9.95E-01 | 6.96E-01 | 1.00E+00 |
| ketone CH2CHO | 30 | | 1.24E-01 | 9.99E-01 | 2.17E+00 | 6.69E-01 | 1.00E+00 |
| aldehyde CHO | 31 | | 1.42E-02 | 1.01E-01 | 5.42E-03 | 1.88E-01 | |
| ester CH3COO | 32 | | | | 5.30E-03 | 3.75E-03 | |
| ether CHO | 36 | | | 8.99E-01 | | 2.96E-01 | |
| carboxylic acid COOH | 37 | | 9.39E-01 | | 4.10E-01 | 1.89E-02 | |
| nitrate CHONO2 | 40 | 1.00E+00 | | | | | |
| hydroxyperoxide CH2O-OH | 42 | | | | | 2.59E-04 | |
| hydroxyperoxide CHO-OH | 43 | | | | | 1.68E-01 | |
| hydroxyperoxide CO-OH | 44 | 1.00E+00 | | | | 8.32E-01 | |
| hydroxyperoxide CHO-OC | 52 | | | 8.99E-01 | | 2.96E-01 | |
| peroxyacyl nitrates PAN | 54 | | | | | | 1.00E+00 |

Figure 2: UNIFAC functional group decomposition for the condensable species in the "Rdc." mechanism.

---

## Author Comment (AC2)

**Authors' Response to Reviews of**

**GENerator of reduced Organic Aerosol mechanism (GENOA v1.0): An automatic generation tool of semi-explicit mechanisms**

Zhizhao Wang, Florian Couvidat, Karine Sartelet
*Geoscientific Model Development,* `https://doi.org/10.5194/egusphere-2022-245`
* * *
RC: *Reviewers' Comment*,     AR: Authors' Response,     ☐ Manuscript Text

**Reviewer # 2**

**RC:** *The paper discusses the development of a semi-explicit reduced organic aerosol mechanism for sesquiterpenes (Bcary), with the aim for employing it in air quality and large scale models. GENOA mechanism is based on the widely used near-explicit Master chemical mechanism (MCM). The mechanism used different strategies namely, lumping, replacing, jumping an removing to reduce the MCM scheme. The reduction procedure is tested under various environmental conditions (RH, temp etc.), resulting in a final reduced mechanism (Rdc.) suitable to simulate SOA. The simulated SOA using Rdc has low average error when compared to the near-explicit MCM scheme. This is a well thought out work, with suitable implications to better reproduce SOA in large scale and air quality models. I would therefore, recommend the publication of this work after the authors have answered the following questions:*

**AR:** We would like to thank reviewer # 2 for the positive comments and constructive suggestions, which are much useful to improve the manuscript. We have carefully considered all of these comments and revised the manuscript accordingly. Please note that the line and section numbers mentioned in the response correspond to the version of the manuscript before revision.

**1. General comments**

**RC:** *The main question is why did the authors chose sesquiterpenes? Why not isoprene or monoterpenes? The motivation to use sesquiterpenes should be highlighted.*

**AR:** Thank you for the questions. Sesquiterpene was selected because it is a well-known source of SOA, and its formation mechanism is well documented. Sesquiterpene is also an ideal candidate for model development and demonstration of the reduction methodology. The oxidation products of sesquiterpene are less volatile and tend to condense more readily than those of lighter molecules such as monoterpene and isoprene.

We have added the explanation to the main paper:

> *Section 1, line 72:*
>
> The application of GENOA to the MCM degradation scheme of $\beta$-caryophyllene (BCARY) (Jenkin et al., 2012) is described in Sect. 3.  $\beta$-caryophyllene  is selected

for investigation and demonstration of the GENOA algorithm, because it is one of the most abundant and representative sesquiterpene (SQT)(e.g., Li et al., 2015; Xavier et al., 2019), and its degradation has been evaluated in chamber simulations (Jenkin et al., 2012). The BCARY scheme in the STOCHEM-CRI mechanism (Khan er al., 2017) and . Sesquiterpenes are well-known source of SOA (Hellén et al., 2020; Tasoglou and Pandis, 2015 and the formation mechanism of BCARY is well documented in the experiment data of Tasoglou and Pandis (2015) are also used for evaluation of near-explicit MCM mechanism (Jenkin et al., 2012). Studies have also compared BCARY SOA yields simulated using the MCM mechanism to chamber data (e.g., Xavier et al., 2019). BCARY is therefore an ideal candidate for model development and demonstration of the reduction methodology. In this paper, the near-explicit MCM BCARY degradation scheme serve as a reliable benchmark for GENOA. The experiment data from Tasoglou and Pandis (2015); Chen et al. (2012) are also compared to the newly developed reduced mechanism in Appendix A.

**RC:** *Since GENOA is a semi-explicit mechanism, can it be used with any box or air quality model?*

AR: Yes. The semi-explicit SOA mechanisms generated by GENOA can be added to the implicit gas-phase mechanism to model SOA in any box or air quality model. In the case of the sesquiterpene SOA, the "Rdc." mechanism adds 14 organic species including six condensables to SOA models, which is computationally feasible even for global 3-D modeling.

Generally, implicit SOA mechanisms applied to 3-D models are often based on the surrogate approach (e.g., Odum's two-product) or the Volatility Basis Set. They are added to implicit gas-phase mechanisms without altering the pathways of oxidants. The same approach is adopted for the semi-explicit SOA mechanisms. The reduced SOA mechanism includes gas-phase chemical mechanism and aerosol properties for condensable species, which can be added to the implicit gas-phase mechanism in air quality models, without altering the pathways of ozone and major radicals.

For example, the 3-D CHIMERE model can simulate SOA using the implicit gas-phase mechanism MEL-CHIOR2 and SOA mechanism $H^2O$ [Couvidat et al., 2018]. With the reduced SOA mechanisms, SOAs in CHIMERE are simulated by MELCHIOR2, the reduced SOA mechanisms generated by GENOA, and $H^2O$ for other SOA precursors not covered by our mechanisms.

In the paper, We have added the explanation of how the coupling is generally resolved between VOC species and oxidants in 3-D models:

*Section 1, line 36:*

Along with To complete implicit gas-phase mechanisms, implicit SOA mechanisms have been developed to model specifically the SOA formation without modifying the concentrations of ozone and major radicals (Kim et al., 2011) . In 3-D modeling, implicit SOA mechanisms or parameterizations are usually added to implicit gas-phase mechanisms, conserving the oxidant chemistry of the implicit gas-phase mechanism.

Implicit SOA mechanisms are often established based on experimental data from smog chamber experiments to represent the formation and evolution of SOA, such as the two-product empirical SOA model (Odum et al., 1996) and the volatility basis set (VBS) that splits VOC oxidation products into a uniform set of volatility "bins" Donahue et al., 2006).

We have also added the following statements about our semi-explicit mechanisms:

> *Section 2, line 80:*
>
> The generated semi-explicit mechanisms are designed to preserve the accuracy of explicit mechanisms for SOA formation, while keeping the number of reactions/species low enough to be suitable for large-scale modeling, particularly  in 3-D AQMs. The focus of the semi-explicit mechanism is solely on the accurate modelling of SOA. Because ozone, major radicals, and other inorganics are also affected by inorganic and other VOC chemistry, their concentrations are not tracked with the semi-explicit mechanism. Instead, they are simulated using existing implicit gas-phase chemical mechanisms.

**RC:** *Although comparison has been made against MCM, the performance of a model can be made by comparing it against exisiting experimental SOA yields. There has been quite a lot of published experimental Bcary SOA yield experiments. I would suggest the authors to discuss GENOA derived SOA yields in comparison to these experiments.*

AR: As GENOA is used to reduce the MCM mechanism, we compared the mechanisms before and after the reduction to ensure that the generated SOA mechanism preserves the performance of the explicit mechanism on SOA formation.

We have also compared SOA yields simulated by MCM and "Rdc." mechanisms to the experimental data from [Tasoglou and Pandis, 2015] and [Chen et al., 2012] in Appendix A. As shown in Fig. A1 of Appendix A, the results of the "Rdc." mechanism (noted as "Rdc.") are in good agreement with the experimental data (noted as "Tasoglou." and "chen.") and the results of the MCM mechanism ("v1b2"). Moreover, as now mentioned in the paper, [Xavier et al., 2019] has already performed some evaluations of BCARY SOA from the MCM mechanism against to chamber data.

**RC:** *Is GENOA a carbon number conserving mechanism. Its is not clear from the manuscript if the mechanism is carbon conserving or not? If it is not then how do the authors justify it?*

AR: Strictly speaking, the carbon number is not conserved explicitly during the reduction process. However, GENOA is designed to provide a good estimation of the contribution of the different functional groups. Carbon number is constrained in reduction by lumping (the difference in the carbon number between lumped species cannot exceed 2), jumping and replacing (the difference cannot exceed 3). For lumping and replacing, there is also a restriction on the total mass ($< 100$ $\mu$g/m$^3$). For removing, there are no restrictions on either mass or carbon number.

Consequently, as we always constrain the total SOA concentration with strict error criteria (i.e., $\epsilon_{ref}$ and $\epsilon_{pre}$), the OM/OC, H/C, and O/C ratios are well reproduced (see Fig. 9 in the paper), so the carbon number should be also be well reproduced. If necessary, a specific restriction on conserving the carbon number can be added by the user in the reduction.

**2. Specific questions**

**RC:** *L13-15: Motivation -> Although the health and climate effects of aerosols are introduced in every paper, the authors should maybe consider to explain these in a few words or a sentence. Also, it would be nice to explain why there is a need to improve the SOA representation in AQMs.*

AR: Thank you for the suggestion. We have rephrased the motivation as follows:

*Section 1, line 1:*

Atmospheric aerosols  attract attentions due to their  effects on climate and human health: they change the earth's radiation balance and cloud formation (Ramanathan et al., 2001; McNeill, 2017); they trigger a wide variety of acute and chronic diseases (Breysse et al., 2013). Because the effects of aerosols on health depend on their size and composition (Schwarze et al., 2006), adequate representations of the aerosol composition, mass and number concentrations are required in air quality models (AQMs).

Besides being directly emitted, aerosols can be secondary, i.e., formed in the atmosphere through chemical reactions and gas-particle mass transfer  Based on the chemical composition, they can be further divided into secondary inorganic aerosol (SIA) and secondary organic aerosol (SOA). SOA, which represents a significant fraction of aerosols (e.g., Gelencsér et al., 2007, is largely formed by the condensation of the oxidation products from the degradation of volatile organic compounds (VOC). As SOA formation involves multiple processes such as the emission of SOA precursor gases, VOC gas-phase chemistry, gas-to-particle partitioning (Kanakidou et al., 2005; Hallquist et al., 2009), there are great complexity and uncertainty to accurately predict SOA formation with the simplified representations currently used in air quality models (Porter et al., 2021). .

**RC:** *L18-22: This sentence seems to contain quite a lot of information. I would suggest the authors to rephrase it into smaller sentences.*

**AR:** Rephrased.

*Section 1, line 18:*

 The state of knowledge on VOC chemistry can be reflected by explicit chemical mechanisms,  which contain all known important reaction pathways in VOC degradation. For instance, Jenkin et al. (1997); Saunders et al. (2003) developed the near-explicit Master Chemical Mechanism (MCM) , which describes detailed gas-phase chemical processes related to VOC oxidation. Another example is the Generator for Explicit Chemistry and Kinetics of Organics in the Atmosphere (GECKO-A) (Aumont et al., 2005), which uses a prescribed protocol to assign complete reactions pathways and kinetic data to the degradation of VOCs. Explicit mechanisms represent the  current understanding of atmospheric chemistry, including information about reaction pathways, kinetics data, and chemical structures (which may be used to deduce thermodynamic properties based on structure-activity relationships).

**RC:** *L24: "box models". Although, it is true that explicit schemes are used in box models due to relaxed computational burdens, they have been also been used in 1-D column models or 2-D Lagrangian models (these are still not as computationally expensive compared to AQMs or Global climate models).*

**AR:** Thank you for pointing this out. The paragraph has been revised to emphasize the limitations of using explicit mechanisms in modelling, and to include the use of explicit mechanisms in other models besides box models:

> *Section 1, line 24:*
>
>  The MCM mechanism has been used by 2-D Lagrangian models to simulate the chemical evolution of major air pollutants and some SOAs in plumes (e.g., Evtyugina et al., 2007; Sommariva et al., 2008; Zhang et al., 2021). Moreover, it has been used for simulating the formation of more complex SOAs at a regional level in 3-D models over a few weeks (e.g., modified MCM with 4642 species and 13,566 reactions in the simulations of Ying and Li (2011), and with 5727 species and 16,930 reactions in the simulations of Li et al. (2015)). Even so, explicit mechanisms of that size are too computationally intensive to be  widely employed in 3-D AQMs for SOA formation.

**RC:** *L32: "carbon-bond" instead of "carbon-bound".*

**AR:** Changed.

**RC:** *L34-35: Are all the above mechanisms (lumped, CB05, MCM, GECKO-A) developed primarily for ozone simulation? Also it would be good to give examples of a few model species.*

**AR:** Because grand-level ozone is one of the most important air pollutants, most gas-phase chemical mechanisms used in air quality models focus primarily on predicting accurate ozone concentration. As this may not be the case for all models, we have revised this phrase to be more precise:

> *Section 1, line 34:*
>
>  Implicit gas-phase mechanisms were developed and validated to simulate the concentrations of oxidants and other conventional air pollutants such as ozone and $NO_2$. In these mechanisms, VOCs have been grouped into a limited number of model species because of computational considerations, and the SOA formation is usually not considered.

**RC:** *L55: "suitable to"*

**AR:** Revised.

**RC:** *L112-113: Why is this order used in the reduction strategy for BCARY? How would any other order influence the reduction strategy?*

**AR:** This reduction order was the most effective among all the reduction orders we tested. As each validated reduction can affect the subsequent reductions, reductions with small changes or in favor of other reductions are preferred to be run first. Hence, we adopted this reduction order:

First, the strategies of removing reactions (deleting trivial reactions) and jumping (jumping over negligible species) are tested, which trim the scheme for further reduction. In the following step, lumping and replacing (extension of lumping) are applied, which results in a significant merge of both reaction pathways and species. Finally, the strategy of removing species is adopted, following removing gas-particle partitioning for condensable species that cannot be removed with removing species.

We have added the explanation to the paper:

> *Section 2.1, line 112:*
>
> For the BCARY reduction, the reduction strategies are employed in the following order: removing reactions, jumping, lumping, replacing, removing species, and finally removing gas-particle partitioning.
>
> The reduction strategies are ordered based on their potential influences on the mechanism. The first applied strategies, removing reactions and jumping, trim trivial reactions and species without altering the properties of the species. They are followed by lumping and replacing (as an extension to lumping), which refine the mechanisms considerably by merging the species and reactions involved. Afterwards, the "removing species" strategy attempts to delete all merged and unmerged species. Finally, the strategy of removing gas-particle partitioning is applied in order to remove the partitioning of condensable species, which cannot be removed by removing species. This current order has been tested and found to be efficient for the BCARY mechanism, but it can be changed by the user along with other user-chosen parameters shown in Table 5.

**RC:** *Table1: Typo in reaction 8. I think it is supposed to say 0.2 BCBOH instead of 0.2 BCAOH.*

 AR:  Actually, both BCAO2 and BCBO2 (Fig. 1) form the same compound BCAOH (Fig. 2) through the self- and cross-reactions of peroxy radicals ( RO2-RO2 reaction). As a consequence, BCBOH does not exist in MCM mechanism.

Figure 1: The molecular structures of the MCM species BCAO2 (left) and BCBO2 (right).

Figure 2: The molecular structure of the MCM species BCAOH.

**RC:** *Table2: Typo in lumped reaction R2: 0.753 * (fw,a BCANO3 + fw,b BCBNO3 + fw,c BCCNO3). I would guess the factor is 0.247 instead of 0.753.*

 AR:  Corrected.

**RC:** *Table2: Cr,b and Cr,c are not defined. It should be defined similar to Cr,a for better clarity.*

 AR:  Thanks for the suggestion. The definitions of $C_{r,b}$ and $C_{r,c}$ have been added to Table 2.

**RC:** *L119: It would be much clear to write it as "In this example, a total of 12 chemical reactions involving three organic compounds are reduced to five reactions (4 lumped (R1-4) + 1 surrogate (R0))".*

**AR:** Thanks for the suggestion. The sentence has be rephrased:

> *Section 2.1.1, line 119:*
>
> In this example, a total of 12 chemical reactions involving three organic compounds are reduced to  five reactions (new surrogate production R0 and four lumped reactions R1 - R4).

**RC:** *L122-123: How are the BCARY isomers undergoing similar reactions with HO2,NO and NO3? Are the authors referring to the R1-4 in the lumped scheme. Please make this clear to the readers.*

**AR:** Here we wanted to point out that MCM species BCAO2, BCBO2, and BCCO2 share similar structures and properties. The fact that they are isomers (which is not necessary for lumping) may explain why all of them reacted with the same species ($HO_2$, $NO$, $NO_3$, and other peroxy radicals ($RO2$)). As long as species meets the lumping criteria, they can be merged together via lumping, and their reactions with different oxidants are also be lumped accordingly.

In order to avoid ambiguity, we have rephrased the sentence as follows:

> *Section 2.1.1, line 122:*
>
> As demonstrated in the tables, the organic compounds BCAO2, BCBO2, and BCCO2 from the original MCM scheme are the peroxy radicals formed from the OH-initiated oxidation of $\beta$-caryophyllene (Table 2). It is evident from their structures (shown in fig. C1) that they are isomers and may share similar chemical properties.. When applying the lumping strategy, BCAO2, BCBO2, and BCCO2 are merged into a new surrogate named "mBCAO2" (Table 3).

**RC:** *Why is Cr,a,b,c an arithmetic mean of 5 day simulations? I.e was this 5 day period selected?*

**AR:** A five-day simulation period is chosen for calculating $C_{r,a,b,c}$ and all other reduction parameters, as a compromise between a shorter period that may not reflect the aging of SOA, and a longer period that may less adequately address the SOA formation and be computational expensive. As the concentrations of ozone, radicals, inorganics and environmental parameters (e.g., temperature, relative humidity) are extracted from the 3-D CHIMERE simulations (24-hour monthly averages). Their diurnal profiles are repeated for five days in 0-D simulations.

To clarify, the following explanation has been added for the settings of 0-D simulation:

> *Section 2.3, line 268:*
>
> Unless stated otherwise,  two simulations are performed for each condition starting at midnight (0 h) and noon (12 h), taking into account both the daytime and nighttime chemistry. All 0-D simulations are run for five days in order to consider adequately SOA formation and aging processes.

**RC:** *L129: kinetic -> kinetics*

**AR:** Corrected.

**RC:** *L130: weighting -> weighing*

**AR:** Corrected.

**RC:** *L135: What is this specific behavior?*

**AR:** Here we wanted to point out that different types of compounds may be involved in different types of reactions. A radical, for instance, may be better lumped with another radical than a condensable compound. A PAN compound has a decomposition reaction that does not have other compounds. Therefore, a restriction on certain structural groups was applied to the BCARY reduction.

The term "specific behavior" has been rephrased for clarity:

> *Section 2.1.1, line 135:*
>
>  Compounds with specific structural groups sharing common chemical behavior may be more appropriately merged together. Thus, compounds containing the following functional groups can only  be lumped with compounds containing the same groups: peroxyacetyl nitrates (PAN), organic nitrates (RONO2), organic radicals (R), oxy radicals (RO), peroxy radicals (RO$_2$), carboxylic acids (RC(O)OH), percarboxylic acids (RC(O)OOH).

**RC:** *L162: Aren't alkoxy radicals are RO.*

**AR:** Corrected.

**RC:** *L179: Can the authors explain what the maximum hourly branching ratio is?*

**AR:** We have added the definition of the maximum hourly branching ratio to the main paper:

> *Section 2.1.4, line 178:*
>
> There is no particular restriction to exclude species from the reduction attempt via the strategy of removing compounds or removing gas-particle partitioning. However, for removing reaction, a threshold  on the branching ratio of the reaction is applied to the reduction. The branching ratio is defined as the ratio of the destruction rate of one reaction to the sum of the destruction rates of all reactions of the targeted species. In the BCARY reduction, a maximum branching ratio ($B_{rm}$) is  defined as a restriction criterion. All reactions with hourly branching ratio (averaged over the training conditions) under this value (reactions that are likely to a minimal effect on SOA formation) are considered as candidate for removal.

**RC:** *L185: Do the authors mean that that after one loop of reduction (as shown in Figure 1), the subsequent reduction is carried out in the reverse order? And why is saturation vapor pressure used only for the lumping strategy. This part needs more clarity.*

**AR:** Sorry for the ambiguity. For each strategy, the search for potential reduction is conducted following the reverse lists of reaction/species. For example, with removing reactions, GENOA attempts to remove the reaction from the end to the beginning of the reaction list. When applied to the jumping strategy, GENOA

tries to jump the species that has the highest generation and then move down to the species that has the lowest generation.

In lumping, we consider that a condensable species should be first grouped with another compound that has a similar volatility. Thus, the saturation vapor pressure is used to determine the most appropriate lumpable species. The saturation vapor pressure of species is not affected by other strategies and is therefore not used as a criterion for them.

We have added the explanation to the paper:

> *Section 2.1.4, line 185:*
>
> Moreover, the searches for viable reductions via removing are conducted in reverse order of the reaction/species list, which means that GENOA attempts to remove reactions from the bottom of the list and moves to the previous reactions. The same reverse sequence is followed for other strategies. When applied to the jumping strategy, for instance, GENOA tries to jump the species that has the highest generation and then move down to the species that has the lowest generation. Among all reduction strategies, only lumping alters the saturation vapor pressure of condensable species. Therefore, a rank of saturation vapor pressure  is used exclusively  in lumping to determine the most appropriate lumpable species.

**RC:** *Table 5: Are the two conditions ADD1 and ADD2 high or low NOx regimes?*

AR: ADD1 is under low $NO_x$ regime and ADD2 is under high $NO_x$ regimes. We have added columns in Table 1 (Table 5 in the paper), i.e., the average SOA concentration and NO reactive ratio with $RO_2$ ($R_{RO_2-NO}$), which more clearly indicates the chemical regimes of conditions. If $R_{RO_2-NO}$ is high, the conditions are in the high $NO_x$ regime. Otherwise, the conditions are in the low $NO_x$ regime. For clarity, the digits in Table 5 of the paper have also been changed for temperature and relative humidity.

**RC:** *Figure 2: How do I interpret this? Is the top bar representing 0H and the bottom one representing 12h? Please add this to the figure caption, since 0 h and 12 h are not represented by empty and hashed lines.*

AR: Thank you for pointing out the error in Figure 2. The hash line for the 12-h condition was not shown. The figure (Figure 2 in the paper) has been revised to Fig. 3.

**RC:** *L233: I wonder what are the conditions which cause such low O3 and NO3 concentrations?*

AR: This condition is located in the northern part of Italy, within the Alpine arch, close to the metropolitan city of Milan. The concentrations of NO transported from polluted areas consume $O_3$ and $NO_3$ and produce $NO_2$, which explains the low concentrations of $O_3$ and $NO_3$ in this area.

We have added the more information about this "ADD2" training condition to the paper:

Table 1: Geographic and meteorological conditions of the training dataset

| Condition Name[a] | Lat | Lon | Time | TEMP | RH | $R_{NO}$[b] | SOA[c] |
|---|---|---|---|---|---|---|---|
| | °N | °E | month | K | % | % | $\mu g/m^3$ |
| OH NO | 36.0 | 15.4 | Jul. | 299.4299 | 78.679 | 60 | 4.1 |
| OH HO$_2$ | 32.0 | -9.4 | Jul. | 295.9296 | 76.777 | 20 | 6.1 |
| NO$_3$ NO | 40.25 | -3.4 | Jul. | 302.4302 | 27.928 | 69 | 4.4 |
| NO$_3$ HO$_2$ | 32.0 | 36.6 | Aug. | 302.2302 | 38.738 | 29 | 5.7 |
| O$_3$ NO | 69.0 | 33.8 | Jan. | 260.7261 | 84.284 | 99 | 5.2 |
| O$_3$ HO$_2$ | 68.0 | 18.2 | Dec. | 265.5266 | 88.789 | 25 | 4.6 |
| ADD1 | 41.5 | -14.2 | Dec. | 288.6289 | 75.876 | 20 | 5.5 |
| ADD2 | 45.75 | 9.0 | Dec. | 279.1279 | 84.585 | 100. | 4.4 |

[a] from left to right: name, latitude, longitude, time period, average temperature, average relatively humidity, daily average NO reacting ratio, simulated total SOA concentration of the training conditions.
[b] the daily average NO reacting ratio is calculated out of the RO$_2$ reactivity of NO, HO$_2$, NO$_3$, and RO$_2$. Conditions with high $R_{NO}$ ratio are considered as in high NO$_x$ regime. [c] the initial concentration of BCARY is 5 $\mu g/m^3$.

[Figure]

Figure 3: A bar plot showing the occupancy of seven reacting ratios in BCARY initiation reactions and RO$_2$ reactions, under the training conditions at midnight (0 h, top bar) and noon (12 h, bottom bar).

> *Section 2.2, line 232:*
>
> One specific exception is the additional condition ADD2,  which is located in the northern part of Italy, within the Alpine arch, close to the metropolitan city of Milan. This condition is in extremely high-$NO_x$ regime, as high concentrations of NO are transported from polluted areas. These high NO concentrations consume $O_3$ and $NO_3$, causing low concentrations of $O_3$ and $NO_3$. At night, ADD2 has a high $R_{OH}$ of 95 % at midnight is not due to an abundance of OH, but rather to extremely low concentrations of $O_3$ ($2.9 \times 10^{-4}$ ppb) and $NO_3$ ($1.1 \times 10^{-9}$ ppb) that leads to an absence of nighttime reactivity.

**RC:** *L258: Is Kelvin effect not taken into account for gas-to-particle partitioning?*

**AR:** The Kelvin effect is not taken into account in the simulations, as thermodynamic equilibrium is assumed between gas and particle phases.

**RC:** *L262: "vapor pressure is computed using Mydral and Talkowsky". This phrase is repeated already in the earlier part of the sentence.*

**AR:** Rephrased.

**RC:** *Appendix A: It should be "v1:Nannolal (2008)" not Nannolal 2004.*

**AR:** Corrected.

**RC:** *Appendix A, L485: There seems to be a contradiction here. The authors say that v1b2 (Nannolal and Jacob and reid) show the best estimate in comparison with the experiments "As shown in Fig.A1, the SOA distribution simulated with "v1b2" agrees best with the experimental data. Therefore, this method with the vapor pressure computed by Myrdal and Yalkowsky (1997) and the boiling point computed by Joback and Reid (1987) is used in the BCARY reduction". But in the manuscript why has Mydral and Talkowsky been selected in place of Nannolal 2008. Why?*

**AR:** Thank you very much for pointing out this error. "v1" is the method of Nannolal et al. (2008), but in the paper, it was incorrectly referred to as the method of Myrdal and Yalkowsky (1997).

We have corrected the typo:

> *Appendix A, line 481:*
>
> Eight methods are provided in UManSysProp, including SIMPOL.1 of Pankow and Asher (2008) ("sim"), EVAPORATION of Compernolle et al. (2011) ("evp"), and six methods out of the combination of two methods to compute the vapor pressure ("v0": Myrdal and Yalkowsky (1997)and "v1": Nannoolal et al. (2008))) and three methods to compute the boiling point ("b0": Nannoolal et al. (2004), "b1": Stein and Brown (1994), and "b2": Joback and Reid (1987)). As shown in Fig. A1, the SOA distribution simulated with "v1b2" agrees best with the experimental data. Therefore, this method with the vapor pressure computed by Nannoolal et al. (2008)) and the boiling point computed by Joback and Reid (1987) is used in the BCARY reduction. The results simulated with the final reduced mechanism "Rdc." is also presented in Fig. A1, which has a great resemblance to the experimental data.

**RC:** *L275: I think it would be better to rephrase the sentence defining the FME. Is it so that the simulation error is the larger FME of the two errors I.e the FME of day 1 and the FME of rest of the simulation days? This has*

*be more clear in the text.*

AR: The following sentences have been rephrased to better explain how we calculate error:

> *Section 2.4, line 274:*
>
>  The error of one simulation is defined as the larger of the FME on day one and the FME on days two to five, in order to address the difference in performance of the reduced mechanisms at the early stage of the simulations (SOA formation dominates) and at the later stage  (SOA aging dominates). This error is used to evaluate reduction by comparing it to the error tolerance specified in training For the evaluation on the training dataset, two errors are estimated compared to the previously verified reduced mechanism with a tolerance denoted $\epsilon_{pre}$, and the MCM mechanism with a tolerance denoted $\epsilon_{ref}$. The error tolerances are used to restrict both the maximum and the average (half of the tolerance) errors of the training conditions. As for the evaluation on the pre-testing dataset,  only the error compared to the MCM mechanism is calculated. The error tolerances $\epsilon_{pre-testing}^{ace}$ and $\epsilon_{pre-testing}^{max}$ are set to the average and maximum errors, respectively.

RC: *Eq 1: What does i1 and i2 represent?*

AR: i1 and i2 are meaningless and have been removed from Eq.1.

RC: *L283: Why does the $\epsilon_{pre}$ vary in such a way with respect to $\epsilon_{ref}$. More explanation is need here.*

AR: $\epsilon_{ref}$ and $\epsilon_{pre}$ are the criteria we set for evaluating the reduction. The difference between the two criteria is that $\epsilon_{ref}$ is compared to the reference mechanism and $\epsilon_{pre}$ is compared to the previous validated mechanism. $\epsilon_{ref}$ is used to track the performance of the reduction, while $\epsilon_{pre}$ is used to avoid large errors introduced by one reduction attempt. So logically, $\epsilon_{pre}$ should be less or equal than $\epsilon_{ref}$.

In practice, when $\epsilon_{ref}$ increases by 1 %, the value of $\epsilon_{pre}$ is set from 1 % (minimum value) to the value of $\epsilon_{ref}$ By doing this, GENOA first accepts reductions that introduce small errors compared to the previous validated mechanism, and then accepts reductions that introduce larger errors up to $\epsilon_{ref}$.

The explanation has been added to the paper:

> *Section 2.4, line 282:*
>
> In order to begin with a conservative BCARY reduction, the initial values of $\epsilon_{pre}$ and $\epsilon_{ref}$ are both set to 1 %. The values of these error tolerances are then increased to larger values, reflecting the looser criteria used throughout the reducing. $\epsilon_{ref}$ is used to track the performance of the reduction, while $\epsilon_{pre}$ is used to avoid large errors introduced by one reduction attempt. Therefore, $\epsilon_{pre}$ is lower or equal than $\epsilon_{ref}$. For every 1 % increase in $\epsilon_{ref}$, $\epsilon_{pre}$ is stepped up by 1 % from 1 % to the value of $\epsilon_{ref}$. By doing this, GENOA first accepts reductions that introduce small errors compared to the previous validated mechanism, and then accepts reductions that introduce larger errors up to $\epsilon_{ref}$.

RC: *L288: Is this true? As mentioned the <3% avg error is for pre-testing + training dataset. Will the average error still be < 3% for test dataset?*

AR: For the BCARY reduction, the pre-testing dataset selected can provide an accurate representation of the average conditions in the testing dataset. Thus, when the average error of the pre-testing condition is less than

3 %, the error of the testing dataset is less than 3 % as well.

Generally, it depends on whether the pre-testing dataset is representative of the testing dataset. As the pre-testing dataset is selected randomly from the testing dataset, it may not initially be a reliable representation of the testing dataset (e.g., the error of pre-testing is much smaller or much larger than the error of testing). It is necessary to modify the pre-testing dataset in such a case. For example, a few conditions with large errors from the testing dataset can be added to the pre-testing dataset to improve the performance of pre-testing if the error of pre-testing is smaller than the error of testing.

We have added the explanation to the paper:

> *Section 2.2.2, line 240:*
>
> Meanwhile, the size of the mechanism has already been significantly reduced, which makes the evaluation of each reduction attempt on the pre-testing dataset less computationally expensive.
>
> In principle, the pre-testing dataset should be able to provide a fairly accurate representation of the testing dataset. However, this may not always be the case, since the pre-testing dataset is selected almost randomly from the testing dataset. Therefore, an adjustment may be required to increase the representativeness of the pre-testing dataset by adding or removing a few conditions.

**RC:** *L303: How does GENOA decide what condensable species to remove? Is it based on species super saturation values?*

AR: In the reduction via removing gas-particle partitioning, GENOA tries to remove the partitioning of each condensable regardless of the saturation vapor pressure. The reduction is accepted only if the errors of training/pre-testing conditions are small enough.

In the late-stage reduction, the aerosol-oriented treatments are applied. Since there is strong competition among reduction strategies at the late stage of the reduction process, these treatments are used to reduce species rather than reactions, thereby reducing condensable species.

To avoid confusion, we have rephrased the statements about the aerosol-oriented treatments:

> *Section 2.5, line 301:*
>
> These treatments, which reduce species rather than reactions, are done when the size of the mechanism is below a certain threshold (20 for BCARY reduction).  Consequently, the late-stage treatments encourage the reduction via the removing of condensable species, and are referred to as the aerosol-oriented treatments.

**RC:** *L326: Effective partitioning coeff. is temperature dependent. What is the assumed temperature for the classification of SVOCs, LVOCs and ELVOCs? And since the authors use a range of temperatures in their training conditions (268-302 K) does it really make any sense in classifying the SVOCs, LVOCs and ELVOCs at an arbitrary temperature?*

AR: The effective partitioning coefficient ($K_p$) is computed at 298 K. We have added the assumed temperature for $K_p$ in the paper.

**RC:** *L332: Are the species in the reduced Rdc. and Khan 2017 mechanism overlapping or identical?*

AR: The mechanism of [Khan et al., 2017] preserves information regarding the reaction/species of the first and

second generations of the MCM mechanism. In contrast, the "Rdc." mechanism may preserve information concerning up to the tenth generation of MCM. Thus, the "Rdc." mechanism may provide a more detailed description of SOA formation and aging than the mechanism of [Khan et al., 2017].

**RC:** *L341: Condensable species drop to less than 20 ? It seems that the condensable species is 10 % of its original values (10% of 493 48/49 species or thereabout) as shown in Figure 3. Also $\epsilon_{pre}$ is not shown in Figure 3. Also on L 346 it is mentioned as 41 species on the 75th time. So what is the correct number?*

**AR:** The number of condensable species indeed dropped to 20 at the $74^{th}$ step, with a fraction of 5.61 % and the initial number of condensable species in MCM BCARY mechanism is 356. As of the beginning of the $75^{th}$ step (by the end of the $74^{th}$ step), the number of species is 41, including both condensable and non-condensable species.

Though the numbers in the text have been verified to be accurate, the number of reduction steps might not be clear. According to figure 3 in the paper, the size at the $n^{th}$ step refers to the size at the end of the $n^{th}$ reduction step. However, in the text, we confused it with the size at the beginning of the $n^{th}$ reduction step. Accordingly, the following statements have been revised:
* * *
*Section 3.1, line 341:*

- Early stage, from the first to the 74 $^{th}$ reduction step. By the end of the 74 $^{th}$ reduction step, the mechanism is reduced to 68 reactions and 41 species (including 20 condensable species). The early-stage reduction is trained only on the training dataset with the seven pre-described reduction strategies. After $\epsilon_{ref}$ reaching 3 %, the list of $B_{rm}$ is changed from [0.05, 0.10, 0.50] to [0.10, 0.50, 1.0].

- Late stage I, from the 75 $^{th}$ to the 107 $^{th}$ reduction step. By the end of the 107 $^{th}$ reduction step, the reduced mechanism consists of 38 reactions and 19 species (including 7  condensable species), and no further reduction can be found within $\epsilon_{ref} \leq 10$ % and $\epsilon_{pre} \leq 10$ %.  In this stage, the reduction is trained on the pre-testing dataset if the condensable species are removed with lumping, replacing, or jumping. For reduction with other types of reduction strategies, it is first trained on the training dataset and then on the pre-testing datasets. From all reduced mechanisms with seven condensable species, GENOA selected the one with the minimum average errors on pre-testing dataset (2.44 %) to start the next stage.

- Late stage II, from the 108 $^{th}$ to the 113 $^{rd}$ reduction step. At this stage, the reduction strategy of removing elementary-like reactions is applied to the training. All reductions that reduce the condensables are  evaluated exclusively on the pre-testing dataset.  The size of the reduced mechanism was reduced  to 23 reactions and 15 species, among which the number of condensable species is reduced  to 6. The average (maximum) error of the final reduced mechanism "Rdc." is 2.65 % (17.00 %) under the pre-testing dataset compared to MCM.
* * *
As $\epsilon_{pre}$ is only compared to the previously validated mechanism, it does not reflect the performance of the mechanism as does $\epsilon_{ref}$. Therefore, we keep $\epsilon_{ref}$ only as an estimation of the performance of the reduction at different stages of the reduction in Figure 3 of the paper.

**RC:** *L346-348: Aren't all the reduction strategies trained first with training data sent and then with pretesting data*

*set? So what is the difference here between, lumping, jumping and replacing compared to other strategies?*

AR: In the late stage of reduction, the reduction via removing is evaluated first on the training dataset, and then on the pre-testing dataset, whereas the reduction via lumping, replacing, and jumping is evaluated only on the pre-testing dataset. The reason is that lumping, replacing, and jumping may be more effective in terms of altering the scheme. Compared to removing, they offer more possibilities for reducing species. Therefore, reductions via lumping, replacing, and jumping are evaluated only on the pre-testing dataset. This treatment was tested and adopted for the BCARY reduction. It can be turned off by the user when applied to other reductions.

RC: *L351: "evacuated"-> "evaluated"*

AR: Corrected.

RC: *L358: It should be specified that lumping reduces the condensable species by 35 %.*

AR: We have removed the misleading statement:

> *Section 3.1, line 357:*
>
> As expected, the reduction strategy of removing reaction contributes the most to the decrease in the number of reactions (48 %), followed by the strategy of removing species with a contribution of 37 %. Meanwhile, both lumping and removing species are significant in the reduction of species, by 35 % and 31 %, respectively.

RC: *L366: There is not mC133O in Figure 4.*

AR: C133O in the "Rdc." mechanism has the same properties as the one with the same name in the MCM mechanism. Therefore, the name C133O remains unchanged after reduction.

RC: *L392: Why is more uncertainty found in regions with low RH and high temperatures? Is it because the training data set does not have enough data to work with in these conditions? Looking at figure 6, it shows that even Russian data points have high uncertainty between 3-6%. This cannot be only due to low RH and high temperature. What could be the other possible reasons for that?*

AR: The relatively high error in the regions with low RH and high temperatures indicates that such conditions are not sufficiently represented in the pre-testing dataset. As these conditions are rare to encounter over Europe, they are not included in the pre-testing dataset, such as not increasing the size of the mechanism.

Considering that the average error for pre-testing is 3 %, there are, of course, conditions with errors greater than 3 %, which are placed in the error category between 3 % and 6 %. As we be seen in Fig. 5 of the main paper, the testing errors in July and August (corresponding to the results in Fig. 6 of the paper) are actually very close to 3 %. These conditions are scattered throughout Europe, such as in the Russian area, in northern Europe, as well as in the Mediterranean.

RC: *Figure 8: Didn't the authors claim that PAN is under-represented in GENOA on L 418? But Figure 8 shows higher PAN concentrations for GENOA and lower for MCM.*

AR: We apologize for the mix-up in group names in the paper, and there is no doubt that the PAN concentrations are overestimated by the reduced mechanism shown in Figure 8 of the paper.

We have corrected the typo:

> *Section 3.2.2, line 415:*
>
> In comparison to MCM, only two condensable species containing nitrogen are retained in the "Rdc." mechanism: NBCOOH and C131PAN, leading to an  organic nitrate group ( 0.31 in MCM and  0.04 in "Rdc.") and an  overestimation of the nitrate mass of the peroxyacetyl nitrate group ( 0.10 $\mu$g m$^{-3}$ in MCM and  0.30 in "Rdc.").

**RC:** *L438: It should be explained why due to different volatility species Rdc delays SOA production. Is it due to low LVOC concentrations or high ELVOC concentrations?*

AR: Sorry for the misleading. The statement "Rdc delays SOA production" may not accurately describe the general situation in simulations that involve extreme SOA loading and large errors.

Thus, we have removed it from the discussion:

> *Section 3.2.3, line 336:*
>
> The result indicates that the "Rdc." mechanism may introduce relatively large uncertainty with extreme SOA loading (larger than 500 $\mu$g m$^{-3}$),  which was outside the range of conditions used for the construction of the "Rdc." mechanism.

**References**

[Chen et al., 2012] Chen, Q., Li, Y., McKinney, K., Kuwata, M., and Martin, S. (2012). Particle mass yield from $\beta$-caryophyllene ozonolysis. *Atmos. Chem. Phys.*, 12(7):3165–3179.

[Couvidat et al., 2018] Couvidat, F., Bessagnet, B., Garcia-Vivanco, M., Real, E., Menut, L., and Colette, A. (2018). Development of an inorganic and organic aerosol model (chimere 2017$\beta$ v1. 0): seasonal and spatial evaluation over europe. *Geosci. Model Dev*, 11(1):165–194.

[Khan et al., 2017] Khan, M., Jenkin, M., Foulds, A., Derwent, R., Percival, C., and Shallcross, D. (2017). A modeling study of secondary organic aerosol formation from sesquiterpenes using the stochem global chemistry and transport model. *J. Geophys. Res.-Atmos.*, 122(8):4426–4439.

[Tasoglou and Pandis, 2015] Tasoglou, A. and Pandis, S. N. (2015). Formation and chemical aging of secondary organic aerosol during the $\beta$-caryophyllene oxidation. *Atmos. Chem. Phys.*, 15(11):6035–6046.

[Xavier et al., 2019] Xavier, C., Rusanen, A., Zhou, P., Dean, C., Pichelstorfer, L., Roldin, P., and Boy, M. (2019). Aerosol mass yields of selected biogenic volatile organic compounds–a theoretical study with nearly explicit gas-phase chemistry. *Atmos. Chem. Phys.*, 19(22):13741–13758.

---

## Author Comment (AC3)

**Authors' Response to Reviews of**

**GENerator of reduced Organic Aerosol mechanism (GENOA v1.0): An automatic generation tool of semi-explicit mechanisms**

**Zhizhao Wang, Florian Couvidat, Karine Sartelet**

Geoscientific Model Development, https://doi.org/10.5194/egusphere-2022-245

RC: *Reviewers' Comment*, AR: Authors' Response,
Manuscript Text

**Reviewer #1**

**1. General comments**

**RC:** This paper discusses procedures developed to reduce the size of large gas/aerosol mechanisms to greatly reduced mechanisms that give predictions of secondary organic aerosol (SOA) that agree with those of the larger mechanism in a selected set of environmental conditions to within a specified tolerance. Because of the complexity of the atmospheric reactions or organic compounds, it is necessary to use reduced mechanisms in practical airshed model applications, but most reduced mechanisms used in airshed model applications were developed primarily focused on accurate ozone predictions. However, SOA predictions are much more affected by the chemical complexity of organic reactions than predictions of ozone, and developing methods to reduce mechanisms for without significantly affecting SOA predictions is an important research priority in atmospheric chemical mechanism development.

In fact, work with GECKO suggest that multi-generation mechanisms may be necessary for reliable SOA predictions, and even the MCM, which greatly lumps reactions of 2nd and higher generation products, may be too reduced for this application. However, the use of MCM as an example is sufficient to illustrate the method, and as discussed below MCM may be about the largest mechanism that could be reduced using the method discussed in this work, given current computer capabilities. In any case, without some way to reduce these huge mechanisms, we have no choice to continue to rely on the empirical and parameterized SOA models that are adjusted to fit SOA yields measured environmental chamber data without consideration of the actual chemistry and how different chemical conditions in the atmosphere affect SOA yields. If we had suitable reduction methods for SOA predictions, then use of SOA models that are based on actual chemistry might become practical.

AR: The authors would like to first thank Dr. William Carter for his detailed comments and insightful suggestions. Many of these suggestions are constructive and have been taken into account for the further development of the GENOA algorithm.

We have carefully considered the comments and revised the manuscript accordingly. Please note that the line and section numbers mentioned in the response correspond to the version of the manuscript before revision.

**RC:** The method discussed here involves use of a 3D grid model representing a large continental domain and various seasons, with SOA calculated using the large mechanism to be reduced as the starting point, and then uses sets of 0D scenarios derived from selected grid cells and times during the grid model simulation in an algorithm to develop the reduced mechanism. This has the disadvantage that it requires a full 3D calculation

with the full mechanism, which may be possible for MCM, but not for huge GECKO-like mechanisms that are probably what are really required. It may be possible to revise or supplement this method so that 0D calculations with selected scenarios may be sufficient to serve as the standard, but that is not discussed in this work. However, the method discussed here is a useful starting point, and is worth publishing for this reason.

AR: In the paper, 3-D simulations were conducted using implicit gas-phase mechanisms rather than explicit gas-phase mechanisms. The 3-D simulation results are used to provide a realistic range of conditions (for ozone and radical concentrations and meteorological conditions) to train semi-explicit SOA mechanisms with GENOA. As the 3-D results have been evaluated using measurement data ([Couvidat et al., 2018]), implicit gas-phase mechanisms can be utilized to simulate the concentrations of oxidants and major air pollutants (e.g., ozone, radicals, other inorganic pollutants), as well as environmental parameters (e.g., relative humidity and temperature). Also, as the reviewer pointed out, 3-D simulations with the full mechanism are too time-consuming. Therefore, 3-D simulations with implicit gas-phase mechanisms are sufficient to obtain near-realistic conditions on which to train the semi-explicit mechanisms.

The paper has been updated to include details about implicit gas-phase mechanisms of the 3-D CHIMERE model and rephrased how 3-D results are used for 0-D simulations:

Section 2.2, line 193:

The version of CHIMERE and its configuration is described in Lanzafame et al. (2022). The monthly average 3-D CHIMERE simulations were conducted with the implicit gas-phase MELCHIOR2 mechanism (Derognat et al., 2003), which contains 120 reactions and less than 80 lumped species. The MELCHIOR2 mechanism describes the degradation of sesquiterpenes by three oxidant initiated reactions (HUMULE reacts with OH, O3, and NO3, respectively), where the the species HUMULE represents the lumped class of all sesquiterpenes.

The monthly diurnal profiles of hourly meteorological data (e.g., temperature, relative humidity), and hourly concentrations of oxidants and oxidant, radical, and other inorganic species were extracted from each location. That information is required in the 0-D simulations with SSH-aerosol (see section 2.4) to reproduce SOA concentrations and compositions under near-realistic conditions. Since the reduced SOA mechanism focuses only on SOA formation, the meteorological data and the concentrations of oxidants, radicals and inorganics are assumed to remain intact during the 0-D SOA simulation. The coordinates and time of each condition are also provided to calculate the solar zenith angle. Because the reduction focuses on the impact on SOA variation, and because no inorganic reactions are considered in the reduced chemical mechanism, the oxidant, radical and inorganic concentrations are fixed as the hourly background, as well as the environmental parameters, are fixed to the diurnal profiles obtained from the CHIMERE data in 0-D SOA simulations. The concentration of HUMULE (denoted  $C_{SQT}$  as the CHIMERE surrogate for sesquiterpene(denoted  $C_{SQT}$ ) is used to estimate the SQT concentration.

- **RC:** The focus of this paper is a specific software package developed for mechanism reduction, which they call GENOA, though the main interest of this paper from a scientific perspective is the method itself. The name of the software is somewhat misleading because it is not actually a mechanism generator, but instead is a method to reduce existing mechanisms, potentially including those developed by actual mechanism generators such as GECKO.
- AR: The role of GENOA is more like a "generator of reduced mechanisms" than as a "generator of mechanisms". In our opinion, the term "generator" might be better than other terms such as "producer" or "reducer". However, it is true that the term "generator" could still be misleading. Thus, GENOA is always referred to as a "generator of reduced mechanisms," rather than a "generator" alone, as GENOA is designed to generate

**mechanisms from reduction, not from scratch.**

- **RC:** This could potentially be a useful tool for the atmospheric chemistry research community, if suitably documented and made publicly available. The fact that they include a users manual along with the Supplementary Information suggests that this is the intention of the authors. If it has sufficient flexibility, it could possibly be used for other criteria besides SOA predictions and perhaps even for other applications besides atmospheric modeling, such as, for example, combustion modeling or reducing large liquid-phase mechanisms. However, the users manual does not contain sufficient information to actually run the model for general applications, other than duplicating the results given in this paper. In particular, it would need information on how to interface this with output of existing 3D air quality models.
- AR: Thank you for the positive comments. Certainly, one of the goals of GENOA is to build the reduction algorithm that can be used for other modellers and even for other applications besides SOA prediction.

As explained previously, the monthly diurnal profiles of hourly meteorological data (e.g., temperature, relative humidity), and hourly concentrations of oxidant, radical, and other inorganic species were extracted from 3-D results. The user can, therefore, simply compute these profiles from 3-D results for GENOA reduction.

We have added the following explanation to the manual:

GENOA user's manual, section 4, page 15:

For example, to run GENOA over a specific domain and using specific 3D model results, the user has to construct the files described in section 2.2.2 to provide the monthly profiles for different variables.

**RC:** It is worth pointing out that the reduced mechanisms developed using this are strictly speaking reliable to give predictions to within the desired tolerances only for the airshed conditions used in its development. They developed this example using the conditions of all of Europe for a whole year, but it may be more practical to use this to optimize mechanisms for specific urban scenarios, for use in regulatory modeling. To be useful for this, the users manual would need to be improved so it can be used with other 3D models and modeling scenarios.

Although this paper is reasonably well written, it does have areas where improvements are needed before it is accepted for publication, and I have some suggestions. These are given below in approximate order of importance.

AR: In the example of BCARY reduction, the reduced "Rdc." mechanism was trained under training conditions that covered both high and low NOx regimes. The training condition "ADD2", for instance, received high NO concentrations from Milan, and thus is considered a typical urban condition. Consequently, the "Rdc." mechanism could be adequate for use over Europe or in specific urban scenarios, as it was developed under both low and high-NOx conditions and efficiently reproduces the SOA concentrations in most metropolises (e.g., Paris, Berlin).

Compared to the "Rdc." mechanism, undoubtedly, more reduction could be achieved if the SOA mechanism was trained specifically for urban scenarios using an urban-specific training dataset. In other words, as pointed out by the reviewer, training with fewer and more targeted conditions will optimize the resulting SOA mechanism (more accurate and smaller), which is intended for a small domain or specific scenarios.

**2. Improvements need to be done**

- **RC:** It looks relative changes in SOA concentrations are used as the criteria to test a reduction approach. This means that 5% error in a grid cell where almost no SOA is formed is given equal weights to 5% error in a grid cell with high SOA, where the model prediction is relatively more important. Wouldn't absolute error be a better criterion, or at least among the criteria employed? Shouldn't there be a cutoff to remove cells with very low SOA, or was this incorporated implicitly by the choice of testing scenarios?
- AR: Thank you for such an insightful suggestion. We used relative errors in the evaluation of the reduction, because they are sensitive to small changes and it is easy to set universal criteria for conditions with a wide range of SOA concentrations. Because sesquiterpene has high SOA yields, all 0-D simulations in the paper resulted in SOA concentrations consistently exceeding 1  $\mu g/m^3$  (with an initial BCARY concentration of 5  $\mu g/m^3$ ). Therefore, for BCARY reduction, there may be no need for a cut-off concentration.

We have added the information to the paper:

*Section 2.3, line 267:*

The initial BCARY concentration is taken equal to 5 set to five  $\mu g m^{-3}$  in order to ensure high SOA production the (SOA concentration is always greater than one  $\mu g m^{-3}$  at all evaluated conditions at all conditions).

However, for other precursors that have SOA yields lower than sesquiterpene, a cut-off concentration may be required for conditions with low SOA production, where large relative errors can occur as a result of low SOA concentrations.

The user manual has been updated to inform the user that there are several options that can be taken to avoid this issue:

GENOA user's manual, section 4, page 15:

It should be noted that reduced mechanisms may cause large errors when there is a low SOA concentration. The reason might not be directly linked to reduction of the mechanism performance, but to the evaluation criterion, based on relative errors, which naturally has large variations when the absolute value is small. There are several solutions to resolve this uncertainty:

- The user can set high initial concentrations of the studied SOA precursor in order to ensure high SOA production.
- The user may specify a threshold SOA concentration for condition selection in the training and pre-testing datasets.
- The user may also evaluate reductions using a different type of error.
- **RC:** The 3D model simulations of continental Europe employing the chemistry-transport model CHIMERE was used as the standard against which the reductions were compared. However, CHIMERE mechanism is not exactly the same as MCM, and I could not find an indication of whether the "CHIMERE surrogate for sesquiterpene" (line 198) is exactly the same as the MCM b-caryophyllene mechanism as used in this work. If that is the case, it should be stated explicitly when CHIMERE is first mentioned, since I couldn't find such a

statement in Lanzafame et al (2022). If it is not the case, then justification needs to be given as to whether this is an appropriate standard against which to test the reductions.

AR: As detailed in reply to one of the general comments, we have not used 3-D but 0-D SOA simulations as a reference to evaluate the reductions. In the 0-D simulation, the concentration profiles of ozone, radicals, and inorganics, as well as environmental parameters, were derived from the 3-D CHIMERE simulations.

In CHIMERE v2020r1, there are several gas-phase chemical mechanisms embedded. We have used the MELCHIOR2 mechanism [Derognat et al., 2003], which contains 120 reactions and less than 80 lumped species. Evidently, it is an implicit gas-phase mechanism that is highly simplified compared to the explicit MCM mechanism.

The species HUMULE in CHIMERE (the MELCHIOR2 mechanism) does not refer to a specific sesquiterpene, but to a lumped class that includes all sesquiterpene species. As we use "HUMULE" concentrations only to determine whether sesquiterpenes are present at a specific location and select several conditions for varying sesquiterpene concentrations in the pre-testing dataset, there is no consistency issue.

Relevant modifications have been done to section 2.2 line 19 of the paper, as mentioned previously.

- **RC:** It is not clear to me whether the concentrations of the inorganic species such as OH, O3, HO2, etc were constrained to be exactly what was calculated by CHIMERE in the 0D models for the selected locations, or if they were calculated using the mechanisms using boundary or initial conditions somehow obtained from CHIMERE. If the former (which I presume to be the case) this should be stated explicitly, and if the latter than more detail about the inputs to the 0D models need to be provided, if only in an Appendix.
- AR: Sorry for the ambiguity. As mentioned in reply to the previous comment, the concentrations of certain species (i.e., ozone, radicals, and other inorganics) are constrained to be exactly what was calculated in the 3-D CHIMERE simulations for the selected conditions (locations + month). The concentrations of other species resulting from sesquiterpene degradation are simulated using the 0-D SSH-aerosol model. As the CHIMERE concentrations represent 24-hour monthly averages, the diurnal profiles are repeated for five days in 0-D simulations (the simulation time is five days to consider adequately SOA formation and aging processes).

We have added the explanation of how the coupling is generally resolved between VOC species and oxidants in modeling:

**Section 1, line 36:**

Along with To complete implicit gas-phase mechanisms, implicit SOA mechanisms have been developed to model specifically the SOA formation without modifying the concentrations of ozone and major radicals (Kim et al., 2011). In 3-D modeling, implicit SOA mechanisms or parameterizations are usually added to implicit gas-phase mechanisms, conserving the oxidant chemistry of the implicit gas-phase mechanism.

Implicit SOA mechanisms are often established based on experimental data from smog chamber experiments to represent the formation and evolution of SOA, such as the two-product empirical SOA model (Odum et al., 1996) and the volatility basis set (VBS) that splits VOC oxidation products into a uniform set of volatility "bins" Donahue et al., 2006).

And the focus of our semi-explicit SOA mechanisms:

**Section 2, line 80:**

The generated semi-explicit mechanisms are designed to preserve the accuracy of explicit mechanisms for SOA formation, while keeping the number of reactions/species low enough to be suitable for large-scale modeling, particularly 3D in 3-D AQMs. The focus of the semi-explicit mechanism is solely on the accurate modelling of SOA. Because ozone, major radicals, and other inorganics are also affected by inorganic and other VOC chemistry, their concentrations are not tracked with the semi-explicit mechanism. Instead, they are simulated using existing implicit gas-phase chemical mechanisms.

- **RC:** The computation of stoichiometric coefficients and rate constants for new lumped species (as shown in Figure 2) based on concentrations and lifetimes need to be discussed. The use of "average produced concentrations from five-day 0D simulations" needs more discussion than as a footnote in the table. Presumably the [HO2], [NO], etc. concentrations used to compute the lifetimes are also averaged? What are the 5 0D simulations they use to compute the average? The later discussion indicates they use 8 scenarios for initial training, and more than that when they are close to being finished.
- AR: Sorry for the ambiguity. As mentioned in the previous reply, all 0-D simulations are conducted for five days in order to consider adequately SOA formation and aging processes. In the reduction by lumping, the weighting ratio is necessary to weigh the proportion of lumped molecules in the new molecule, as well as to compute the stoichiometric coefficients and rate constants in the new reaction. In order to calculate the weighting ratio, we tried different methods and eventually decided to compute it based on average concentration over the lifespan of the particle under training conditions. A five-day period is chosen as a compromise between a shorter period that may not reflect aging, and a longer period that may less adequately address the SOA formation. Considering the limited computational capacity, only one weighting ratio calculation method is utilized in the BCARY reduction.

To ensure coherency, all reduction parameters are calculated based on five-day 0-D simulation results of training conditions. For example, in the BCARY reduction, the lifetime of inorganics (e.g., HO2, NO) is calculated as the average of 16 simulations (two five-day simulations starting at 0h/12h for eight training conditions). Considering that inorganic concentrations repeat the diurnal profiles of 3-D simulations, the five-day average is equivalent to the daily average. This is different from the concentrations of sesquiterpene oxidation products, where a five-day average could take into account both formation and aging processes.

For clarity, the following explanation has been added to the paper:

**Section 2.1.1, line 126:**

As detailed in Table 3,  $f_w$  is computed as a function of chemical lifetime  $\tau$  following the computation of Seinfeld and Pandis (2016), and the reference concentrations  $C_r$  that are the arithmetic mean concentrations of a set from five-day 0D simulations of calculated from 0-D simulations using the explicit VOC mechanism<del>under</del>. Both  $\tau$  and  $C_r$  are based on averages of simulations across all training conditions.

Section 2.3, line 268:

Unless stated otherwise, a 5-day simulation is performed two simulations are performed for each condition starting at midnight (0 h) and noon (12 h)for each condition, focusing on , taking into account both the daytime and nighttime chemistry. All 0-D simulations are run for five days in order

to consider adequately SOA formation and aging processes.

- **RC:** The maps indicate that the reduction errors are the greatest in Southern Europe than in the North. Is that because there is more secondary SOA predicted for the South, which should be more photochemically reactive? It might be useful to show a relationship between SOA level predicted and the reduction error for the various grid cells or scenarios.
- AR: In order to illustrate the relationship between testing errors and SOA levels, we have added the SOA concentration map (Fig. 1) to the main paper, which corresponds to the results in the error map (Figure 3 of the paper):

Figure 1: SOA concentration (left panel) and testing error (right panel) maps of the testing results of the "Rdc." mechanism in summer (July and August).

We can see that SOA concentrations are generally higher under Nordic and North African conditions (The same initial BCARY concentration is set (5  $\mu$  g/m3 for all conditions). As only one photolysis reaction is preserved in the "Rdc." mechanism (248 are preserved in the original MCM), the "Rdc." mechanism is probably not be very sensitive to photochemical reactivity. As can be seen from the bar plot in Fig. 5 in the paper, the testing errors in July and August over Southern Europe are actually close to 3 %, although they seem to be higher (placed in the error category between 3 % and 6 %) than others. In northern Africa, the high error might be partially attributed to the photochemical reactivity, but the main cause is still the high temperatures and dry climate.

It should also be noted that although some conditions on the map appear to have large errors (e.g., in northern Africa), the results may still be acceptable as long as they adhere to the pre-testing criteria (average error  $\leq 3$ %, and maximum error  $\leq 20$ %). As the average SOA concentration map of all testing results may be too complex to interpret, it is not included in the main paper but in Appendix C for reference (Fig. 2):

**RC:** On line 169 they state that "jumping" is restricted to cases where the intermediate to be removed results in the formation of only a single compound. However, this is not the case for the examples they show on Table 4, where each intermediate forms more than one compound. (Reactions 14 and 15 in Table 5 reflect merged competing reactions, and are not explicit.) In fact, any rapidly reacting intermediates that have several reactions that form different products, can be "jumped" without affecting simulation results as long as their reactions are either unimolecular or with O2, so lifetimes would not vary with conditions. This is

---

## Author Response (AR2)

**Authors' Response to Reviews of**

**GENerator of reduced Organic Aerosol mechanism (GENOA v1.0): An automatic generation tool of semi-explicit mechanisms**

Zhizhao Wang, Florian Couvidat, Karine Sartelet
*Geoscientific Model Development,* `https://doi.org/10.5194/egusphere-2022-245`
* * *
EC: *Editors' Comment*,   RC: *Reviewers' Comment*,   AR: Authors' Response,   ☐ Manuscript Text

**1. Uploaded files validated on 05 Oct 2022**

Notification to the authors:

Please ensure that the colour schemes used in your maps and charts allow readers with colour vision deficiencies to correctly interpret your findings. Please check your figures using the Coblis – Color Blindness Simulator (https://www.color-blindness.com/coblis-color-blindness-simulator/) and revise the colour schemes accordingly.

AR: All figures have been checked. As a result, we have revised figures 3, 5, 7, 11 in the main paper and figures A1, B1 in the Appendix.

**2. Topical Editor decision: Publish subject to minor revisions (review by editor) on 28 Oct 2022**

EC: *Dear Authors,*

*I am pleased to inform you that your revised manuscript is accepted for publication after some minor/technical revisions. Please find below a list of final reviewer comments. Please revise your manuscripts accordingly or write a reply in case you consider the comments as not appropriate.*

*Best regards,*
*Andrea Stenke*

AR: Thank you editor and reviewers for the positive comments. We have carefully considered the following comments and revised the manuscript accordingly. Please note that the line and section numbers mentioned in the response correspond to the version of the manuscript before revision.

RC: *My most substantive concern involves the fact that they apparently did not consider (or did not state that they considered) differences in volatility among their acceptance criteria for lumping. It seem to me that this should be the most important criterion given that this work is focused on SOA modeling. The first additional example given in Section C1 involves lumping a compound with an aldehyde group with one with a carboxylic acid group. I would think that these groups would have substantially different effects on vapor pressures and therefore SOA formation impacts.*

AR: In BCARY reduction, the difference in saturation vapor pressure was indeed not explicitly considered when lumping condensables. However, when searching for a lumpable target for one condensable, GENOA ranks all other condensables according to their difference in saturation vapor pressure. In other words, species with similar saturation vapor pressures are lumped together first. Meanwhile, saturation vapor pressures, which result from molecular structures, may also be constrained by restricting some key molecular structures of lumpable species. Therefore, the lumping of condensables may take into consideration the difference in vapor pressure to some extent.

The lumping examples presented in Appendix C1 are intended to demonstrate the possibility of lumping dissimilar compounds (with real examples from BCARY reduction). For this reason, we have selected examples of lumped species with distinct molecular structures. Despite the first example in Appendix C1 having different functional groups, the difference in their saturation vapor pressure is only around two orders of magnitude: $1.5 \times 10^{-12}$ atm for C1313NO3 and $2.8 \times 10^{-14}$ atm for C152NO3.

Additionally, GENOA offers a user-chosen option to lump condensables based on their saturation vapor pressure, allowing lumping only if their saturation vapor pressures are within the user-specified log-scale range. The BCARY reduction was tested with this option, however, a more effective reduction was obtained without it (which is the "Rdc." mechanism described in the paper). The result could indicate that BCARY, which typically derives low volatile organic compounds, may not need a stringent restriction on saturation vapor pressure for lumping (i.e., current limitations may be sufficient).

Nevertheless, we totally agree with the reviewer that saturation vapor pressure plays a crucial role in SOA formation, and this lumping option could be beneficial for other SOA precursors. In order to inform the user, we have included this lumping option (although not used in BCARY reduction) in the supplementary Excel file titled "User-chosen reduction options and parameters". We have also added the explanation to the paper:

> *Section 2.2.3, line 221:*
>
> The difference in saturation vapor pressure between lumpable condensables is not explicitly restricted in BCARY reduction. However, it is implicitly considered, as GENOA searches and attempts to lump species with similar saturation vapor pressures first. Nonetheless, the user can activate the option to limit the range of saturated vapor pressure differentials between lumpable condensables, along with other user-chosen reduction options listed in the supplementary material.

RC: *On line 130 they define "extremely fast reaction" suitable for pre-reduction as one with a reaction rate grater than $10^6 \ sec^{-1}$. However, rates have units of concentrations per unit time, not $time^{-1}$. They should state that this is a unimolecular or pseudo-unimolecular rate constant.*

AR: Sorry for the misleading. We have corrected it in the paper:

> *Section 2.1, line 130:*
>
> This process skips extremely fast unimolecular reactions (i.e., the reaction rate constant of $10^6 \ s^{-1}$ corresponding to a lifetime of 1 $\mu$s) to avoid numerical problems.

RC: *On line 183 they call the MCM species BCALOO an alkoxy radical. It actually is supposed to represent a Criegee intermediate.*

AR: Thank you for pointing out the mistake. We have corrected it in the paper:

> *Section 2.2.2, line 182:*
>
> As shown in Table 2, the  Criegee intermediate BCALOO formed during the ozonolysis of BCAL (reaction No. 11 in Table 1) is jumped over to its only destruction product BCLKET.

**RC:** *Couldn't the "jumping" example on Table 2 have been done at the "pre-reduction" step since it involves no change in predicted organic species? It does involve changes in inorganics, but as pointed out their reduced mechanisms are only used for SOA, which depends only on organic product predictions.*

**AR:** Although both pre-reduction and reduction via jumping are capable of jumping reactions without altering organics, they serve different purposes and act on different reaction pathways.

This pre-reduction is not intended to reduce the mechanism, but rather to provide a reliable reference mechanism. Therefore, it only involves very fast degraded species that undergo a single unimolecular reaction with a constant kinetic rate coefficient (e.g., no temperature effect). As these reactions are extremely fast (i.e., rate constant of $10^6$ s$^{-1}$) and independent of environmental conditions, they only cause numerical issues in simulation and should be removed from the reference mechanism.

In contrast to pre-reduction, jumping is used to search for all possible reductions, which may involve reactions that are relatively slow or affected by environmental conditions. It is necessary to validate the reduction with an evaluation. As shown in Table 2, even though BCALOO degrades into a single species BCLKET, the process involves five bimolecular reactions that may vary depending on the environment (i.e., BCLKET-derived SOA may be affected by variations in inorganic concentrations and relative humidity). As a result, the example in Table 2 may be more appropriate for jumping than pre-reduction and requires further evaluation.

A few sentences have been added to the paper to clarify the difference between jumping and pre-reduction:

> *Section 2.2.2, line 188:*
>
> There are similarities between reduction by jumping and pre-reduction in the sense that both can jump reactions without affecting organic compounds. However, the two processes serve different purposes, as pre-reduction is intended to provide a reliable reference mechanism for training, whereas jumping is used in training to search for possible reductions. On the one hand, the current pre-reduction only reduces very fast degraded species that undergo a single unimolecular reaction with a constant kinetic rate coefficient (e.g., no temperature effect). In this case, one species may lead to several degradation products. As these reactions are extremely fast and independent of atmospheric conditions, they only cause numerical issues in simulation and should be removed from the reference mechanism. On the other hand, jumping may be relatively slow or affected by environmental conditions, and therefore, an evaluation is necessary. Jumping is currently limited from one species to another at a time. As shown in Table 2, the degradation of BCALOO into BCLKET involves five bimolecular reactions, which may affect SOA formation under different atmospheric conditions (e.g., with different inorganic concentrations and relative humidity (RH)).

**RC:** *Section C1 says figure C1 has the structures of the nitrates in the first lumping example, but I could not find them in this figure. I also could not find the structure of the species mentioned in the second example there.*

**AR:** Sorry for the mistake. We have added the molecular structures of C1313NO3, C152NO3, C1310OH, and BCALBOC to Fig. 1 (Fig. C1 in the appendix).

[Figure]

Figure 1: Molecular structures of the MCM species that are mentioned in the paper. For more information, please visit the MCM website.

**RC:** *The example shown on Tables 3 and 4 could be applied without conducting model simulations if the chemical lifetimes of all the species are the same, since the weighting ratios would then just be the branching ratios for formations of the 3 radicals. They are close to the same because the RO2+RO2 reaction is the only one where rate constants are different, and this is usually a very minor process compared to the competing reactions (see Figure 2), and the "removal" process should have removed these reactions before going to the "lumping" step. Maybe having an additional step between "removal" and "lumping" should be done to see what lumping can be done independent of environmental conditions."*

AR: Tables 4 and 5 demonstrate lumping and replacing (Original reactions in Table 3), which are not the real cases in the BCARY reduction, but were chosen for ease of explanation. Currently, all reduction strategies, including lumping, allow only a reduction of up to two species at a time. During the actual BCARY training, BCAO2, BCBO2, and BCCO2 were merged into one surrogate with two reductions: first, BCAO2 and BCBO2 were lumped together into mBCAO2 in a reduction via lumping, then BCCO2 was replaced by mBCAO2 in a reduction via replacing.

As the reviewer pointed out, it is logical to lump the MCM species BCAO2, BCBO2, and BCCO2 together due to their similar structures and reactions. Their weighting ratios for lumping can be derived directly from their branching ratios in the OH reaction of BCARY (Reaction No.17). In fact, we have tested this reduction, and it only introduced an average error of less than 0.1% (max error of 0.64%) on the testing dataset, indicating that it is a highly effective reduction.

Although the current training can lead to a similar result with two reductions, extra errors might be introduced due to the computation of the weighting ratios. Therefore, we agree with the reviewer that this type of lumping, which is solely based on species' theoretical similarity, can be an effective extension of the current lumping strategy. It may allow the lumping of multiple species at the same time and not compute weighting ratios based on the selection of the training dataset. Investigations are required to determine the general criteria for this new type of lumping.

We have added the explanation to the paper:

> *Section 2.2.3, line 209:*
>
>  Chemical lifetimes and reference concentrations may be close for species that share similar structures and undergo analogous reactions. In cases where these species originate from the same reaction, they can be lumped directly, with the branching ratios of the formation reaction serving as weighting ratios. As an example, BCAO2, BCBO2, and BCCO2 undergo equivalent reactions, with the exception of the $RO_2$ reaction of BCBO2. Since the BCARY degradation is not much sensitive to $RO_2$, BCAO2, BCBO2, and BCCO2 can be lumped together with $f_{w,a}$, $f_{w,b}$ and $f_{w,c}$ equal to the branching ratios of reaction No.17, i.e., 0.408, 0.222, and 0.37, respectively.
>
> Most lumping involves species that are not isomers and undergo different reactions, which makes lumping multiple species at the same time highly uncertain. Therefore, in practice, GENOA attempts to lump only two species in a single reduction in order to ensure  the effectiveness of computation. A lumping of multiple species can be achieved by combining several reductions (e.g., first lumping BCAO2 with BCCO2 to form mBCAO2, and then lumping BCBO2 into mBCAO2).